# WNT-dependent interaction between inflammatory fibroblasts and FOLR2+ macrophages promotes fibrosis in chronic kidney disease

Camille Cohen[1,2], Rana Mhaidly[1,2,10], Hugo Croizer[1,2,10], Yann Kieffer ⬤[1,2], Renaud Leclere ⬤[3], Anne Vincent-Salomon ⬤[3], Catherine Robley[1,2], Dany Anglicheau ⬤[4], Marion Rabant[5], Aurélie Sannier[6], Marc-Olivier Timsit[7], Sean Eddy ⬤[8], Matthias Kretzler[8,9], Wenjun Ju ⬤[8,9] & Fatima Mechta-Grigoriou ⬤[1,2] ✉

Chronic kidney disease (CKD) is a public health problem driven by myofibroblast accumulation, leading to interstitial fibrosis. Heterogeneity is a recently recognized characteristic in kidney fibroblasts in CKD, but the role of different populations is still unclear. Here, we characterize a proinflammatory fibroblast population (named CXCL-iFibro), which corresponds to an early state of myofibroblast differentiation in CKD. We demonstrate that CXCL-iFibro co-localize with macrophages in the kidney and participate in their attraction, accumulation, and switch into FOLR2+ macrophages from early CKD stages on. In vitro, macrophages promote the switch of CXCL-iFibro into ECM-secreting myofibroblasts through a WNT/β-catenin-dependent pathway, thereby suggesting a reciprocal crosstalk between these populations of fibroblasts and macrophages. Finally, the detection of CXCL-iFibro at early stages of CKD is predictive of poor patient prognosis, which shows that the CXCL-iFibro population is an early player in CKD progression and demonstrates the clinical relevance of our findings.

Chronic kidney disease (CKD) is a significant public health burden, affecting 9.1% of the global population, and accounting for 2.6 million deaths per year[1]. As CKD is becoming one of the leading causes of death, there is an urgent medical need to develop efficient therapeutic strategies and thus to better understand the molecular mechanisms underlying CKD progression. CKD results from the development of interstitial fibrosis, which results from the elevated production of extracellular matrix (ECM) proteins by myofibroblasts[2]. Important efforts have been made to decipher the origin(s) of myofibroblast in kidney fibrosis. Several pieces of evidence previously demonstrated

[1]Institut Curie, Stress and Cancer Laboratory, Equipe labélisée par la Ligue Nationale contre le Cancer, PSL Research University, 26, rue d'Ulm, F-75248 Paris, France. [2]Inserm, U830, 26, rue d'Ulm, Paris F-75005, France. [3]Department of Diagnostic and Theragnostic Medicine, Institut Curie Hospital Group, 26, rue d'Ulm, F-75248 Paris, France. [4]Department of Nephrology and Kidney Transplantation, Necker Hospital, AP-HP, Paris Cité University, Inserm U1151, 149 rue de Sèvres, 75015 Paris, France. [5]Department of Pathology, Necker Hospital, AP-HP, Paris Cité University, 149 rue de Sèvres, 75015 Paris, France. [6]Department of Pathology, AP-HP, Bichat-Claude Bernard Hospital, Paris Cité University, Inserm, U1148, 46, rue Henri Huchard, 75877 Paris, France. [7]Department of Urology, Européen George Pompidou Hospital, APHP, Paris Cité University, Paris, France. [8]Department of Internal Medicine, University of Michigan, Ann Arbor, MI 48109, USA. [9]Department of Computational Medicine and Bioinformatics, University of Michigan, Ann Arbor, MI 48109, USA. [10]These authors contributed equally: Rana Mhaidly, Hugo Croizer. ✉e-mail: fatima.mechta-grigoriou@curie.fr

that myofibroblasts mainly originate from resident kidney interstitial cells, with several possible lineages, such as pericytes[3–7]. Nevertheless, the intermediate step of differentiation toward myofibroblasts, as well as the cellular interactions leading to their differentiation, are not yet fully understood. Recently, single cell transcriptomic data from human and mouse kidney showed a significant heterogeneity in fibroblast populations during kidney fibrosis development[8,9]. Nevertheless, the role of these different fibroblasts in the differentiation process of myofibroblasts, in their interactions with surrounding cellular populations and thus in the development of the disease remains poorly known. Thus, better characterization of these different fibroblast populations in kidney disease is a pre-requisite to better understand this pathology and provide innovative treatments to patients.

In cancer, the molecular heterogeneity and functional diversity of fibroblasts (referred to as Cancer-Associated Fibroblasts, or CAF) have been highlighted in recent years. Indeed, several CAF populations have been identified in different cancer types by combining the study of several CAF markers, such as Fibroblast Activation Protein (FAP), Smooth Muscle-α Actin (SMA) and Regulator of G protein signaling 5 (RGS5)[10–24], later confirmed with the development of single cell RNA sequencing (scRNAseq)[14,17,20,21,25–33]. These CAF populations are phenotypically and functionally heterogeneous. FAP + SMA + RGS5- (also referred to as CAF-S1) fibroblasts are characterized by an accumulation of ECM proteins and inflammatory signatures, while FAP- SMA + RGS5+ (CAF-S4) fibroblasts are defined by a perivascular signature[15,16,18,19,25,34]. Among the CAF-S1 population, inflammatory CAF (iCAF) and myofibroblastic CAF (myCAF) were first discovered in pancreatic cancer[13,26,35] and have now been confirmed in a high diversity of other cancer types[17,20,24,29,34,36,37]. iCAF are characterized by the secretion of inflammatory mediators, while myCAF exhibit a high expression of ECM proteins but a lack of inflammatory cytokines. One of the most resolutive single cell datasets of the CAF-S1 population from breast cancer has highlighted that this population is composed of 8 distinct cellular clusters, including 3 iCAF and 5 myCAF clusters[29]. Interestingly, specific myCAF clusters (i.e., ECM-myCAF, TGFβ-myCAF and Wound-myCAF clusters) are associated with primary resistance to immunotherapy in metastatic melanoma and in non-small cell lung cancer[28,29,35].

In the current paper, we take advantage of the detailed characterization of CAF populations in cancer to highlight fibroblast heterogeneity all along kidney disease progression. Indeed, by combining study on human samples using cutting-edge technologies and functional assays, we observe an important heterogeneity in fibroblast populations during kidney fibrosis development. More precisely, we are focusing on a population of inflammatory fibroblasts (referred to as CXCL-iFibro), which represents an intermediate state in the differentiation process towards ECM protein-secreting myofibroblasts. By combining single cell analysis, spatial transcriptomics and in vitro functional assays using primary fibroblasts isolated from patients, we provide some clues suggesting that these pro-inflammatory CXCL-iFibro fibroblasts might attract and activate FOLR2+ macrophages. In turn, FOLR2+ macrophages participate in the differentiation of CXCL-iFibro into ECM-secreting myofibroblasts through a WNT/β-catenin-dependent pathway. Finally, by analyzing transcriptomic data from a cohort of CKD patients, we show that the presence of CXCL-iFibro at early stages of CKD is associated with poor patient outcomes. Altogether, by identifying a new population of inflammatory fibroblasts (CXCL-iFibro) and its interaction with FOLR2+ macrophages, our study unravels a new mechanism driving CKD progression, and identifies a potential therapeutic target to limit its progression.

## Results

### Single cell RNA sequencing identifies different clusters of mesenchymal cells in kidney disease
To assess fibroblastic heterogeneity during kidney fibrosis development, we took advantage of an existing human single cell RNA sequencing (scRNAseq) dataset from patients with or without chronic kidney disease (CKD)[8]. After quality control and doublet elimination, 2908 mesenchymal cells were selected for further analyses. Unsupervised graph-based clustering identified 13 clusters, visualized with the Uniform Manifold Approximation and Projection (UMAP) algorithm (Fig. S1a). To simplify our message and increase the relevance of our comparisons, we focused on the first 6 clusters, as they represented more than 85% of the total mesenchymal population (Fig. 1a), while the other clusters contained less than 80 cells per cluster (Fig. S1b). All clusters were found in at least 2 patients, albeit in different proportions from one patient to the other (Fig. 1b and Fig. S1b). We confirmed that these different clusters were differently distributed between control and CKD patients (Fig. 1c, d). Indeed, the content in clusters 0, 3, 4, and 5 was increased in CKD patients, while the clusters 1 and 2 accumulated in controls (Fig. 1d). We first confirmed that almost all cells expressed *VIM (VIMENTIN)* and *PDGFRB*, confirming their mesenchymal origin (Fig. 1e). We also observed that high expression of *RGS5* (Regulator of G protein signaling 5) and *NOTCH3*, specific pericyte markers, were mainly detected in clusters 1 and 2, while the pan-fibroblast marker genes *PDGFRA* and *DCN (DECORIN)* were expressed in clusters 0, 3, 4, and 5 (Fig. 1e). We thus hypothesized that clusters 1 and 2 might be pericytes-like and clusters 0, 3, 4, 5 fibroblasts-like. Differential gene expression and functional enrichment analysis confirmed this assumption and revealed that each cluster was also characterized by a specific transcriptional profile (Table S1). Although all the fibroblastic clusters were associated with extracellular matrix (ECM) remodeling pathways, each cluster was associated with specific processes: cluster 0 was also characterized by cytokine signaling pathways and inflammation, cluster 3 by response to wounding, cluster 4 by complement and coagulation cascades, and cluster 5 by translation elongation and wound-healing, among top-ranked pathways in each cluster (Table S1). In addition, the pericyte-like cluster 1 was associated with IFNα/β-dependent signaling and blood vessel development, and cluster 2 with muscle system process and oxidative phosphorylation (Table S1). In agreement with these observations, *CXCL12* was highly expressed in cluster 0, and the expression of *COL1A1, COL3A1,* and *POSTN* (some of the main ECM components in fibrosis and marker of myofibroblasts) was mainly detected in clusters 3 and 5 (Fig. 1f).

Interestingly, we found that the aforementioned pathways detected in fibroblasts isolated from kidney diseases were highly reminiscent of those previously identified in specific CAF populations in cancer[14–17,20,21,25–32,35]. Indeed, the CAF-S1-specific gene signature[29], which identifies populations of inflammatory and myofibroblastic FAP + CAF associated with metastatic spread and immunosuppression[15,16,18,19,29] was highly detected in the fibroblastic clusters (clusters 0, 3, 4, 5) in the kidney disease dataset, while the CAF-S4 signature (perivascular-like CAF, gene signature in Table S2) highlighted the pericyte-like cells (clusters 1, 2) (Fig. 1g). On the one hand, the signatures from detox-iCAF, IL-iCAF and IFNγ-iCAF subsets previously identified by scRNAseq in the CAF-S1 (FAP + CAF) population from breast cancer[29] highlighted mainly the clusters 0 and 4 (Fig. 1h), thereby confirming that these clusters were inflammatory. On the other hand, the ECM-myCAF, TGFβ-myCAF and Wound-myCAF signatures from CAF-S1 highlighted the clusters 3 and 5 (Fig. 1g). Interestingly, all the gene signatures from CAF-S1 fibroblasts, but not CAF-S4, showed an increased expression in CKD compared to controls (Fig. 1g, h), in total agreement with accumulation of clusters 0, 3, 4 and 5 (fibroblast-like) but not of clusters 1 and 2 (pericyte-like) in CKD patients compared to controls (as shown in Fig. 1c, d). We next sought to focus our further analyses on these clusters 0, 3, 4, and 5, which accumulate in CKD patients. To define specific markers of these clusters, we performed a pairwise analysis of the genes differentially expressed between the different clusters and we identified SFRP1 (Secreted frizzled Related Protein 1) as significantly up-regulated in

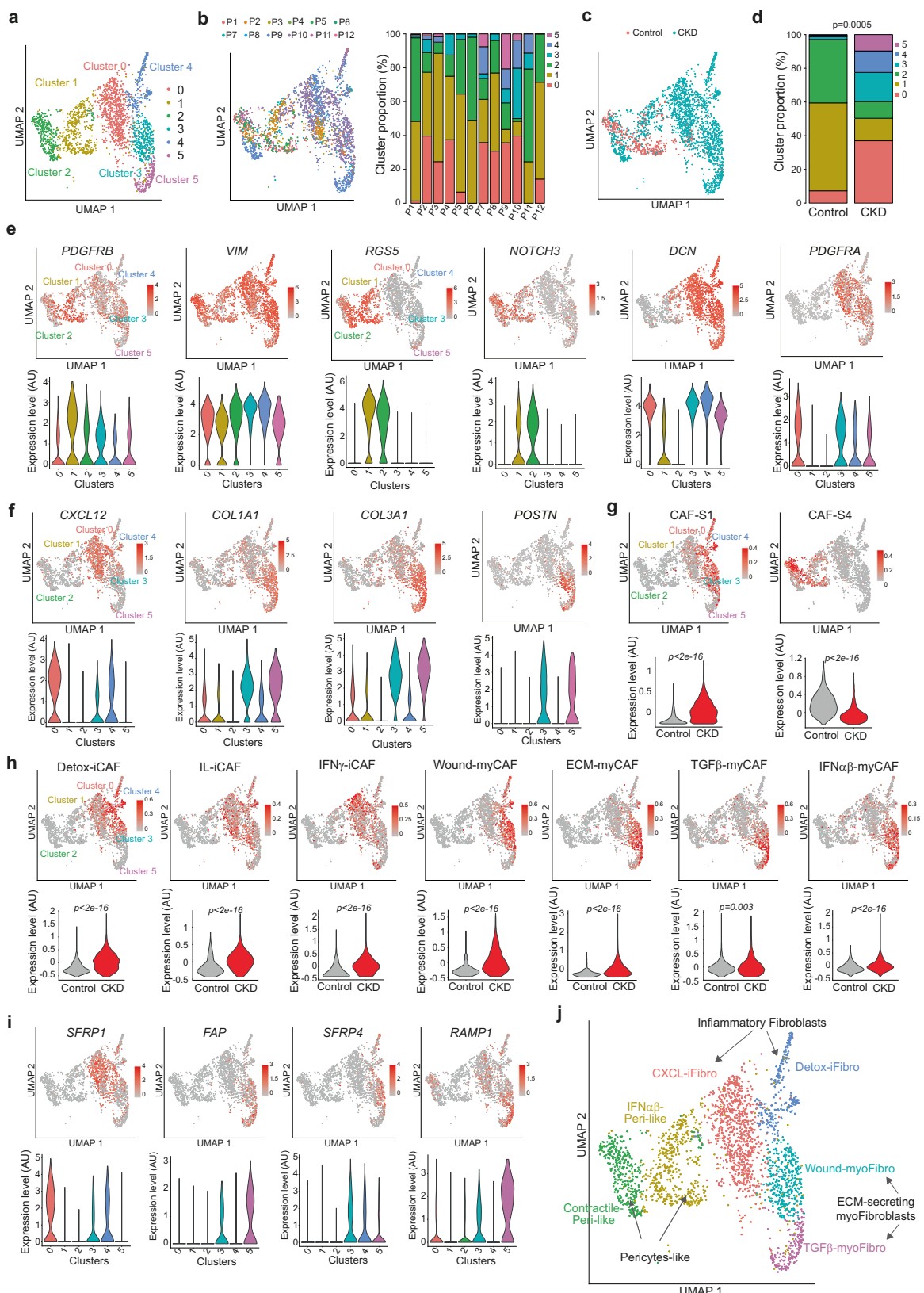

cluster 0 (although a few cells in clusters 3 and 4 also expressed SFRP1), FAP (Fibroblast Activation Protein) expressed in both clusters 3 and 5, SFRP4 (Secreted frizzled Related Protein 4) in clusters 3 and 4 and RAMP1 (Receptor Activity Modifying Protein 1) in cluster 5 (Fig. 1i and Fig. S1c). By this way, cluster 0 could be identified as SFRP1 + SFRP4- FAP- RAMP1-; cluster 4 as SFRP1 ± SFRP4 + FAP- RAMP1-; cluster 3 as

SFRP1 ± SFRP4 + FAP + RAMP1- and cluster 5 as SFRP1- SFRP4- FAP + RAMP1 + . Taken as a whole, by combining the transcriptomic profiles of these different clusters, their gene similarities with CAF-S1 clusters, and their expression of specific markers, we defined the CKD clusters 0 and 4 as inflammatory (and thus referred to as iFibro), and the CKD clusters 3 and 5 as ECM-secreting myofibroblasts (then termed

**Fig. 1 | Identification of distinct mesenchymal clusters in CKD. a** UMAP of scRNAseq data from 2495 mesenchymal cells[8] across 12 patients suffering or not from CKD, allowing the visualization of the first 6 clusters (clusters 0 to 5). Colors show the different clusters defined by graph-based clustering method. **b** Left, Same UMAP as in (**a**) showing cell repartition across patients (P1 to P12). Right, Barplot representing the proportion of cells of the first 6 clusters (clusters 0 to 5) in each patient (P1 to p12). **c** Same UMAP as in (**a**) showing cell repartition according to disease status (Control in red for patients without CKD; CKD in blue for patients with chronic kidney disease). **d** Bar plot showing the percentages of the different clusters according to disease status, i.e., Control or CKD (N = 6 and 6, respectively). *P*-value from two-sided Fisher Exact test. **e** UMAP (top) and Violin plot (bottom) showing expression of marker genes according to the different clusters for mesenchymal cells (*PDGFRB*, *VIM*), pericytes (*RGS5*, *NOTCH3*) and fibroblasts (*DCN*, *PDGFRA*). **f** UMAP (top) and Violin plot (bottom) showing expression of representative genes for the different clusters. **g** UMAP (top) and Violin plot (bottom) showing the average z-score of genes that compose specific signatures of CAF-S1[29] and CAF-S4 (Table S2). Statistical test = two-sided Fisher Test. **h** Same as in (**g**) with the different CAF-S1 clusters identified in[29]. Statistical test = two-sided Fisher Test. **i** UMAP (top) and Violin plot (bottom) showing expression of representative genes for each cluster identified in CKD. **j** UMAP showing new annotations (defined by differential gene expression pathways) of the 6 clusters (0 to 5) identified in Control and CKD patients.

myoFibro). Moreover, based on their expression profiles (Table S1), we annotated CKD clusters more precisely as follows: cluster 0, CXCL-iFibro; cluster 4, Detox-iFibro; cluster 3, Wound-myoFibro and cluster 5, TGFβ-myoFibro. The pericyte-like clusters 1 and 2, which were down-regulated in CKD, were referred to as IFNα/β-Peri-like and Contractile-Peri-like, respectively (Fig. 1j). Taken as a whole, these data show an important heterogeneity of fibroblasts in CKD, which is reminiscent of the different CAF populations recently identified in cancer.

As Kuppe et al. described several clusters of mesenchymal cells using PDGFRβ+ sorted cells[8], we sought to identify the similarities between our fibroblast annotations and those from this study (as reported in Fig. S1d). Hence, we performed a label transfer analysis on PDGFRβ+ sorted cells from[8]. The dataset of 2495 mesenchymal cells with our own annotations (described in Fig. 1j) was used as a reference, and the dataset of PDGFRβ+ sorted cells (described in Fig. S1d), with original annotations from[8] was used as a query. Interestingly, the pericyte-like populations that we identified (IFNαβ-Peri-like and Contractile-Peri-like) showed a high degree of similarity with pericytes and vascular smooth muscle cells described by Kuppe et al (Fig. S1e and Table S3). Similarly, the Detox-iFibro we described corresponded to the Fibroblast 1 population that Kuppe described as being a SCARA5 + MEG3+ non-activated fibroblasts, and the TGFβ-myoFibro was similar to the Myofibroblast 1, the population exhibiting the highest level of ECM protein secretion. In contrast, CXCL-iFibro, and to a lesser extent Wound-myoFibro were identified as a mix of different fibroblast and/or myofibroblast populations, according to original annotations. In particular, the CXCL-iFibro population that we identified as being a homogenous population was dispersed throughout several clusters in the Kuppe et al. study and characterized as a mix of Fibroblast 2a (30.5% of cells), Myofibroblast 3a (22.1%), Myofibroblast 3b (17.2%), Fibroblast 2b (14.9%) and Myofibroblast 2b (12.3%) (Fig. S1e and Table S3). Thus, annotating mesenchymal cells based on similarities with CAF allowed us to identify a population of fibroblasts with inflammatory properties that has not been explored further previously.

## CXCL-iFibro and ECM-myFibro clusters accumulate at early and late CKD stages, respectively

Inflammatory fibroblasts (iFibro) have been poorly described in kidney fibrosis. To further our understanding, we performed trajectory inference and pseudotime analysis using Monocle 3 (Fig. 2a–c). Despite a longstanding debate regarding the origins of myofibroblasts in the kidney, several reports identified pericytes, as a major source of myofibroblasts in kidney fibrosis development[3–8]. Therefore we defined the root of the pseudotime in pericytes-like mesenchymal cells. These analyses indicated that CXCL-iFibro (cluster 0) might be an intermediate stage in the differentiation from pericyte-like cells (clusters 1 and 2) toward ECM-secreting myoFibro (clusters 3 and 5) (Fig. 2a, b). The expression of specific marker genes according to pseudotime confirmed this finding (Fig. 2c), and our observations in human CKD were consistent with previous lineage tracing experiments performed in mouse CKD models[3,5–7]. To validate the temporal dynamics of inflammatory fibroblasts and ECM-secreting

myofibroblasts, we took advantage of publicly available dataset. First, we analyzed bulk RNAseq data from mice undergoing unilateral ureteral obstruction (UUO) at different time points[38]. UUO is a classical and well characterized model of kidney fibrosis in mice, and we analyzed samples from control mice, and from mice 3 days after UUO (early time point with infiltration but no fibrosis), 7 days after UUO (intermediate level of fibrosis) and 14 days after UUO (with a high level of interstitial fibrosis). First, we calculated an ECM score characterized by the expression of ECM-related genes as described in Naba et al.[39], which allowed us to validate that this score increased progressively from day 0 to day 14 after UUO (Fig. 2d), and confirm the progressive increase in interstitial fibrosis. Then, to calculate the proportion of each cell type according to each time point, we performed a deconvolution of the bulk RNAseq data using BayesPrism[40]. To do so, we first built a comprehensive cellular atlas (Fig. 2e) based on scRNAseq dataset from CKD and normal kidney tissues[8]. This cellular atlas was composed of 49 226 cells corresponding to 9 different cell types (Fig. 2e). Interestingly, deconvolution results showed that kidneys from control mice were composed of mainly proximal tubular epithelial cells (around 80%), whereas after injury the proportion of proximal tubular cells decreased (Fig. 2f). This result validated our approach, as it is largely known that the cellular composition of a normal mouse kidney is approximately 70-80% of proximal tubular epithelial cells, and that this proportion significantly decreases after injury[41]. In addition, we observed a progressive increase in the proportion of mesenchymal cells, as well as of the immune compartment and injured tubules (Fig. 2g). To have more insights into the mesenchymal compartment, we focused our next analysis specifically on these cells. While pericytes were not detected in high proportion, we observed a gradual switch from inflammatory fibroblasts to ECM-secreting myofibroblasts from day 3 to day 14 (Fig. 2g). Interestingly, inflammatory fibroblasts were virtually absent at day 14 (Fig. 2g). Finally, we confirmed our observations from pseudotime analysis in patients (Fig. 2a–c) as we saw inflammatory fibroblasts arising early after injury (day 3), being maintained at intermediate stages (day 7), before disappearing at advanced stages of interstitial fibrosis (day 14) (Fig. 2h), meanwhile the proportion of ECM-secreting myofibroblasts increased progressively up to day 14 (Fig. 2i). Finally, to increase the resolution of our analysis, we analyzed a single cell RNAseq dataset from mice who underwent UUO, with available data for sham-operated mice, UUO day2, UUO day7 and reversal UUO. Reversal UUO corresponds to the reimplantation of the ureter in the bladder after 7 days of UUO to observe a healing phase, where a decrease in Collagen deposition, myofibroblast activation and macrophage infiltration was observed[42]. To assess the temporal dynamics of inflammatory fibroblasts in this model, we performed a label transfer analysis using our cellular atlas as reference. Interestingly, we observed a progressive increase in the proportion of inflammatory fibroblasts from day 2 after UUO to day 7 after UUO, followed by a decrease in the reversal UUO (Fig. 2j), confirming that inflammatory fibroblasts are associated with fibrosis expansion. Altogether, these data identify a population of inflammatory fibroblasts, which arises early after kidney injury, before decreasing at late stages when ECM-secreting myofibroblasts expand.

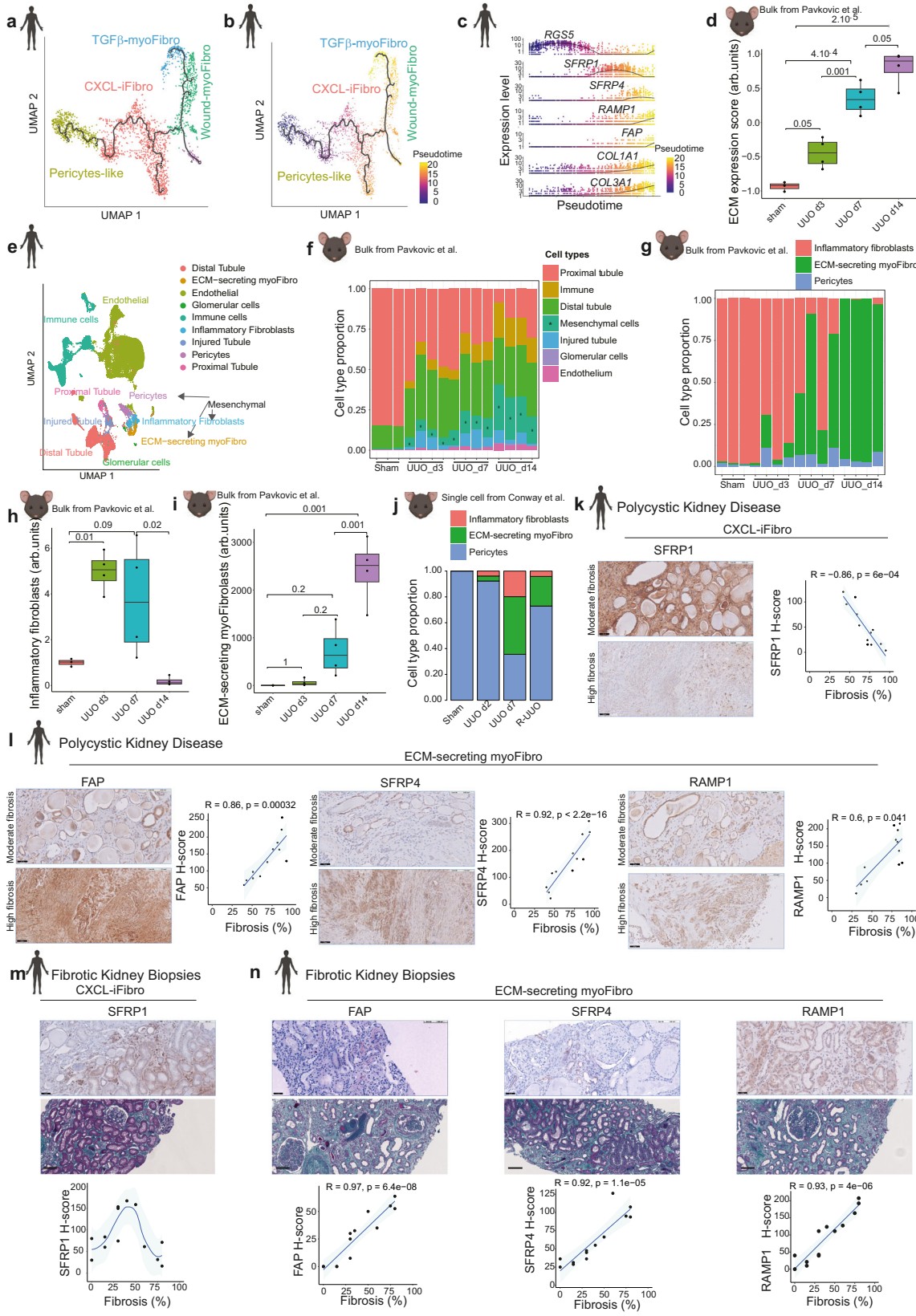

We next sought to validate the presence of these different clusters of fibroblasts in human tissue sections at different stages of the disease (Fig. 2 k–n and Fig. S2a, b for specificity of staining). In order to capture the fibroblast heterogeneity highlighted in scRNAseq data, we first focused on explants from patients with polycystic kidney disease (PKD), an autosomal dominant genetic disease causing cyst

development and ultimately leading to end-stage renal disease (Fig. 2k, l). PKD kidneys exhibited very large fields of fibrosis, which characterized the terminal stage of kidney fibrosis development. Immunohistochemistry (IHC) on PKD explants targeting CXCL-iFibro (SFRP1), ECM-secreting myoFibro (FAP), Wound-myoFibro (SFRP4) and TGFβ-myoFibro (RAMP1) validated the existence of these different

**Fig. 2 | Accumulation of CXCL-iFibro and ECM-myFibro clusters at distinct stages of chronic kidney disease. a** UMAP showing trajectory inference using Monocle 3. **b** Same UMAP as in (**a**) showing computed pseudotime by Monocle 3. **c** Expression of genes of interest according to Monocle 3 pseudotime. **d** ECM-score calculated on the kidney bulk RNAseq data from mice undergoing UUO at different time points (n = 15 mice). Statistical test = two-sided Mann-Whitney U-test with Benjamini-Hochberg adjustment. **e** UMAP of scRNAseq data from 49 226 cells from[8] with annotation used for bulk RNAseq deconvolution. **f** Results of the deconvolution of bulk RNAseq data from UUO mouse model. **g** Same as (**f**) but in mesenchymal cells. **h** Proportion of inflammatory fibroblasts estimated by deconvolution after UUO relative to control mice (n = 15 mice). Statistical test = two-sided Mann-Whitney U-test with Benjamini-Hochberg adjustment. **i** same as (**h**) for ECM-secreting myofibroblasts (n = 15 mice). Statistical test = two-sided Mann-Whitney U-test with Benjamini-Hochberg adjustment. **j** estimated cell proportion of inflammatory fibroblasts, pericytes and ECM-secreting myoFibroblasts after label transfer on a single cell RNAseq dataset from mice after UUO from[42]. IHC showing staining of SFRP1 (**k**), FAP, SFRP4 and RAMP1 (**l**) in PKD patients. Left: Representative serial IHC; Right: Corresponding quantification of H-scores in interstitial cells and the average percentage of fibrosis per quantified fields. N = 12 PKD patients. Scale bar = 50 µm (upper panel) and 100 µm (lower panel). Statistical test = two-sided Spearman correlation test. IHC showing staining of SFRP1 (**m**), FAP, SFRP4 and RAMP1 (**n**) in fibrotic kidney biopsies. Top: Representative images of IHC staining; Middle: Representative images of Masson's trichrome staining of biopsies shown in IHC; Bottom: Corresponding quantifications of H-scores and percentage of fibrosis. N = 13. Scale bar = 50 µm (upper panel), 100 µm (lower panel). Statistical test = two-sided Spearman correlation test. For (**d**), (**h**) and (**i**): boxplot represents the median (centre), first (Q1) and 3rd (Q3) quartiles (bounds of the box), Q1 + 1.5 × Interquartile range (IQR) and Q3-1.5 × IQR (whiskers). For (**k−n**) the error bar represents the 95% confidence level interval for predictions from a linear model, except for (**m**), where the model is "Loess".

clusters in tissue sections (Fig. 2k, l and Fig. S2a, b). Co-staining experiments between SFRP1 or FAP and αSMA showed that SFRP1+ and FAP+ cells were mainly αSMA+, consistent with them being activated fibroblasts (Fig. S2c, d). We also observed that in CKD but not in normal kidneys some tubular cells could also be positive for SFRP1 (Fig. 2k, l and Fig. S2e). Consistent with this, it is known that, during CKD, some tubular cells can express some mesenchymal markers (such as VIM or αSMA) and undergo partial epithelial to mesenchymal transition[43,44]. Strikingly, CXCL-iFibro (cluster 0) was more abundant in zones with persistent epithelial structures, while ECM-secreting myoFibro (Wound-myoFibro, cluster 3 and TGFβ-myoFibro, cluster 5) were mainly observed in fibrosis-enriched zones (Fig. 2k, l). Quantification of the histological scores (H-scores) of the different fibroblast markers, confirmed that the quantity of CXCL-iFibro (SFRP1+, cluster 0) was anti-correlated with the percentage of fibrosis (evaluated by a nephrologist and a pathologist at diagnosis) within PKD sections (Fig. 2k). In contrast, we found a linear positive correlation between the percentage of fibrosis and markers of ECM-secreting myoFibro (such as FAP, as well as SFRP4 and RAMP1), and specific markers of Wound-myoFibro (cluster 3) and TGFβ-myoFibro (cluster 5), respectively in PKD patients (Fig. 2l). Because PKD explants correspond to end-stage disease, we next measured the content of these different clusters at earlier stages in kidneys with a lower percentage of fibrosis (Fig. 2m, n). In that aim, we performed IHC on kidney biopsies from patients suffering from vascular nephropathy with interstitial fibrosis ranging from 0 to 80% of the parenchyma. The CXCL-iFibro and the two ECM-secreting myoFibro clusters, including the Wound-myoFibro and TGFβ-myoFibro clusters, were detected in kidney biopsies at these early stages of chronic kidney disease (Fig. 2m, n). Interestingly, H-score quantification integrating the proportion of positive cells within the interstitium revealed that CXCL-iFibro expanded in the interstitium at initial stages of fibrosis, before decreasing when the fibrosis percentage increased (Fig. 2m). In contrast, histological scoring of the different ECM-secreting myoFibro markers were linearly correlated with the percentage of fibrosis (Fig. 2n), indicating a constant increase in their content when fibrosis gradually developed in patients. The decrease in the content of the CXCL-iFibro cluster concomitantly to the linear increase in ECM-secreting myoFibro clusters when fibrosis was above 50% was consistent with the in-silico analysis using pseudotime and trajectory inference (shown in Fig. 2b, c). Altogether, these data validate the existence of CXCL-iFibro in kidney fibrosis at early stages and suggest that these inflammatory fibroblasts might be an early state of differentiation towards ECM-secreting myofibroblasts both in human CKD patients and a representative mouse model.

### The content in CXCL-iFibro correlates with FOLR2+ macrophage infiltration in kidney disease

Based on the accumulation of the CXCL-iFibro cluster at early phase of kidney fibrosis, we next aimed to investigate its functional role in CKD.

CXCL-iFibro expressed high levels of several cytokines and chemokines, suggesting potential interactions with immune cells. Interestingly, we observed that the CXCL-iFibro accumulated in COL1A1-negative zones (Fig. S2f), consistent with the lack of *COL1A1* expression in scRNAseq data from CXCL-iFibro (as shown in Fig. 1e). In contrast, the majority of ECM-secreting myoFibro (SFRP1- FAP+) were mainly detected in COL1A1-positive zones (Fig. S2f), as expected based on the high expression of the *COL1A1* gene in these clusters (Fig. 1e).

As macrophages are well-known to be key players in the development of fibrosis[45–49], we assessed the spatial distribution of CXCL-iFibro and macrophages in PKD explants and fibrotic kidney biopsies by performing co-staining of both CXCL-iFibro (SFRP1 + FAP-) and macrophages (CD68) (Fig. 3a, b and Fig. S3a, b for corresponding low magnification). Thus, by performing IF staining using specific markers, we confirmed the identity of these cellular clusters in PKD explants and fibrotic kidney biopsies. Interestingly, we observed a proximity and a strong positive correlation between the number of CXCL-iFibro (SFRP1+ FAP-) and the number of macrophages (CD68+) per mm² in both PKD explants (Fig. 3a and Fig. S3a) and fibrotic kidney biopsies (Fig. 3b and Fig. S3b). Interestingly, within each patient tissue section, each field that showed macrophage infiltration also exhibited CXCL-iFibro accumulation in both PKD and fibrotic kidney biopsies (Fig. 3a, b, quantifications in bottom panels). Conversely, each field that did not show CXCL-iFibro infiltration exhibited poor macrophage infiltration, confirming that the colocalization between SFRP1+ and CD68+ cells was not patient-dependent (Fig. 3a, b). Importantly, this correlation was not observed between the content in ECM-secreting myoFibro (FAP+) and macrophage infiltration in PKD (Fig. 3c and Fig. S3c). Indeed, while ECM-secreting myoFibro accumulated in the highly fibrotic zones in PKD explants, very few CD68+ cells were observed in these zones and mainly in FAP-negative zones (Fig. 3c and Fig. S3c). The co-localization between pro-inflammatory CXCL-iFibro and CD68+ macrophages suggests that CXCL-iFibro could be instrumental in attracting and promoting CD68+ macrophage accumulation at early stages of fibrosis development.

We next sought to identify more precisely the CD68+ myeloid cells in kidney fibrosis. Recent efforts have been made, especially in cancer, to better characterize the heterogeneity of myeloid cells and particularly monocytes and macrophages. Several types of tumor-associated macrophages (TAM), such as FOLR2+ (Folate receptor beta) and TREM2+ (Triggering receptor expressed on myeloid cells 2) TAM have been identified[50–52]. Interestingly, these subtypes of macrophages drive either pro-inflammatory response (FOLR2+) or immunosuppression (TREM2+)[50–52]. As these macrophage subtypes have not yet been identified in kidney diseases, we first analyzed the different myeloid cell clusters detected in publicly available scRNAseq data from CKD and controls[8]. After quality control and doublet elimination, 3960 myeloid cells were conserved for further analyses. Unsupervised graph-based clustering identified 6 clusters (Fig. 3d). These myeloid

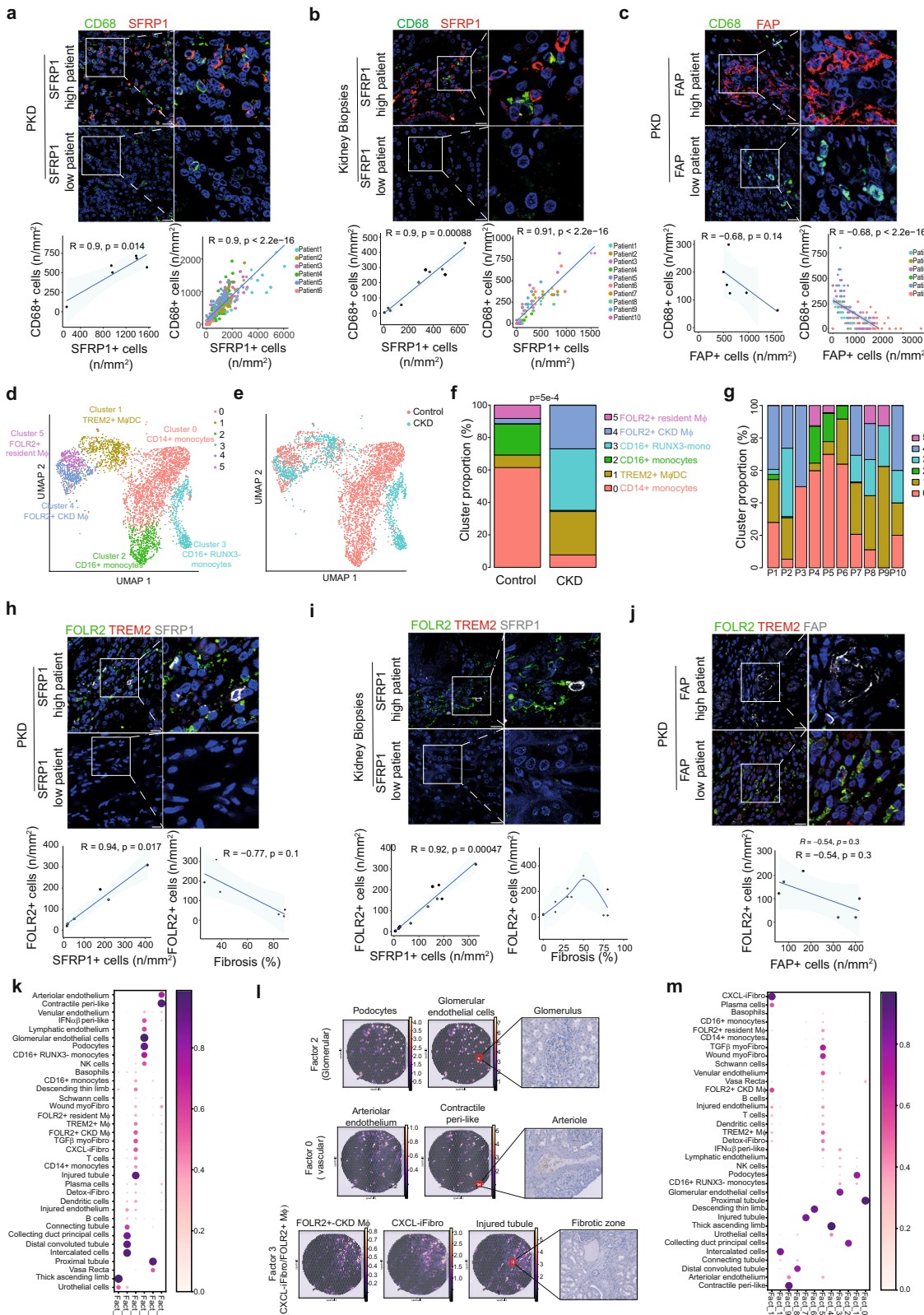

clusters were differently distributed in control and CKD patients, with a significant decrease in the content of clusters 0, 2 and 5 and accumulation of clusters 1, 3, and 4 in CKD patients compared to controls (Fig. 3e, f). These clusters were represented, albeit in different proportions, in several patients (Fig. 3g). Among these myeloid clusters, 3 clusters (0, 2 and 3) were identified as monocytes based on *VCAN* (Versican) expression, while three others (clusters 1, 4 and 5)

corresponded to macrophages based on *C1QA* (Complement C1q A chain) and *MRC1* (Mannose receptor C type 1) expression (Fig. S3d). As previously described[50], we identified two types of monocytes: one expressing CD14 that we annotated as CD14+ monocytes (cluster 0), and the other expressing CD16, hence referred to as CD16+ monocytes (clusters 2 and 3) (Fig. S3e). More precisely, Cluster 2 expressed *RUNX3* whereas cluster 3 did not (Fig. S3e). Interestingly, in the macrophage

**Fig. 3 | Inflammatory fibroblasts are in close vicinity of FOLR2+ macrophages.**
**a** Representative images (top) and corresponding quantifications (bottom) of IF co-staining of CD68 and SFRP1 (inflammatory fibroblast marker) in PKD patients (N = 6). Bottom left panel represents the average number of positive cells per patient. Bottom right panel represents the correlation between the number of positive cells per surface unit, each dot representing a field of 18 600μm². Scale bar = 20 μm. N = 6 PKD patients. Statistical test = two-sided Spearman correlation test. **b** Same as in (**a**) in fibrotic kidney biopsies. N = 10 patients with fibrotic kidney. Statistical test = two-sided Spearman correlation test. **c** Same as (**a**) but between CD68 and FAP (ECM-secreting myofibroblasts). Scale bar = 20 μm. N = 6 PKD patients. Statistical test = two-sided Spearman correlation test. **d** UMAP of scRNA-seq data from 3960 myeloid cells from[8]. Macrophages (Mφ), Dentritic cells (DC). **e** same UMAP as in (**d**) showing cell repartition in Control (red, n = 6) or CKD (blue, n = 6) patients. **f** Percentages myeloid cells clusters according to Control or CKD.

Two-sided Fisher Exact test. **g** Proportion of myeloid cell cluster in each patient (n = 10). **h** Representative images (top) and corresponding quantifications (bottom) showing IF co-staining of FOLR2, TREM2 and SFRP1 in PKD patients (N = 6). Scale bar = 20 μm. (N = 6 PKD patients). Statistical test = two-sided Spearman correlation test. **i** Same as in (**h**) in fibrotic kidney biopsies (N = 11 patients with fibrotic kidney). Statistical test = two-sided Spearman correlation test. **j** Same as in (**h**) for FOLR2+, TREM2+ cells, and FAP+ cells in PKD patients (N = 6). Scale bar = 20 μm (**k**) Non-negative matrix factorization of the deconvolution output with 6 factors high-lighting different microenvironment. Color and size of the dots represent the proportion of cells of each cell type. **l** Results of the deconvolution for different compartment using Cell2Location on Patient 1, described in Table S5. The number of predicted cells is plotted on the tissue. **m** Same as the (**k**) but with 11 factors. For (**a**–**c**) and (**h**–**j**) the error bar represents the 95% confidence level interval for predictions from a linear model.

populations, we could identify specific clusters expressing some of the recently published TAM markers including. TREM2 (left part of cluster 1) and FOLR2 (clusters 4 and 5) (Fig. S3f). It has been described that kidney resident macrophages express both CD74 and CD81[53]. Interestingly, this population corresponded mainly to cluster 5, which was also positive for FOLR2 (Fig. S3f). We thus defined FOLR2+ CD81+ macrophages (cluster 5), which decreased in CKD, as FOLR2+ resident macrophages and FOLR2+ CD81- macrophages (cluster 4), which accumulated in CKD, as FOLR2+ CKD macrophages (Fig. S3g and Fig. 3d–f). Differential gene expression and functional enrichment analysis confirmed that these 2 FOLR2+ populations were also characterized by a specific transcriptional profile. FOLR2+ resident macrophages showed activation of RHO-GTPase pathway, regulation of protein catabolic process, cytokine production and regulation of proteolysis, potentially corresponding to a scavenging-macrophage phenotype, while FOLR2+ CKD macrophages showed an enrichment in the cellular response to stress, inflammatory response and chemotaxis, corresponding to a more inflammatory phenotype (Table S4). Finally, cluster 1 was composed of TREM2+ macrophages, and dendritic cells (DC), which were positive for MRC1, CD1E and CD1C expression (Fig. S3f). We thus annotated cluster 1 as TREM2+ macrophages / DC (Fig. 3d, f and Fig. S3g).

As these different subtypes of FOLR2+ and TREM2+ macrophages have been poorly described but very recently detected in diabetic kidney disease[54], we tested whether we could detect them in PKD explants and fibrotic kidney biopsies, and we evaluated if they were localized in close vicinity to inflammatory fibroblasts (Fig. 3h, i and Fig. S3h, i for corresponding low magnification). Co-staining of SFRP1, FOLR2 and TREM2 markers by IF in these tissue samples showed a clear increase in the proportion of FOLR2+ macrophages in CKD upon fibrosis development (Fig. 3h, i). Strikingly, we observed a strong correlation between the number of SFRP1 + CXCL-iFibro and FOLR2+ macrophages per mm² (Fig. 3h, i), correlation not detected with FAP + ECM-secreting myoFibro (Fig. 3j and Fig. S3j for corresponding low magnification). On the other hand, TREM2+ macrophages were identified in kidney samples, but in different fields than CXCL-iFibro and FOLR2+ macrophages (Fig. S3k).

To further validate the spatial distribution of the different fibroblast and myeloid populations we identified in CKD, we performed spatial transcriptomics on 2 patients suffering from kidney fibrosis (Table S5 for clinical characteristics of these patients). To analyze these data, we used the comprehensive cellular atlas described in Fig. 2e, based on scRNAseq data sets from CKD and normal kidney tissues[8]. We finely annotated cell populations, resulting in 35 different cell types and states (Fig. S4a, see also *Methods*). We then mapped the localization and the abundance of each cell population by performing deconvolution using the Cell2location algorithm[55] with our single cell-based cellular atlas as reference. We applied non-negative matrix factorization analysis to underlie structures and patterns of co-localizing cell types and states. At low-resolution, we first confirmed some

expected colocalization, such as between glomerular capillaries and podocytes (glomerular compartment), arteriolar endothelium and pericytes, or proximal tubular cells and vasa recta (Fig. 3k, l). We next observed colocalization between injured tubules, fibroblasts including CXCL-iFibro and immune cells (Fig. 3k, l). This was of interest, as it has recently been suggested that injured tubules exhibit a pro-inflammatory transcriptomic profile and could trigger immune cell infiltration during CKD progression in mice[56–58]. To go further in our analysis of colocalization between CXCL-iFibro and immune cells, we increased the number of factors in our colocalization analysis, allowing a more resolutive discrimination between the different cell types (Fig. 3m). Strikingly, we observed that FOLR2+ −CKD macrophages mainly colocalized with CXCL-iFibro and plasma cells (Fig. 3l, m and Fig. S4b, c). On the other hand, other fibroblast subsets (IFNαβ-peri like, Detox-iFibro, Wound-myofibro and TGFβ-myofibro colocalize together with monocytes, FOLR2 resident macrophages, B cells, T cells, dendritic cells and TREM2+ macrophages (Fig. 3l, m). Altogether, these data show that CXCL-iFibro and FOLR2+ macrophages are co-localized during CKD, suggesting that a reciprocal crosstalk could exist between these two populations.

## CXCL-iFibro attract CD14+ monocytes and induce their differentiation into FOLR2+ macrophages

To decipher the molecular crosstalk between CXCL-iFibro and macrophages, we performed functional assays. To do so, we first generated in vitro cellular models recapitulating the main characteristics of both CXCL-iFibro and ECM-secreting myoFibro clusters. Based on our expertize on iCAF and myCAF isolation from CAF-S1 in breast cancer[29], we found that kidney-derived primary fibroblasts acquired distinct phenotypes according to the coating conditions used to expand them in culture (Fig. 4a, b). Indeed, when fibroblasts were expanded on collagen-coated plates, they exhibited higher SFRP1 and lower SFRP4, FAP and αSMA protein levels than cells cultured on plastic-dishes (Fig. 4a, b), showing that collagen-cultured primary fibroblasts were reminiscent of CXCL-iFibro and plastic-cultured fibroblasts of ECM-secreting myoFibro, as previously observed for CAF-S1[29]. In addition, as shown for iCAF and myCAF from pancreatic and breast cancer[26,29], we observed that the increase in αSMA protein level was a marker differentiating CXCL-iFibro from ECM-secreting myoFibro in vitro (Fig. 4a, b). Of note, neither collagen- nor plastic-cultured cells expressed E-Cadherin / CDH1, a marker of epithelial cells (Fig. 4a), thereby confirming that the primary cells established from human kidney are fibroblasts.

We then performed RNA sequencing on collagen- or plastic-cultured fibroblasts to confirm the respective identity of these cells. Consistent with the inflammatory identity detected in CXCL-iFibro, we identified several inflammatory genes, including *CXCL12, IL1B* and *IL34*, upregulated in collagen-cultured compared to plastic-cultured cells (Fig. S5a, b). These proteins are of interest, because of their role in chemoattraction of immune cells (CXCL12)[59], pro-inflammatory

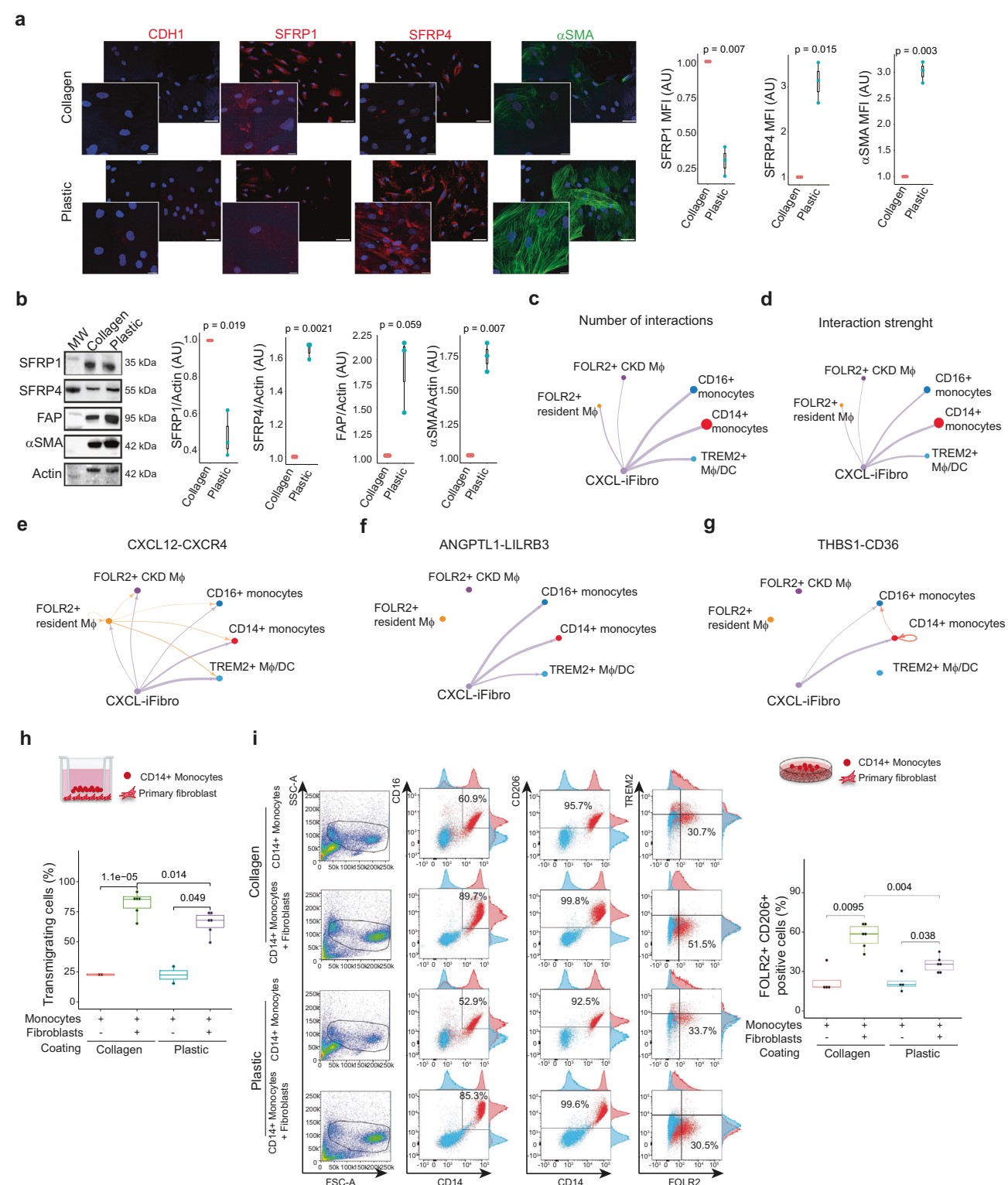

response (IL1B)[60,61] or in promoting the differentiation and viability of monocytes and macrophages through the colony-stimulating factor-1 receptor (IL34)[62]. Similarly, consistent with their ECM-secreting myo-Fibro identity, plastic- cultured fibroblasts exhibited increased expression of ECM-related genes, such as *COL4A1* and *COL4A5* (Fig. S5d), which are known to be important in the fibrotic process and basement membrane integrity[63]. We also identified other components of the ECM, belonging to the integrin (*IGFBP1*) or the laminin (*LAMA3*) families (Fig. S5d). Altogether, these data suggest that collagen-

cultured and plastic-cultured fibroblasts are reminiscent of CXCL-iFibro and ECM-secreting myoFibro, respectively.

To investigate interactions between CXCL-iFibro and monocytes/macrophages, we performed an *in-silico* ligand-receptor interaction study from scRNAseq data using the CellChat algorithm, which quantitatively infers intercellular communications[64]. By focusing our analysis on CXCL-iFibro and myeloid cells, we observed that the highest number and strength of interactions of CXCL-iFibro were with CD14+ monocytes (Fig. 4c, d). Interestingly, this in silico analysis suggested

**Fig. 4 | CXCL-iFibro attract CD14+ monocytes and induces a switch into FOLR2+ macrophages. a** Representative images (left panel) and quantifications (right panel) showing the expression of E-Cadherin/CDH1, SFRP1, SFRP4 and αSMA in primary fibroblasts cultured on collagen- (top) or plastic-dishes (bottom). Images at the bottom left corner are higher magnifications of other images from the same experiment. Quantification represents the average MFI of at least 100 cells per condition per independent experiment. Data are expressed as fold change to the paired collagen condition. n = 3 independent experiments. Scale bars = 20 μm. Statistical test = two-sided Mann-Whitney U-test. **b** Representative western blots (left) and corresponding quantifications (right) showing the expression of SFRP1, SFRP4, FAP and αSMA in primary fibroblasts cultured on collagen- or plastic-dishes. *P*-values from Mann-Whitney test (n = 3 independent experiments). Statistical test = two-sided Mann-Whitney U-test. **c–g** Cellchat analysis of the ligand-receptor interaction between CXCL-iFibro and myeloid cells. Chordplots of the number of significant interactions (**c**) and the strength of interactions (**d**) between CXCL-iFibro and myeloid cells. Chordplots showing the CXCL12-CXCR4- (**e**), the ANGPTL1-

LILR3- (**f**) and the THBS1-CD36- (**g**) ligand-receptor interaction between CXCL-iFibro and myeloid cells. **h** Quantification of the percentage of CD14+ monocytes transmigrating through a transwell, in presence or not of collagen- or plastic-cultured fibroblasts. *P*-values from Kruskall-Wallis tests (n = 3 independent experiments with three different cells lines and 2 PBMC from healthy donors). **i** Representative plots and corresponding quantification of flow cytometry analysis aiming at characterizing macrophage phenotype after 24 h of co-culture of CD14+ monocytes with either collagen- or plastic-cultured fibroblasts. From left to right columns are represented FSC-A/SSC-A, CD14/CD16, CD14-CD206 and FOLR2/TREM2 expression. Quantifications on the right show the percentage of FOLR2+ CD206+ macrophages among alive CD14+ monocytes. *P*-values from Kruskall-Wallis test (n = 4 independent experiments with three different cell lines and 4 PBMC from healthy donors). For (**a**, **b**) and (**h**, **i**): boxplot represents the median (centre), first (Q1) and 3rd (Q3) quartiles (bounds of the box), Q1 + 1.5 × Interquartile range (IQR) and Q3-1.5xIQR (whiskers). Abbreviations: MW molecular weight.

that 3 couples of ligand-receptor might drive this interaction, respectively CXCL12-CXCR4, ANGPTL1-LILRB3 and THBS-CD36 (Fig. 4e–g). Because we observed a close vicinity between CXCL-iFibro and FOLR2+ macrophages (Fig. 3h, i), we hypothesized that CXCL-iFibro could attract CD14+ monocytes and modify their phenotype into a FOLR2+ macrophage phenotype, hypothesis consistent with the preferential interaction detected *in-silico* between CXCL-iFibro and CD14+ monocytes. To test this hypothesis, we designed co-culture experiments between collagen-cultured primary fibroblasts (CXCL-iFibro) and CD14+ monocytes isolated from the peripheral blood mononuclear cells (PBMC) of healthy donors, to assess if CXCL-iFibro could indeed attract CD14+ monocytes and potentially promote their differentiation into a FOLR2+ macrophage phenotype. We first checked whether CXCL-iFibro could attract CD14+ monocytes by performing transwell migration assays. We observed that CXCL-iFibro stimulated the migration of CD14+ monocytes more efficiently than ECM-secreting myoFibro (Fig. 4h). Moreover, co-culture experiments of CXCL-iFibro or ECM-myoFibro with CD14+ monocytes showed that CXCL-iFibro significantly increased the proportion of FOLR2 + CD206+ (MRC1) macrophages from CD14+ monocytes (Fig. 4i). ECM-secreting myoFibro also induced a switch towards FOLR2+ macrophages, but to a much lesser extent than CXCL-iFibro (Fig. 4i). Thus, taken together, our data highlight that CXCL-iFibro attract monocytes and promote their differentiation into FOLR2+ macrophages.

**Macrophages induce the transition from CXCL-iFibro to ECM-secreting myoFibro through the WNT/β-catenin pathway**
Based on the crosstalk between CXCL-iFibro and FOLR2+ macrophages, we next tested if, in turn, macrophages could modify the phenotype of kidney-derived primary fibroblasts. We observed that upon co-culture with CD14+ cells, CXCL-iFibro experienced an increase of both αSMA and SFRP4 proteins to a level close to the one of ECM-secreting myoFibro (Fig. 5a), suggesting that myeloid cells could induce the switch of CXCL-iFibro into ECM-secreting myoFibro. To decipher the molecular pathways involved in the transition from CXCL-iFibro towards ECM-secreting myoFibro, we performed an *in-silico* analysis, using transcription factor inference models on the trajectory inference analysis shown in Fig. 2a, particularly looking at the node between CXCL-iFibro and ECM-secreting myoFibro (as shown Fig. S6a, Table S6). Using the Dorothea algorithm[65], we observed that TCF4 and TCF12, two key mediators of the WNT/β-catenin pathway, were in the top-10 transcription factors involved in the transition from CXCL-iFibro to ECM-secreting myoFibro (Fig. 5b and Table S7). We also validated this finding by applying another *in-silico* approach by using Monocle 3. Monocle 3 classifies differentially expressed genes into modules of genes, which are co-regulated along the trajectory[66]. We identified 7 gene modules, which were expressed at different points of the trajectory and specifically characterized these distinct states

(Fig. S6b). Interestingly, we identified that module 4 highlighted the transition from CXCL-iFibro to ECM-secreting myoFibro (Fig. 5c, d). Indeed, we observed that the module 4 signature highlighted both CXCL-iFibro and Wound-myoFibro on the UMAP, whereas module 3 was exclusively highlighted in CXCL-iFibro (Fig. 5c). Transcriptomic profiles of the gene-modules 3 and 4 through the Metascape platform revealed that module 3 exhibited an inflammatory signature, while module 4 showed an ECM-related signature (Table S8 and Table S9), in agreement with the transition from CXCL-iFibro to ECM-secreting myoFibro. By performing functional enrichment using the TRRUST database[67] on module 4 genes, we identified TCF4 as one of the main transcription factors involved in the switch from CXCL-iFibro to ECM-secreting myoFibro (Fig. 5e). Strikingly, the TCF4 transcription factor was thus identified by the 2 approaches (Fig. 5b, e), which highlighted its relevance. Moreover, *TCF4* expression was more prominent in ECM-secreting myoFibro (Fig. S6c). *TCF4* expression along the pseudotime defined in Fig. 2b showed that its expression increased late in the differentiation, in ECM-secreting myoFibroblasts (Fig. S6d). Finally, expression of TCF4-target genes was also increased in Wound-myoFibro, strengthening the idea that TCF4 could be instrumental in the transition from CXCL-iFibro to ECM-secreting myoFibro (Fig. S6e). Interestingly, macrophages have been shown to participate in kidney regeneration by activating the WNT/β-catenin pathway in epithelial cells[68,69]. We next aimed to validate the role of the WNT/β-catenin pathway in macrophage-induced switch of CXCL-iFibro into ECM-secreting myoFibro by functional assays. First, we treated CXCL-iFibro with two different specific agonists of the WNT/β-catenin pathway (CAS 853220-52-7; SKL2001) and observed that the activation of this pathway was concomitant with the increase of αSMA and SFRP4 protein levels (Fig. 5f), suggesting that its activation could drive the phenotype from CXL-iFibro to ECM-secreting myoFibro. Second, we observed that cultured ECM-secreting myoFibro showed more nuclear β-catenin staining than CXCL-iFibro (Fig. 5g), consistent with enhanced activation of the WNT-pathway in ECM-secreting myoFibro compared to CXCL-iFibro. We next tested if the macrophage-induced switch from CXCL-iFibro to ECM-secreting myoFibro could be driven by the WNT/βcatenin pathway. To do so, we co-cultured CD14+ myeloid cells with CXCL-iFibro either treated or not with an inhibitor of the β-catenin/TCF interaction (iCRT3). We observed that addition of the β-catenin/TCF interaction inhibitor did not affect the proportion of FOLR2+ macrophages upon co-culture (Fig. S7), indicating that the treatment had no impact on the differentiation of CD14+ monocytes into FOLR2+ macrophages. Without treatment, we confirmed by both IF and western blots that co-culture of CXCL-iFibro with CD14+ monocytes promoted the nuclear translocation of β-catenin in fibroblasts (Fig. 5h, i). Moreover, this was concomitant to the up-regulation of ECM-secreting myoFibro markers (Fig. 5j, k). Interestingly, we observed that inhibition of β-catenin/TCF interaction prevented the macrophage-induced

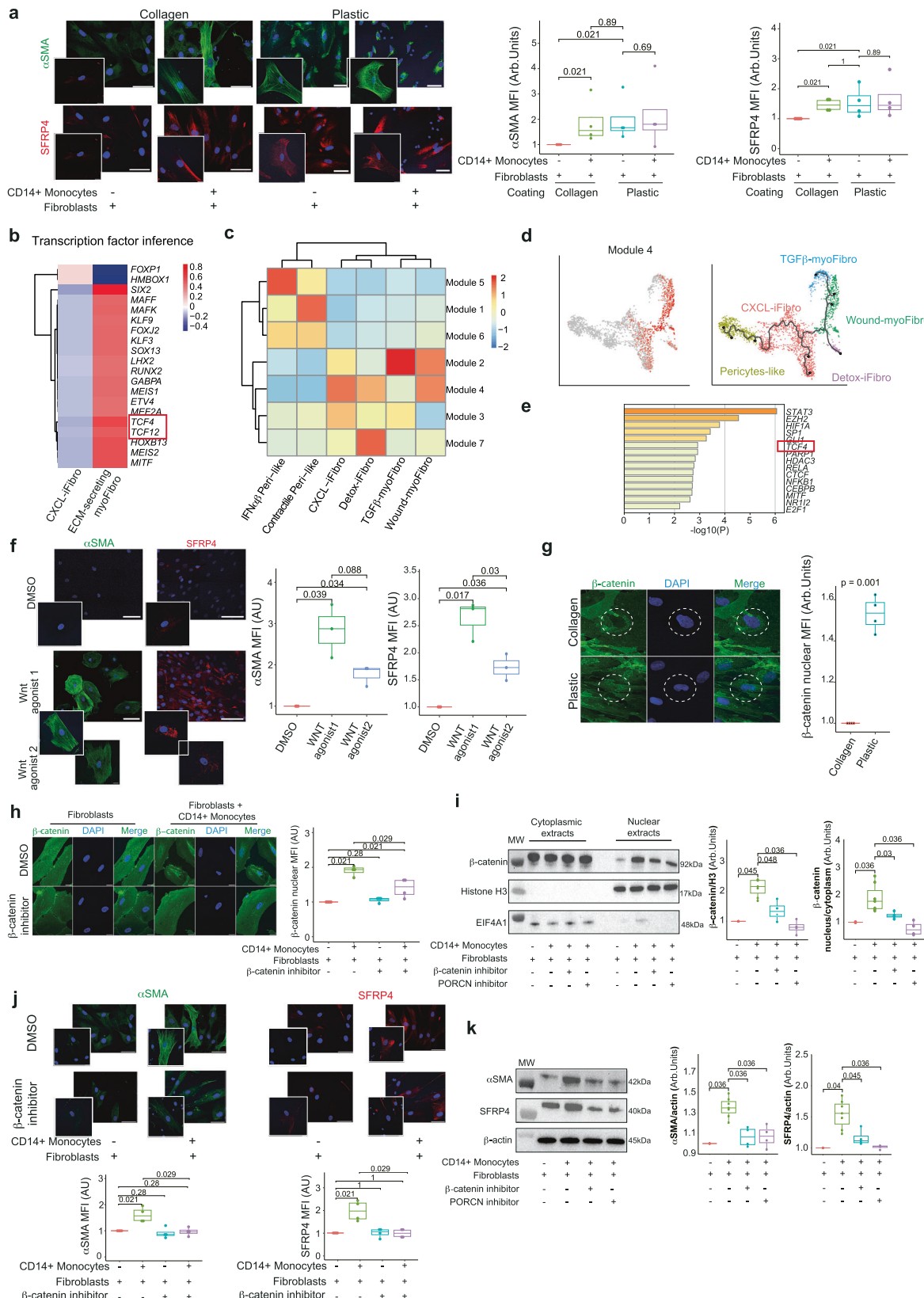

switch from CXCL-iFibro into ECM-secreting myoFibro (Fig. 5j, k). Indeed, following inhibition, CXCL-iFibro showed a significant reduction of β-catenin nuclear translocation in fibroblasts and did not experience any increase in either αSMA or SFRP4 staining when cocultured with CD14+ monocytes (Fig. 5i, k), indicating that the WNT/β-catenin pathway was required for the differentiation of CXCL-iFibro into ECM-secreting myoFibro. Finally, because macrophages have been shown to secrete WNT ligands to promote tissue repair in the intestine or kidney, we confirmed that blocking WNT ligand secretion by C-59 (a PORCN inhibitor) reduced β-catenin nuclear translocation and prevented the fibroblast phenotypic switch from CXCL-iFibro into ECM-secreting myoFibro (Fig. 5i, k). Altogether, these data show that

**Fig. 5 | Macrophages induce a switch from CXCL-iFibro to ECM-secreting myoFibro through a WNT/β-catenin dependent pathway. a** IF image and quantification showing DAPI (blue), αSMA (green), and SFRP4 (red) staining in fibroblasts plated on collagen- or plastic- dishes ± co-culture with CD14+ monocytes. Scale bar = 50 μm (main image) or 20 μm (higher magnification). Adjusted *p*-values (BH) from two-sided Mann-Whitney test (n = 3 independent experiments). **b** Heatmap showing the result of the transcription factor inference using Dorothea algorithm between CXCL-iFibro and ECM-secreting myofibro. **c** Heatmap showing the expression of gene modules identified by Monocle 3 according to fibroblast cluster. **d** UMAP showing the average z-score of module 4 gene expression (left) in the UMAP obtained by Monocle 3 (right, same as Fig. 2a). **e** TRRUST analysis of genes specifically upregulated in module 4, using Metascape.org. **f** same as (**a**) in collagen-cultured fibroblasts ± WNT agonists. Scale bar = 100 μm (main image) or 20 μm (higher magnification). Adjusted *p*-values (BH) from two-sided Mann-Whitney test (n = 3 independent experiments). **g** IF image and quantification showing DAPI (blue) and β-catenin protein (green) in fibroblasts in collagen- (top) or plastic- (bottom) dishes. Scale bar = 20 μm. *P*-value from two-sided Mann-Whitney test. (n = 4 independent experiments). **h** Same as in (**g**) for collagen-cultured fibroblasts ± co-culture with CD14+ monocytes, ±inhibitor of β-catenin/TCF interaction (iCRT3). Scale bar = 20 μm. Adjusted *p*-values (BH) from two-sided Mann-Whitney test. (n = 4 independent experiments). **i** Western blots and quantifications showing β-catenin, Histone H3, and EIF4A1 protein following cytoplasmic and nuclear fractionation in collagen-cultured fibroblasts ± co-culture with CD14+ monocytes, ±β-catenin/TCF interaction (iCRT3) and PORCN (C59) inhibitors. Adjusted *p*-values (BH) from two-sided Mann-Whitney test. (n = 4 independent experiments). **j** Same as in (**h**) for DAPI (blue), αSMA (green) and SFRP4 (red) Scale bar = 50 μm (main image) or 20 μm (higher magnification). Adjusted *p*-values (BH) from two-sided Mann-Whitney test. (n = 4 independent experiments). **k** Western blots and quantifications of αSMA and SFRP4 in fibroblasts ± co-culture with CD14+ monocytes, ±β-catenin/TCF interaction (iCRT3) and PORCN(C59) inhibitors. Adjusted *p*-values (BH) from two-sided Mann-Whitney test. (n = 4 independent experiments). For (**a**) and (**f–k**), boxplots are defined similarly than in Fig. 3.

macrophages stimulate the differentiation of the CXCL-iFibro into ECM-secreting myoFibro through activation of the WNT/β-catenin pathway.

## Accumulation of CXCL-iFibro at early stage of chronic kidney disease predicts progression and poor prognosis of CKD patients

As the role of CXCL-iFibro and FOLR2+ macrophages is poorly defined in CKD progression, we sought to assess if their presence could predict CKD progression. As CXCL-iFibro represents an intermediate state in the differentiation process into ECM-secreting myoFibro, we hypothesized that accumulation of CXCL-iFibro might be indicative of CKD progression at early phase of the disease. To address this question, we defined a CXCL-iFibro transcriptomic signature based on the differentially expressed genes in the different clusters identified from scRNAseq data (Fig. 1 and Table S10). We confirmed that this signature highlighted CXCL-iFibro in the scRNAseq-based cellular atlas we built (Fig. 6a, b), thereby validating that we could use this gene signature to specifically detect the CXCL-iFibro population in bulk RNAseq data. To evaluate the association of the CXCL-iFibro signature with patient longitudinal outcome, we took advantage of a prospective observational cohort from the Nephrotic Syndrome Study Network (NEPTUNE) with available transcriptomic profiles and longitudinal clinical data[70,71]. We selected adult patients with a mild to moderate reduction in kidney function (CKD stage 1 to 3a defined by an eGFR>45 mL/min/1.73m²) at the time of the clinically indicated renal biopsy (N = 134 patients) to determine if the CXCL-iFibro signature could predict patient outcome at an early stage of kidney disease. The patients' characteristics were summarized in Table 1. Interestingly, we observed that the expression rate of CXCL-iFibro-specific genes was associated with a composite outcome of end-stage renal disease (ESRD), or reduction of baseline estimated glomerular filtration rate (eGFR) of more than 40% (Fig. 6c and Fig. S8a). Optimal stratification of patients for survival analyses was performed using an iterative method. Patients were ranked by their expression of CXCL-iFibro transcriptomic signature and thresholds were defined for each level of expression (Fig. S8b, Table S11). For each threshold, patients were separated into low and high CXCL-iFibro expression scores and a Log-rank test was applied. We selected the threshold, which displayed the most significant p-value and separated 50 patients with low-score from 84 patients with high-score (Fig. S8b, Table S11). Indeed, patients with a high expression score of CXCL-iFibro showed significantly poorer longitudinal outcomes compared to patients with low CXCL-iFibro score (Fig. 6c). Moreover, univariate analysis showed that the eGFR at the time of biopsy, presence of arterial hypertension (HTN) at the time of the biopsy, the urinary protein to creatinine ratio (UPCR), and the CXCL-iFibro signature expression were each significantly associated with poor outcomes (Table 2). Strikingly, CXCL-iFibro expression

score was one of the most predictive variables, with a hazard ratio of 2.8. We then performed a multivariate analysis using a Cox regression model integrating eGFR, UPCR, HTN, age and CXCL-iFibro expression score as covariables. We observed that UPCR at the time of biopsy, presence of HTN and high CXCL-iFibro score were independently associated with ESRD or decrease of 40% of eGFR in this cohort of mild-to-moderate CKD patients (Fig. 6d). Finally, the eGFR slope, defined by the change of eGFR per year, was available for 128 patients. We classified patients as fast or slow progressors, based on the level of the eGFR slope (more or less than −5mL/min/1.73m², respectively). We observed that fast progressor patients exhibited a higher CXCL-iFibro expression score than slow progressor patients at the time of kidney biopsy (Fig. 6e). We also defined patients with low- or high-CXCL-iFibro expression score, defined by an expression score below or above the median of the expression score, respectively. We observed that the number of fast-progressor patients was higher in the subgroup with a high CXCL-iFibro score than in the low-CXCL-iFibro score subgroup (23 *vs* 6 respectively, *p* = 0.0006, Fig. 6f). Finally, the CXCL-iFibro expression score was anti-correlated with the eGFR slope (Fig. 6g). Based on data presented above, we hypothesized that CXCL-iFibro and FOLR2+ macrophages might be interdependent variables during CKD progression. We thus analyzed the *FOLR2* expression level in this same cohort (Fig. 6h–l). We found that *FOLR2* expression was associated with poor patient outcome, as observed with the CXCL-iFibro score. Indeed, patients with a high *FOLR2* expression level experienced poorer outcomes than those with low *FOLR2* expression (Fig. 6h, Fig. S8c, d, and Table S12). Multivariate analysis using a Cox regression model showed that UPCR at biopsy, presence of HTN and *FOLR2* expression were associated with poor outcomes in an independent manner (Fig. 6i). As for the CXCL-iFibro expression score, we observed that fast progressor patients exhibited a higher *FOLR2* expression than slow progressor patients (Fig. 6j) and that a higher proportion of fast progressor patients exhibited a high *FOLR2* expression score (defined by an expression above the median, Fig. 6k). Interestingly, Cox regression model integrating eGFR, UPCR, age, presence of HTN, CXCL-iFibro expression score and *FOLR2* expression as covariables, showed that only UPCR at biopsy and the presence of HTN were associated with outcome (Fig. S8e), indicating that *FOLR2* and CXCL-iFibro expression scores were not independent. Consistent with this observation, the expression of *FOLR2* and of CXCL-iFibro-specific genes was tightly correlated (Fig. 6l). This finding was in agreement with data showing co-staining of SFRP1 (marker of CXCL-iFibro) and FOLR2 protein in patient tissues (as show above Fig. 3g, h). Finally, we wanted to assess if the ability of CXCL-iFibro expression score or FOLR2+ macrophages expression to predict kidney outcome was independent of the expression of other monocytes/macrophages markers. To do so, we evaluated the role on CKD progression of CD14, a wide marker of myeloid cells (monocytes, macrophages, and

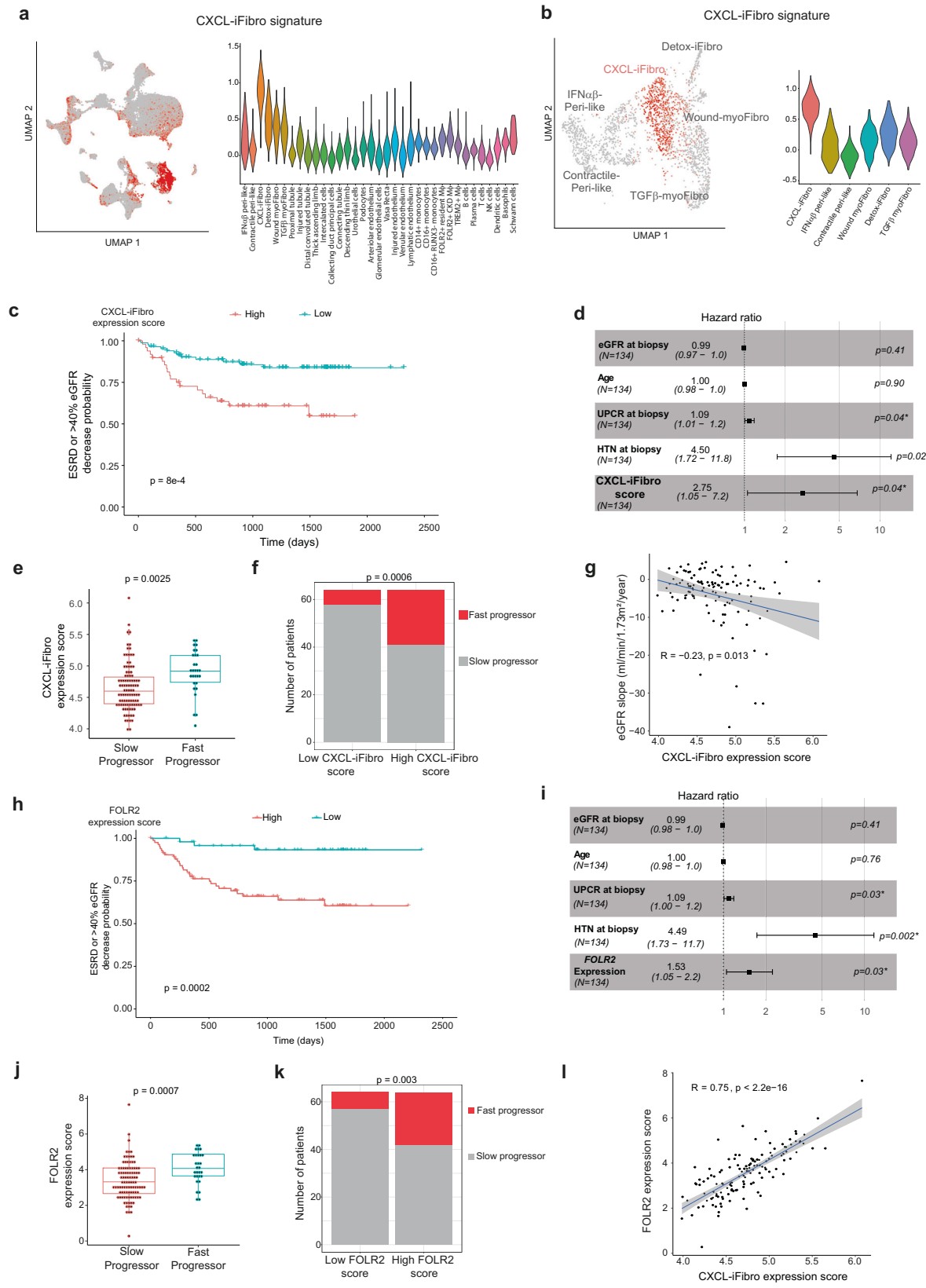

dendritic cells), and MRC1, a marker of M2-like macrophages, suspected to be involved in CKD progression[72,73]. Interestingly, neither CD14 nor MRC1 were associated with poor patient outcome in univariate analysis (*p* = 0.14 for CD14, *p* = 0.095 for MRC1). As for MRC1, the *p*-value was less than 0.1, we added it in the multivariate Cox model. Interestingly, the CXCL-iFibro expression score and FOLR2 expression

remained significantly associated with kidney outcomes (Fig. S8f, g). These data suggest that the early identification of CXCL-iFibro or FOLR2+ macrophages could refine the risk prediction of CKD progression, independently of the number of CD14+ cells or MRC1+ cells in the kidney. Altogether, these data show that CXCL-iFibro and FOLR2+ macrophages are interconnected biomarkers in CKD

**Fig. 6 | Expression of CXCL-iFibro gene signature predicts poor outcome of early CKD patients. a** same UMAP as in Fig. S3a and violin plots showing the average z-score expression of CXCL-iFibro gene signature. **b** same UMAP as in Fig. 1 and violin plots showing the average z-score expression of CXCL-iFibro gene signature. **c** Kaplan-Meier curve for the composite outcome (ESRD or loss of more than 40% of eGFR) according to the expression of CXCL-iFibro signature. N = 134 patients; events: 4 vs 27 in the low- and high- expression group respectively. *P*-value from Log-rank test. **d** Results of Cox multivariate analysis according to the following variables: eGFR at biopsy, age of the patient, UPCR, presence of hypertension and CXCL-iFibro expression score (N = 134 patients). Statistical model = multivariable Cox model. **e** Mean expression of CXCL-iFibro gene signature according to the progressor status of CKD patients (Slow progressor N = 99; Fast progressor N = 29). Statistical test = two-sided t-test. **f** Proportion of CKD patients with low (below median) or high (above median) CXCL-iFibro expression score according to patient progression status. *P*-value from two-sided Fisher Exact test. **g** Correlation between the CXCL-iFibro expression score and the eGFR slope.

*P*-value from two-sided Spearman correlation test. **h** same as (**c**) according to *FOLR2* expression. N = 134 patients; events: 3 vs 28 in the low- and high- expression group respectively. P-value from Log-rank test. **i** Results of Cox multivariate analysis according to the following variables: eGFR at biopsy, age of the patient, UPCR, presence of hypertension, and *FOLR2* expression (N = 134 patients). Statistical model = multivariable Cox model. **j** Same as in (**e**) but for *FOLR2* expression. Statistical test = two-sided t-test. **k** Same as in (**f**) but for *FOLR2* expression. Statistical test = two-sided exact Fisher test. **l** Correlation between *FOLR2* and the CXCL-iFibro expression score. P-value from two-sided Spearman correlation test (N = 128 patients). For (**d**), (**i**), (**g**), and (**l**) the error bar represents the 95% confidence interval. For (**e**) and (**j**): boxplot represents the median (centre), first (Q1) and 3rd (Q3) quartiles (bounds of the box), $Q1 + 1.5 \times$ Interquartile range (IQR) and $Q3 - 1.5 \times IQR$ (whiskers). Abbreviations: eGFR estimated glomerular filtration rate, HTN hypertension, UPCR urinary protein-to-creatinine ratio, ESRD end-stage renal disease.

progression and demonstrate that our findings could be relevant in clinical practice.

## Discussion

In this study, we took advantage of the detailed characterization of CAF populations in cancer to highlight fibroblast heterogeneity during kidney disease progression. By combining single cell RNAseq data, spatial transcriptomics, image analysis of patients' tissues and in vitro functional assays using primary fibroblasts isolated from CKD patients, we highlight a significant fibroblastic heterogeneity in fibrotic kidney and demonstrate the key function of specific fibroblast populations during kidney fibrosis development. Indeed, we identified two distinct clusters of an already-known population of ECM-secreting myofibroblasts, but we also revealed the function of a kidney fibroblast population characterized by an inflammatory phenotype, that we named CXCL-iFibro. Interestingly, inflammatory fibroblasts are well-known in cancer, but have poorly been described in CKD[10-19,22-24,33,74]. We validated the existence of the different aforementioned fibroblast populations in human tissue sections by analyzing patients at different stages of CKD. By this way, we show that the CXCL-iFibro population expands early during the disease course, before shrinking at late stages meanwhile the ECM-secreting myofibroblast population grows. This suggests that CXCL-iFibro could be an early intermediate state in the differentiation process giving rise to ECM-secreting myofibroblasts. Consistent with the accumulation of the CXCL-iFibro population at

early stages of kidney disease, we show its probable function in the progression of the pathology by highlighting its reciprocal crosstalk with FOLR2+ macrophages. Indeed, we show that the inflammatory signature of CXCL-iFibro is functionally relevant, as CXCL-iFibro are able to attract CD14+ monocytes and to polarize them into FOLR2+ macrophages, which have been recently described in cancer and diabetic kidney[50-52,54]. In turn, in vitro, FOLR2+ macrophages promote the differentiation of CXCL-iFibro into ECM-secreting myofibroblasts through a WNT/β-catenin dependent mechanism. Finally, we show that detecting CXCL-iFibro or FOLR2 at early stage of kidney disease in a large cohort of CKD patients is predictive of poor patient outcome. This confirms that the CXCL-iFibro population might be an early player in CKD progression and demonstrates the clinical relevance of our findings in CKD.

ECM secreting myofibroblasts are recognized as essential players in CKD progression by enhancing fibrosis[75]. Important efforts have been made to decipher the origin of these cells, and several studies conducted both in mice and humans concluded that myofibroblasts can be of multiple origins, including pericytes, fibroblasts or even rarely monocytes or epithelial cells[4]. More recently, by using scRNAseq of human kidney samples with CKD, a recent study confirmed the multiple origins of ECM-secreting myofibroblasts, but also identified a molecular heterogeneity in the stromal compartment[8]. Nevertheless, the functions of these fibroblast subtypes were not deciphered. Based on our expertize on CAF heterogeneity and their multiple functions in cancer[15,16,18,19,29], we hypothesized that the different fibroblast populations identified in CKD could exert different roles. In several cancer types, CAF have been classified as inflammatory (iCAF) or myofibroblastic (myCAF)[17,29,34,36,37,76]. Interestingly, by performing computational analysis of scRNAseq data of mesenchymal cells from CKD, and comparing them with different CAF populations, we identified a population of inflammatory fibroblasts in CKD, that we referred to as CXCL-iFibro. Inflammatory fibroblasts have been poorly described in fibrosis, especially in the kidney, but have been described in rheumatoid arthritis[74,77,78], where they play an important role in the pathogenesis of this disease and in long-term fibrosis. We show here that inflammatory fibroblasts are a transitional state in the differentiation process toward ECM-secreting myofibroblasts in CKD, which might explain why they have not been identified earlier in CKD. Indeed, CXCL-iFibro expand at early stages of CKD, before decreasing. Moreover, as observed for iCAF in cancer[26,29], CXCL-iFibro in CKD express low αSMA protein levels and do not secrete collagens, showing that they are not terminally differentiated myofibroblasts. Few studies have performed lineage tracing in mouse models of kidney fibrosis to define the origin of myofibroblasts and determined perivascular mesenchymal stem cell-like, as one population of origin[3-7]. In agreement with these findings, we also identified 2 populations of pericyte-like mesenchymal cells in human CKD and normal tissue. Interestingly,

## Table 1 | Clinical and biological data of the 134 patients with mild to moderate CKD

| | |
|---|---|
| Age (years) | 46.5 (30.5–59.0) |
| Sex (male, n = ) | 84 (62.6%) |
| Ethnicity | Caucasian/white: n = 89; Afro-American: n = 23; Asian/Asian American: n = 15; Multi-racial: n = 4; |
| Diagnosis | MN: n = 43; MCD n = 27; FSGS: n = 36; IgAN n = 17; other: n = 11 |
| HTN at biopsy (n = ) | 71 (53.0%) |
| eGFR at biopsy (mL/min/1.73 m²) | 84.0 (65.0–104.9) |
| UPCR at biopsy (g/g) | 3.8 (1.5–6.7) |
| Follow-up (months) | 47.0 (30.3–55) |
| ESRD or decrease of 40% eGFR (n = ) | 31 (23%) |
| eGFR slope (mL/min/1.73 m²) | −1.62 (−4.6–1.1) |
| Fast progressors (n = ) | 36 (26.9%) |

*MN* membranous nephropathy, *MCD* minimal change disease, *IgAN* IgA nephropathy, *HTN* hypertension, *eGFR* estimated glomerular filtration rate, *UPCR* urinary protein-to-creatinine ratio, *ESRD* end-stage renal disease.

**Table 2 | Results of the univariate analysis of the composite outcome (ESRD or decrease of 40% of eGFR)**

| | beta | HR (95% CI for HR) | wald.test | p.value |
|---|---|---|---|---|
| Age | 0.017 | 1 (1–1) | 2.4 | 0.12 |
| eGFR at biopsy | −0.017 | 0.98 (0.97–1) | 4.2 | 0.041 |
| HTN at biopsy | 1.4 | 4.2 (1.7–10) | 9.8 | 0.002 |
| Sex | −0.41 | 0.66 (0.31–1.4) | 1.1 | 0.3 |
| UPCR at biopsy | 0.065 | 1.1 (1–1.1) | 3.8 | 0.052 |
| Interstitial Fibrosis | 0.02 | 1 (0.99–1.1) | 1.7 | 0.19 |
| CXCL-iFibro expression score | 1.1 | 2.9 (1.3–6.3) | 6.7 | 0.009 |
| FOLR2 expression | 0.36 | 1.4 (1.1–1.9) | 6.4 | 0.011 |
| CD14 expression | 0.38 | 1.5 (0.89–2.4) | 2.2 | 0.14 |
| MRC1 expression | 0.34 | 1.4 (0.94–2.1) | 2.8 | 0.095 |

Statistical test = Cox univariate model.

*eGFR* estimated glomerular filtration rate, *HTN* hypertension, UPCR urinary protein-to-creatinine ratio.

by performing trajectory inference, we showed that CXCL-iFibro constitute an intermediate state between these pericyte-like clusters and ECM-secreting myoFibro. Moreover, by studying a dataset from a time-course experiment in a well-established relevant mouse model of kidney fibrosis[38], we validated the evolution of the proportions of the different cellular populations that we identified after injury and during fibrosis development in the human pathology. In addition, the transitional CXCL-iFibro population has been identified in human CKD by transcriptomic profiling at single cell level, thus reaching a high-resolution rate not yet achieved in lineage tracing.

Inflammatory fibroblasts have been defined based on their transcriptomic profiles, showing a high expression of interferon-related genes. This is also the case in cancer[17,26,29,36,37,76,79] and in chronic inflammatory diseases[80], where transcriptomic analysis first helped to identify this population. Moreover, several lines of evidence have shown the functional relevance of inflammatory fibroblasts in cancer and inflammatory diseases[13,17,26,28,29,78,80–82]. Here, we provide additional evidence on the functional role of CXCL-iFibro in CKD by showing attraction, close vicinity and reciprocal crosstalk between CXCL-iFibro and macrophages. Macrophages are well-known since a long time to participate in CKD progression[45,47,48,83]. Our study goes a step further in the characterization of these macrophages, thanks to recent advances on TAM[50–52,84]. Indeed, we identify several populations of macrophages in CKD, including FOLR2+ and TREM2+ macrophages, two TAM subsets recently discovered in cancer and still poorly described in kidney[50–52,54,84]. If the role of macrophages is well-established in renal fibrosis, the role of FOLR2+ macrophages is not known. Interestingly, we identified 2 populations of FOLR2+ macrophages in kidneys, one which seems to correspond to resident macrophages with a scavenging phenotype, and the other one with a pro-inflammatory phenotype, which is more represented in CKD patients. Their different localizations within kidney tissue as revealed by spatial transcriptomics strongly suggest that these 2 populations correspond to 2 different types of macrophages. Our in vitro studies suggest that FOLR2 + CKD macrophages originate from CD14+ monocytes, but further studies are needed to confirm this hypothesis. Here, we show that FOLR2+ macrophages interact with CXCL-iFibro, while the few TREM2+ macrophages detected in CKD are distant from CXCL-iFibro. Interestingly, we observe that the WNT/β-catenin pathway is a key player in the reciprocal crosstalk between CXCL-iFibro and FOLR2+ macrophages. Sustained activation of WNT/β-catenin pathway is key in cell-to-cell communications and has previously been associated with the development of renal fibrotic lesions[85–90]. Despite being

relatively silent in normal adult kidney, WNT/β-catenin signaling is re-activated in a large number of renal injury mouse models and in human kidney fibrosis[85,87,88,90,91]. Moreover, blocking Wnt secretion by genetic depletion of Wntless, a cargo receptor, in renal tubular epithelial cells in mice markedly reduces myofibroblast activation and kidney fibrosis[92]. These data underline the role of WNT/β-catenin pathway in kidney disease through epithelial-to-mesenchymal communication[85,87,88,93]. In addition, activation of the WNT/β-catenin pathway also stimulates macrophage polarization and contributes to kidney fibrosis[94]. Moreover, macrophage-secreted Wnt ligands, such as Wnt7b, stimulate epithelial responses in injured renal tissue and participate in kidney or intestine epithelial repair[68,69]. Here, we identify the role of the WNT pathway on CXCL-iFibro fibroblasts. Indeed, we show in vitro that FOLR2+ macrophages induce the activation of CXCL-iFibro into ECM-secreting myofibroblasts through activation of WNT/βcatenin pathway. Interestingly, inhibition of β-catenin/TCF interaction or blocking WNT ligand secretion prevents the activation of myofibroblasts, suggesting that this mechanism could be druggable to preclude kidney fibrosis and CKD progression. Whether inhibition of WNT ligand secretion by macrophages will be sufficient to prevent the switch of CXCL-iFibro into ECM-secreting myofibroblasts remains to be determined in patients. In line with this goal, we focused our analyses on human samples and primary-derived cells all throughout our study. Datasets combining kidney transcriptomic data with clinical and biological follow-up are quite scarce, but we had the opportunity to study the NEPTUNE cohort, which focuses on patients with glomerular diseases and nephrotic syndromes, including patients with CKD. Here, we demonstrate that the detection of CXCL-iFibro at early stages of CKD is clinically relevant. Indeed, very interestingly, we observed that the presence of CXCL-iFibro in patients with mild- to moderate-CKD is predictive of poor patient outcome. Whether this observation is similar in other diseases, such as diabetic or vascular nephropathy, needs to be validated in prospective studies. Similarly, identifying the right threshold of CXCL-iFibro expression that most efficiently discriminates between patients with low or high risk of CKD progression should be assessed in a specific follow-up prospective study. Despite these relative limitations, our study gives a strong argument in favor of the early role of CXCL-iFibro in fibrosis development.

In conclusion, by combining transcriptomic analysis with imaging and functional assays, our study unravels new mechanisms driving CKD progression. We herein identify CXCL-iFibro as an intermediate fibroblast population, which exhibit pro-inflammatory properties, attracting and activating FOLR2+ macrophages. This new population correlates with poor prognosis in patients with mild-to-moderate CKD. Therefore, CXCL-iFibro may be both a prognostic marker as well as a therapeutic target to prevent CKD progression.

## Methods
### Cohorts of patients
The reporting of clinical data complies to the STROBE guidelines. The study developed here is based on samples taken from surgical residues or kidney biopsies available after histopathologic analyses and not required for diagnosis. There is no interference with clinical practice. Analysis of kidney samples was performed in accordance with the relevant national law and with recognized ethical guidelines (Declaration of Helsinki) on the protection of people taking part in biomedical research. All patients hospitalized at Necker or Bichat hospital received a welcome booklet explaining that their samples may be used for research purposes. All patients included in our study were thus informed by their referring nephrologist that biological samples collected through standard clinical practice could be used for research purposes and they gave their verbal and written informed consent. In the case of patient refusal, which could be either orally expressed or

written, residual samples were not included in our study. The pathology lab of Necker and Bichat hospital are authorized to store and manage human biological samples according to French legislation (declaration number DC-2009-955).

**Patients with Polycystic Kidney Diseases (PKD).** PKD is an autosomal dominant kidney disease leading to the development of cysts and ultimately to end-stage renal disease around the age of 50. Because of the size of these organs, it is sometimes needed to undergo a nephrectomy to make room for a kidney transplant. After macroscopic examination of the explanted kidney, and harvest of the zone of interest for diagnostic, samples for research were collected, fixed in formol then embedded in paraffin (FFPE samples). Thirteen patients (male, N = 7, female, N = 6) samples have been collected for this study.

**Kidney biopsies with interstitial fibrosis.** Kidney biopsies samples were from the routine diagnostic samples stored in the pathology department of Necker hospital. Patients (male N = 8, female N = 4) with diagnosis of vascular nephropathy with interstitial fibrosis from 0 to 80% were selected. Briefly kidney biopsy was performed with a 16-gauge needle, and the sample was immediately immerged in acetic acid formaldehyde (AFA) fixative. After fixation, tissue was embedded in paraffin and stored as FFPE samples. Interstitial fibrosis was evaluated independently of our study by a pathologist.

**Pre-transplant kidney biopsies from deceased donors.** Two male patients' kidney biopsies were harvested in tha back table after organ harvesting from two deceased kidney donors. Kidney biopsy sample was immediately immerged in acetic acid formaldehyde (AFA) fixative. After fixation, tissue was embedded in paraffin and stored as FFPE samples.

**Nephrectomy samples with fibrosis for spatial transcriptomics.** Nephrectomy samples for spatial transcriptomics were from the routine diagnostic samples stored in the pathology department of Bichat hospital. Samples of two male patients who underwent a total nephrectomy for kidney cancer and who had non-tumoral tissue available were systematically reviewed by a pathologist. Briefly, if the patient presented kidney dysfunction at the time of the nephrectomy, a sample of non-tumoral kidney was harvested and examined by a pathologist after fixation in formol and paraffin embeddings. Patients with interstitial fibrosis were selected.

**NEPTUNE cohort.** NEPTUNE is a multicenter observational, prospective cohort study of children and adults with proteinuric glomerular disease, for which comprehensive clinical and molecular phenotyping data was collected at 21 sites at the time of first clinically indicated renal biopsy[70]. Biospecimens were collected after informed consent and with approval of the local ethics committee[95]. Pathologic diagnosis is confirmed by review of digital whole-slide images by study pathologists[70,96]. Patients with secondary glomerular disease (such as diabetic kidney disease, lupus nephritis, and amyloidosis) were excluded. For this analysis, 134 adult patients (male, N = 84, female, N = 50) with mild to moderate kidney dysfunction, defined as patients with an eGFR calculated by a Modification of Diet in Renal Disease (MDRD) > 45 mL/min/1.73 m$^2$ for which transcriptomic data as well as follow up clinical data were available were included in the analysis. Progressor status was defined as a decrease of more than 5 mL/min/1.73 m$^2$ per year.

**Sex and gender reporting.** We used the biological variable "sex". It was self-reported.

**Ethics statement.** All the performed studies were validated by the local ethics committee from Institut Curie (Poesie DATA220128) and

Assistance-Publique Hopitaux de Paris (APHP): Comité d'Ethique de la Recherche (CER) Paris Nord, Institutional Review Board -IRB 00006477- of HUPNVS, Paris 7 University, AP-HP, file CER-2022-174).

## Single cell RNAseq data analysis
Publicly available single cell RNAseq data for 12 patients with kidney disease from Kuppe et al.[8] including matrix count and annotations were downloaded from Zenodo data archive (https://zenodo.org/record/4059315, https://doi.org/10.5281/zenodo.4059315).

**Stromal cells**
**Selection of stromal cells. Quality control and doublet elimination.** First, stromal cells from the dataset were selected, based on their annotation by the authors. As a quality-control step, we first filtered out low-quality cells, empty droplets, and multiplet captures based on the distribution of the unique genes detected (nonzero count) in each cell for each patient. Cells with less than 200 genes were excluded. Doublets were identified using the DoubletFinder method (https://github.com/chris-mcginnis-ucsf/DoubletFinder), homotypic doublet proportion estimation was done using the function *modelHomotypic* and doublet identification using the doubletFinder_v3 function, high confidence doublet identified were removed for downstream analysis.

**Normalization and scaling.** Library-size normalization of each cell using *NormalizeData* function with default parameters from Seurat[97] was performed. Scaling using *ScaleData* function from Seurat was performed by regressing on the number of count and patient identity.

**Clustering and data visualization.** Principal component analysis (PCA) dimensionality reduction was run using default parameters. Number of the included components (PCs) was assessed using the *JackStraw* procedure implemented in *JackStraw* and *ScoreJackStraw* functions. Thirty PCs were conserved. Graph-based clustering approach was used to cluster the cells from the first dataset using *FindNeighbours* (k = 20) and *FindClusters* functions (res = 0.6). Thirteen stromal cells clusters were obtained at this resolution. For visualization of the data, the nonlinear dimensional reduction technique UMAP was applied using the *RunUMAP* function from Seurat with default parameters.

**Analysis of differential gene expression and signaling pathways.** Genes specifically upregulated in each of the 13 clusters of the first dataset were identified using the Seurat function *FindAllMarkers* with default parameters. For each cluster, functional enrichment was done using the Metascape tool (http://metascape.org) using all genes significantly upregulated in each of the 13 initial clusters (one cluster vs. all other clusters; function *FindAllMarkers* with following parameters: logfc.threshold = 0.25, test = wilcox for Wilcoxon rank sum test). Because cluster 0 to 5 represented more than 85% of the cells, we focused our further analysis on these clusters. The pathways listed in Table S1 were top-ranked.

**Label transfer.** To identify similarities between the original annotation of cells from the PDGFRβ+ sorted cells single-cell RNAseq dataset from ref. 8 and our annotations, we used the Label Transfer algorithm using the *FindTransferAnchors* and the *TransferData* functions from Seurat. The dataset of 2495 mesenchymal cells with our own annotation described in Fig. 1j was used as a reference, and the dataset of PDGFRβ+ sorted cells, described in Fig. S1d, with original annotations from ref. 8 was used as a query. The alluvial plot was then generated using the *do_AlluvialPlot* function from SCpubr package, using the original annotations as the first group, and the predicted Id in the final group. To identify the temporal dynamics of inflammatory fibroblasts in mice after UUO, we analyzed the single RNAseq data from ref. 42 and subset the dataset to fibroblasts and myofibroblasts according to original

annotations. The dataset was downloaded from the National Center for Biotechnology Information Gene Expression Omnibus database (accession number GSE140023). We used the Label Transfer algorithm using the *FindTransferAnchors* and the *TransferData* functions from Seurat. The dataset of 2495 mesenchymal cells with our own annotation described in Fig. 1j was used as a reference. The proportion of cells according to their *predicted.id* was then plotted in Fig. 2j.

**Gene signatures analysis of CAF-S1, CAF-S4 and CAF-S1 Clusters.** Specific gene signatures were previously published in ref. 29 for CAF-S1 and CAF-S1 clusters. CAF-S4 gene signature is listed in Table S2. Z-score average was calculated for each signature, and then plotted on the generated UMAP using *FeaturePlot* function from Seurat.

**Deconvolution of bulk RNAseq.** Publicly available bulk RNAseq data from 15 mice undergoing unilateral ureteral obstruction (UUO) were downloaded on the Gene Expression Omnibus under the number GSE118339. After normalization and log transformation of the Transcript per Million (TPM), an ECM expression score was calculated. The list of core-matrisome genes from Naba et al.[39] was selected to calculate a z-score average according to the different experimental conditions (sham, UUO day3, UUO day7, UUO day14). For deconvolution, the cellular atlas was generated from publicly available single cell RNAseq data for 12 patients with kidney disease from Kuppe et al. was used (n = 49 226 cells after doublet removal and quality controls). Cell type annotation was similar to that of the original article (annotation level "V3"), except for the stromal where we annotated pericytes, inflammatory fibroblasts and ECM-secreting myofibroblast according to our annotation. Human to mouse orthology was performed using *Nichnetr* package. Deconvolution was performed using *BayesPrism* package[40], using our cellular atlas as input for prior information, after filtering of ribosomal genes, *Actin*, Chromosome X or Y expressed genes. A specific signature using differentially expressed genes using *get.exp.stat* function from *BayesPrism* (*cell.count.cutoff = 50*) then *select.marker* with *pval.max = 0.01* and *lfc. min = 0.1* was identified. Then deconvolutionof the bulk RNAseq data was performed using the *new.-prism* followed by the *run.prism* function, with following parameters: *outlier.cut = 0.01, outlier.fraction = 0.1*. Default parameters to control Gibbs sampling and optimization were used otherwise.

**Gene signatures of CXCL-iFibro.** Specific gene signature from CXCL-iFibro was defined by performing a differential analysis between clusters (Wilcoxon rank-sum test) with the Seurat function *FindAllMarkers*. Differentially expressed genes between clusters (one cluster vs. all other clusters) with a *P*adj < 0.05 were selected. This signature was used for detecting CXCL-iFibro in bulk RNA-seq data from kidney biopsies of the NEPTUNE cohort (see section Bulk RNAseq analysis from the Neptune cohort). Genes defining this signature are detailed in Table S10. Expression score of this signature was calculated using *AddModuleScore* function from Seurat, and then plotted on the generated UMAP using *FeaturePlot* function from Seurat. Violin plot was generated using *VlnPlot* function from Seurat.

**Trajectory inference analysis.** Trajectory inference models were performed using Monocle 3 package[66]. Briefly, after normalization and scaling, a trajectory was defined using the *learn_graph* function from Monocle 3 and plotted with *plot_cells* with default parameters. Root of the trajectory was defined using two methods. First, data from literature defined pericytes as a major source of myofibroblast during kidney disease. Then, the function *get_earliest_principal_node* function already described helped us to define the right root[66]. A pseudotime was then calculated, allowing to classify cells in order of this pseudotime. Genes of interest have been ordered in pseudotime using the *plot_genes_in_pseudotime* function.

**Transcription factor inference.** Transcription factor inference was performed using the Dorothea package[65]. Briefly, Dorothea regulons with high confidence score (level A, B and C) were conserved. Then a transcription activity score was computed using Viper for each cell. After computing the mean scaled activity score per cluster, the 20 most variable transcription factors between clusters of interest were selected for representation (in our study, we selected cells at the node of differentiation between CXCL-iFibro and ECM-secreting myofibro on the Monocle3 UMAP using *choose_cells* function from Monocle3). CXCL-iFibro corresponds to cluster 0 and ECM-secreting myofibro clusters 3 + 5).

**Gene signatures of TCF4 target genes.** Gene identified in Dorothea as being regulated by TCF4 were used as the TCF4 target gene signature. Gene with no expression in our dataset, as well as one expressed in all cells at high level were removed. The gene list is shown in Table S13. The expression score of this signature was calculated using *AddModuleScore* function from Seurat, and then Violin plot was generated using *VlnPlot* function from Seurat.

#### Myeloid cells
**Selection of myeloid cells. Quality control and doublet elimination.** Myeloid cells from the dataset were selected, based on their annotation by the authors. Quality control, normalization, and scaling as well as clustering and data visualization were performed similarly than for stromal cells (except a resolution of 0.3 for the *FindClusters* function).

**Cell-Cell interaction prediction.** To predict cell-cell interaction, we used the Cellchat package version 1.6.1[64] on fibroblasts and myeloid cells. Briefly, ligand-receptor interactions between cells were estimated through a manually curated database containing 1939 validated molecular interactions. Inference of cell communications were performed through identification of over-expressed secreted ligands and over-expressed receptors in different cell groups, using the *identifyOverExpressedGene* and *identifyOverExpressedInteractions* functions. The communication probability between 2 cell types was then inferred with the *computeCommunProb* function. For the representation of the number of interactions, the strength of the interactions, and ligand-receptor pairs, we calculated the aggregated cell-cell communication using the *aggregateNet* function, by subsetting the source to CXCL-iFibro for clarity purposes. For the purpose of the analysis, CD16+ monocytes and CD16+ RUNX3- monocytes were pooled in one group of cells called CD16+ monocytes.

**Generation of the cellular atlas.** Publicly available single cell RNAseq data for 12 patients with kidney disease from Kuppe et al. was used for the generation of the atlas (n = 49 226 cells after doublet removal and quality controls). Cell type annotation was similar to that of the original article, except for the stromal and myeloid populations, for which we used the annotation we defined in the study. In total, we identified 36 cell types in this dataset.

#### Spatial transcriptomics
Two CKD FFPE samples were selected for spatial transcriptomics analysis based on their tissue structure and RNA quality (DV200 > 50%). The Visium Spatial for FFPE Gene Expression Kit, Human Transcriptome (10X Genomics, #PN 1000338) was used according to manufacturer's instructions. Briefly, the 10 μm thick sections were placed on Visium Spatial Gene Expression slides. Visium slides containing FFPE tissue sections were first deparaffinized and then stained with Hematoxylin. The stained slides are then coverslipped and

imaged with a Philips scanner. After the coverslip is removed, a decrosslinking step is performed.

The spatial gene expression process, including probe hybridization, probe ligation, probe release and extension were performed according to the manufacturer's instructions. cDNA quality was evaluated using Agilent High sensitivity DNA Kit (Agilent, #5067-4626) and spatial gene libraries were constructed using the Visium Spatial Library Construction Kit (10X Genomics, PN-1000184). Spatial data processing was carried out using SpaceRanger software v1.2.2 (10X Genomics). Spatial raw base call (BCL) files were demultiplexed and mapped to the reference genome GRCh38. Loupe Browser from 10X Genomics was used to align the slide's barcoded spot patterns and select spots in the tissue. The resulting count matrices were processed in Seurat v4.1.0 for log2 normalization, scaling, and dimension reduction.

Consents to use their samples and to publish clinical data have been obtained from the patients.

## Deconvolution analysis

The scRNA-seq dataset (n = 49 226 after quality control) from Kuppe et al.[8] was used to perform the deconvolution for the two Visium samples. The dataset covered 36 different cell types. Cell type annotation from the original article were conserved, except for the stromal and myeloid populations, for which we used the annotation we defined in the study. The spatial sections were decomposed using cell2location version 0.1[55], implemented in Python3. The CKD atlas was used to compute reference cell type signatures with patient ID as categorical covariate and default parameters. Only cells coming from CKD and not healthy tissue were kept, because we only analyzed sections from patients with CKD. The spatial mapping of cell types was performed by supplying each Visium section to Cell2location and setting *N_cells per location* to 15 and *detection_alpha* to 200 after manual examination of the tissue. 30,000 epochs were used.

After the deconvolution of the spatial sections was performed, Non-Negative Matrix Factorization (NMF) was applied to the deconvolution output using the Cell2location function *run_colocation* with default parameters. NMF is a dimension reduction technique that decomposes the data matrix into a product of two lower-dimensional matrices. In this case, the data matrix was the *q05_cell_abundance_w_sf* cell abundancy matrix obtained from the Cell2location deconvolution, and the factorization was performed to identify underlying patterns and structure in the data. The algorithm was run three times with different random initializations to ensure that the final result was robust and not influenced by the initial conditions. After the NMF decomposition was completed, the resulting component matrices were used to generate a heatmap and clustering analysis to identify patterns in the data and relationships between different cell type.

## Bulk RNAseq analysis from the Neptune cohort

The renal biopsy sample was manually dissected and isolated into tubulointerstitial and glomerular compartments per established protocol[98]. The tubulointerstital dataset was used for further analysis. For RNA-sequencing (RNA-seq) profiles, mRNA samples were prepared using the Illumina TruSeq mRNA Sam- ple Prep v2 kit. Multiplex amplification was used to prepare cDNA with a paired-end read length of 100 bases using an Illumina HiSeq2000. RNAseq was performed by the University of Michigan Advanced Genomics Core (https://brcf. medicine.umich.edu/cores/advanced-genomics/). Quality of the sequencing data were assessed using the FastQC tool (http://www. bioinformatics.babraham.ac.uk/ projects/fastqc/). Read counts were extracted from the fastq files using HTSeq (version 0.11). RNA-seq profiles from different batches were voom-transformed and batch corrected using ComBat[99]. The CXCL-iFibro expression score was defined as the mean expression of genes that compose the CXCL-iFibro signature.

## Isolation and culture of kidney primary fibroblasts

Fresh samples from PKD explant received after surgery were cut into fragment of approximatively 1mm³. Fragments were harvested in either non-coated or type I Collagen coated (9 µg/mL, Institut De Bio-technologie Jacques Boy, #207050357) Petri dishes. Cells were cultured in DMEM (Gibco, #41966-029) supplemented with 10% heat inactivated FBS (Biosera, #FB-1003-500) and 1% streptomycin and penicillin (Sigma, #p4333) for 2–3 weeks at 37 °C, in an incubator delivering 5% CO2 and 1.5% of O2. Media was renewed every 3 days for 2–3 weeks, until cells reached 50% confluency to perform first passage. All experiments with fibroblasts were performed with fibroblasts from passage 5 to 10, to avoid cellular senescence.

## Isolation of CD14 + PBMC

PBMC were obtained from healthy donor peripheral blood obtained from "Etablisssement Francais du sang". PBMC were isolated using Lymphoprep (STEMCELL #07861), and CD14 + PBMC were isolated using a specific isolation kit (Miltenyi #130-050-201).

## Characterization of fibroblasts phenotype by immunofluorescence

$2 \times 10^4$ fibroblasts were cultured on glass coverslip coated or not with type I collagen, as described above. On the next day, cells were approximatively at 50% confluency. After PBS wash, cells were fixed in paraformaldehyde 4% for 15 min. After several washes with PBS, cells were stained according to the immunofluorescence protocol described below.

## Characterization of fibroblasts phenotype by RNA sequencing

For RNAseq experiments, fibroblasts cultured on collagen- or plastics-dishes, as described above, were collected. RNAs were then extracted using Qiagen miRNeasy Kit (Qiagen, #217004) according to the manufacturer's instructions. RNA integrity and quality were analyzed using the Agilent RNA 6000 nano Kit (Agilent Technologies, #5067-1511). cDNA libraries were prepared using the TruSeq Stranded mRNA Kit (Illumina, #20020594) followed by sequencing on NovaSeq (Illumina). Reads were mapped on the human reference genome (hg38; Gencode release 26) and quantified using STAR (version 2.5.3a) with parameters "outFilterMultimapNmax = 20; alignSJoverhangMin = 8; alignSJDBoverhangMin = 1; outFilterMismatchNmax = 999; outFilterMismatchNoverLmax = 0.04; alignIntronMin =;20; alignIntronMax = 1000000; alignMatesGapMax = 1000000; outMultimapperOrder = Random." Only genes with one read in at least 5% of all samples were kept for further analyses. Normalization was conducted with DESeq2 R package and raw read matrix was log2 transformed using the *rlog* function from DESeq2 package. Differentially expressed genes depending on the culture condition were identified with DESeq2 R package, after subsetting the count matrix to inflammatory genes (as defined in the Hugo Gene Nomenclature Committee (HNGC) https://www.genenames.org/) or to matrisome-core genes described in[39].

## Transwell migration assay

$3 \times 10^4$ fibroblasts were plated on the lower chamber of transwell 24 wells plate (5 µm pore size, Corning HTS Transwell 24 wells #CLS3421) coated or not with type I Collagen in the lower part of the transwell, in 500 µL of DMEM supplemented with 10% heat-inactivated FBS and 1% Penicillin streptomycin at 1.5% O2. After 24 h, medium was changed for DMEM supplemented with 1% heat inactivated FBS and 1% Penicillin streptomycin. Then $1.5 \times 10^5$ CD14 + PBMC in 150 µL of DMEM supplemented with 1% heat inactivated FBS and 1% Penicillin streptomycin were plated in the upper chamber and incubated at 37 °C for 6 h. After incubation, CD14 + PBMC were harvested in the upper and lower chamber separately and incubated with 0.5 µl of 10 µm carboxylated beads (Polyscience #18133) and DAPI (3 µM). CD14 + PBMC counting

was performed by Flow Cytometry on a LSRII (BD biosciences) using precision beads for normalization and expressed as percentage of migration, being the ratio of the CD14+ cell number in the lower chamber by the total number of cells.

## Coculture of fibroblasts and CD14 + PBMC

For all antibodies used in our study, see also Table S14 listing antibody references and dilutions.

For macrophage analysis, $3 \times 10^4$ fibroblasts were plated in 24 wells plate coated or not with type I collagen for 24 h. Then $1.5 \times 10^5$ CD14 + PBMC in DMEM supplemented with 1% heat inactivated FBS and 1% Penicillin streptomycin were added. After 24 h adherent and non-adherent CD14+ monocytes were harvested, washed and stained first with LIVE/DEAD dye (1:1000, Thermo Fischer, #L34955) for 10 min at room temperature (RT) in PBS to exclude dead cells. Cells suspensions were then incubated for 20 min at RT with antibody mix containing anti-CD14-BV510 (1:50, BD biosciences 563079), anti-CD16-BV650 (1:50, BD biosciences 563692), anti-CD206-BV711 (1:50, BioLegend 321136), anti-FOLR2-PE (1:50, BioLegend 391704), anti-TREM2-biotinilated (1:50, R&D BAF1828) followed by incubation with streptavidin-PECy5 (1:100, BioLegend 405205) for 15 min. Isotype controls were BV510 mouse IgG1k (1:50, BD biosciences 56294, BV650 mouse IgG1k (1:50, BD biosciences 563231), BV711 mouse IgG1k (1 :50, BD biosciences 56344), mouse IgG1k (1:50, BioLegend 400112), Goat IgG control (1:50, R&D AB-108-C). Cells were analyzed by LSRFortessaTM analyzer (BD biosciences). Data were examined using FlowJo 10.5.2.

For fibroblast phenotype analysis, $2 \times 10^4$ fibroblasts were plated in 24 wells plate with 13 mm glass coverslips coated or not with type I collagen for 24 h. Then $1 \times 10^5$ CD14 + PBMC in DMEM supplemented with 1% heat inactivated FBS and 1% Penicillin streptomycin were added. After 24 h of coculture, cells were washed with PBS, then fixed with 4% PFA for 15 min. After three washes with PBS, cells were stained following immunofluorescence protocol described below.

## WNT/β-catenin pathway stimulation

$2 \times 10^4$ fibroblasts were plated in 24 wells plate with 13 mm glass coverslips coated with type I collagen for 24 h. Cells were then stimulated with either control (DMSO), Wnt agonist 1 at the dose of 700 nM (Sigma-Aldrich 681665, CAS 853220-52-7), or Wnt agonist 2 at the dose of 10 μM (Sigma-Aldrich 681667, SKL2001) for 24 h. Wnt agonist 1 is a cell-permeable pyrimidine compound that acts as a potent and selective activator of Wnt signaling without inhibiting the activity of GSK-3β. It is shown to mimic the effect of Wnt and induce β-catenin and TCF (T-cell fate)-dependent transcriptional activity. Wnt agonist 2 is a cell-permeable imidazolyl-isoxazolamide compound that upregulates β-catenin-regulated transcription by disrupting β-catenin and Axin interaction, thereby preventing β-catenin phosphorylation (Ser33/Ser37/Thr41/Ser45) and proteasomal degradation, without affecting the activities of GSK-3α/β. After that, cells were washed with PBS, then fixed with 4% PFA for 15 min. After three washes with PBS, cells were stained following immunofluorescence protocol described below.

## Nuclear translocation of β-catenin in fibroblasts upon co-culture with CD14 + PBMC evaluated by western blots

$5 \times 10^4$ fibroblasts were seeded in 24-well plates coated with type I collagen in DMEM supplemented with 10% heat-inactivated FBS and 1% Penicillin streptomycin at 1.5% $O_2$ overnight for complete adherence. Then, the medium was removed and $2,5 \times 10^5$ CD14$^+$ monocytes resuspended in 500 μl of DMEM supplemented with 1% heat inactivated FBS and 1% Penicillin streptomycin were added. At that time, cells were stimulated with either DMSO or β-catenin/TCF Inhibitor III, iCRT3 (Sigma Aldrich, 219332) or PORCN inhibitor, C59 (Sigma Aldrich, 5004960001) at the dose of 20 μM. 24 h after co-culture, adherent and non-adherent cells were harvested, and the fibroblasts were separated

from the monocytes by negative selection using a specific isolation kit (Miltenyi #130-050-201). Nuclear and cytoplasmic fractionation was performed by following the manufacturer's protocol for NE-PER™ Nuclear and Cytoplasmic Extraction reagents (ThermoFisher #78833). Briefly, fibroblasts were collected in 1,5 ml microcentrifuge tubes and washed with PBS, and ice-cold cytoplasmic reagents (CER I and II) were added. Cells were vigorously vortexed at the highest speed and centrifuged at 16,000 g for 10 min; then the supernatant containing the cytoplasmic extracts was transferred to a new ice-cold tube. The cell pellet was resuspended with a Nuclear extraction reagent (NER), vortexed and the extract was obtained after centrifugation. All the steps were performed on ice and the samples were next sonicated for 15 min (cycles of 30 s ON/30 s OFF) and centrifuged during 10 min at 13.000 × g at 4 °C. The western blot was then performed as detailed in *#Protein extraction and western blot*. EIF4A1 and Histone-H3 were used as the respective cytoplasmic and nuclear makers for monitoring fraction quality by Western blot. Primary antibodies used were EIF4A1 (1:1000, Cell signaling #2490), Histone H3 (1:10000, abcam #ab1791) and β-catenin (1:1000, cell signaling #9562).

## β-catenin/TCF interaction inhibition during fibroblasts-CD14 + PBMC coculture

$2 \times 10^4$ fibroblasts were plated in 24 wells plate with 13 mm glass coverslips coated or not with type I collagen for 24 h. Then $1 \times 10^5$ CD14 + PBMC in DMEM supplemented with 1% heat-inactivated FBS and 1% Penicillin streptomycin were added. At that time, cells were stimulated with either DMSO or β-catenin/TCF Inhibitor III, iCRT3 (Sigma Aldrich, 219332), at the dose of 20 μM. For fibroblast analysis, after 24 h of coculture, cells were washed with PBS, then fixed with 4% PFA for 15 min. After three washes with PBS, cells were stained following immunofluorescence protocol described below. For macrophage analysis, after 24 h of coculture and stimulation, adherent and non-adherent CD14+ monocytes were harvested, washed, and stained as described above (coculture of fibroblasts and CD14 + PBMC section).

## Immunohistochemistry

**Experiments**. 3-μm sections of paraffin-embedded human kidneys underwent antigen retrieval by microwave heating in a solution of EnVision FLEX Target Retrieval Solution high-pH (Dako, K800421) for 20 min. Staining was performed using the Lab Vision Autostainer (Thermo Fisher Scientific). Sections were incubated with the following primary antibodies: FAP (1:100, abcam ab207178), SFRP1 (1:200, Abcam ab126613), SFRP4 (1:200 abcam ab154167), RAMP1 (1:400, EMD Millipore MABS1904). Antigen detection was performed using HRP-DAB revelation system, using Vectastain elite ABC kit, Rabbit IgG. Slides were counterstained with Mayer hematoxylin freshly prepared (Dako, #S3309) and submitted to serial gradients of xylen and mounted with coverslip in an automatic device (Sakura, Tissue-Tek DRS). Images were acquired by a Philips scanner.

**Quantification**. For each slide, staining of fibroblasts markers was evaluated as a histological score (H-score) defined by staining intensity (from 0 to 4) multiplied by the percentage of stained interstitial cells (from 0% to 100%). The whole section was considered, and quantification was performed at 20x magnification. Fibrosis was evaluated as the percentage of fibrotic tissue per field. Fibrosis percentage was evaluated blindly by two independent researchers with very good concordance.

## Immunofluorescence
### Experiments

**In vitro**. Fibroblasts were cultured on coverslips and fixed in 4% paraformaldehyde for 15 min, permeabilized in PBS containing Triton X-100 0.1%, BSA 3%, then incubated overnight at 4 °C with primary

antibodies: mouse monoclonal anti-αSMA (1:200, DAKO clone 1A4 ref M0851) rabbit monoclonal anti-SFRP1 (1:50, Abcam ab126613), rabbit monoclonal anti-SFRP4 (1:400, Abcam ab154167), rabbit monoclonal anti-E-Cadherin (1:200, Cell Signaling Technology, 4065) mouse monoclonal anti-β-catenin (1:1000, Bio SB, Ref: BSD 5088). After 3 washes with PBS, cells were incubated in secondary antibody goat anti mouse alexa fluor 488 (1:400, Invitrogen A11001) or goat anti rabbit alexa fluor 555 (1:400, Invitrogen A21428). Nuclei were counterstained with Hoescht before mounting with anti-fade diamond mounting medium (Invitrogen P36970). Images were acquired by an upright widefield Apotome (Zeiss) or LSM 700 Zeiss confocal. At least 10 fields at 40x were acquired.

**For human tissue section.** 3-μm sections of paraffin-embedded kidneys underwent antigen retrieval by microwave heating in a solution of EnVision FLEX Target Retrieval Solution high-pH (Dako, K800421) for 20 min. Sections were incubated overnight at 4 °C with the following primary antibodies: CD68 (1:200, Abcam ab955), αSMA (1:400, Dako clone 1A4 M0851), FOLR2 (1:200, Invitrogen MA5-26933), TREM2 (1:100, R&D MAB17291) or collagen I (1:400, Invitrogen MA1-26771). Antigen detection was performed with alexa-fluor coupled antibodies: alexa fluor 488 goat anti mouse (for αSMA, Collagen I, CD68, FOLR2, Invitrogen A11001), alexa fluor-555 goat anti rabbit (for FAP, SFRP1, SFRP4 Invitrogen A21428), alexa fluor-647 goat anti rabbit (for FAP, SFRP4 Invitrogen A21245) or alexa fluor-555 goat anti rat (for TREM2 Invitrogen A21434). Nuclei were counterstained with Hoescht before mounting with anti-fade diamond mounting medium (Invitrogen P36970). Images were acquired by an upright widefield Apotome (Zeiss) or LSM 700 Zeiss confocal.

### Quantification
**In vitro.** For quantification of MFI of markers of interests, at least ten fields at 40x were acquired. Region of Interest (ROI) was manually defined as the contour of an individual cell. At least 100 cells were quantified per condition per independent experiment, and the average of MFI per condition was then compared. For β-catenin nuclear MFI, nucleus was defined as a ROI using the Hoescht staining by performing binarization and thresholding. Then, MFI of β-catenin staining was quantified in this ROI.

**For human tissue section.** For quantification of SFRP1, FAP, CD68, FOLR2 per mm², each 40x magnification acquired image was segmented in images of 18.600 μm² (136 × 136 μm images). Positive cells per field were then manually counted. Then, the positive number of cells per mm² could be calculated. For SFRP1 or FAP colocalization with Collagen I, images were segmented in 80 × 80 μm images. The percentage of stained area was quantified after binarization of the staining using thresholding. Correlation between the percentage of the stained area of interstitial SFRP1 or FAP in these small images is then a reflect of colocalization.

### Protein extraction and western blot
**Protein extraction.** Cells cultured in a 6 well plate coated or not with Collagen I were washed with cold sterile PBS (Gibco #14190) and collected after the addition of 150 μl of Laemmli buffer (BioRad, #1610737) supplemented with DTT (Thermoscientific, #11896744) and scratching. The solution was next boiled at 95 °C for 5 min. Samples were next sonicated for 10 min (cycles of 30 s ON/ 1 min OFF) and centrifuged during 10 min at 13.000 x g at 4 °C. The protein extract was then short-term stored at −80 °C.

**Western blot.** A volume of 15 μl of proteins was loaded onto a NuPAGE Novex 4–12% bis tris midi protein gel ten wells (Invitrogen, #WG1403BOX). The gel was transfer onto a 0,45 μm nitrocellulose membrane (GE Healthcare #10600002) in 1X TGS buffer (Biorad #161-

0772) at 100 V for 2 h at 4 °C. Membrane was next blocked during 30 min in TBS-Tween 0.1% complemented with 5% BSA (Euromedex #04-100-812-C) before blotting with primary antibodies diluted in TBS-Tween 0.1% complemented with 5% BSA at 4 °C overnight. Primary antibodies used were FAP (1:1000, abcam ab207178), SFRP1 (1:500, Abcam ab126613), SFRP4 (1:1000 abcam ab154167), αSMA (1:1000, Dako clone 1A4 M0851), actin (1:10.000; Sigma #A5441). The next day, incubation with secondary antibody anti-mouse (Jackson ImmunoResearch Laboratories, INC., #115-035-003) or anti-rabbit (Jackson ImmunoResearch Laboratories, INC., #115-035-003) in TBS-Tween 0.1% + 5% BSA for 1 h at RT was done. The membrane was incubated for 1 min at RT with ECL (ratio 1:1) (Western Lightning Plus-ECL, PerkinElmer, #NEL105001EA). The detection of the signal was done in a Chemidoc device for detecting chemiluminescence. The protein bands were analyzed by ImageJ software for protein quantification.

### Statistical analysis
The graphical representation of the data and statistical analyses were done using R environment (https://cran.r-project.org). Bar plots or scatter plots are represented with mean ± standard error of the mean (SEM). Statistical tests used agree with data distribution. To assess the normality of the distribution of variables, we first applied the Shapiro–Wilk test. According to normality Shapiro-Wilk test, parametric or non-parametric two-tailed tests were applied. The correlation coefficient and its significance between two independent variables were evaluated by Spearman's correlation test. Survival curves were established using the Kaplan-Meier method and compared with the Log-rank test using survival R package. Stratification of patients for survival analyses was performed using an iterative method. Patients were separated in two groups according to their expression of CXCL-iFibro transcriptomic signature or FOLR2. Each value of expression was tested iteratively. The threshold with the lowest p-value was selected as the threshold differentiating low and high expression levels of CXCL-iFibro transcriptomic signature or FOLR2. Univariate analysis was performed using the Wald test. A multivariate Cox model was applied, using as variable all parameter with a significance of $p < 0.1$ in univariate analysis. All applied statistical tests are indicated in the legends. Differences were statistically significant when p-values were ≤ 0.05.

### Reporting summary
Further information on research design is available in the Nature Portfolio Reporting Summary linked to this article.

## Data availability
New generated data, as well as source data and codes have been deposited in Figshare. Raw counts for bulk RNAseq for primary human fibroblasts can be downloaded on Figshare under the https://doi.org/10.6084/m9.figshare.24049380 (https://figshare.com/search?q=10.6084%2Fm9.figshare.24049380). Spatial transcriptomics processed data for the two patients with kidney fibrosis can be downloaded on Figshare under the https://doi.org/10.6084/m9.figshare.24049410 (https://figshare.com/search?q=10.6084%2Fm9.figshare.24049410). Raw data have been deposited on the European Genome Phenome Archive (EGA) and can be requested through the Data Access Committee EGAC00001000581. Publicly available single cell RNAseq data for 12 patients with kidney disease from Kuppe et al.[8] including matrix count and annotations were downloaded from Zenodo data archive (https://zenodo.org/record/4059315, DOI: 10.5281/zenodo.4059315). Publicly available bulk RNAseq data from 15 mice undergoing unilateral ureteral obstruction (UUO) were downloaded on the Gene Expression Omnibus under the number GSE118339[38]. The publicly available single cell RNAseq dataset from mice UUO was downloaded in the National Center for Biotechnology Information Gene Expression Omnibus database (accession number GSE140023). Source data are

provided with this paper and under the https://doi.org/10.6084/m9.figshare.24049305 (https://figshare.com/search?q=10.6084%2Fm9.figshare.24049305%20). Uncropped blots are available at https://doi.org/10.6084/m9.figshare.24049938 (https://figshare.com/search?q=10.6084%2Fm9.figshare.24049938). Source data are provided with this paper.

## Code availability

All the codes used for this study are available on Figshare under the https://doi.org/10.6084/m9.figshare.24049350 (https://figshare.com/search?q=10.6084%2Fm9.figshare.24049350). They can be also accessible on GitHub (https://github.com/StressAndCancerLab/).

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

## Acknowledgements

We are grateful to Geraldine Gentric and Agathe Peltier for fruitful discussions and critical reading of the manuscript. We thank Coralie Guerin and Lea Guyonnet at the cytometry core facility. F.M.-G. is a permanent scientist at Inserm. C.C. was supported by the Fondation Chercher et Trouver, R.M. by the Simone and Cino del Duca Foundation, the SIGN'it 2019 program for the Foundation ARC and the post-doctoral grant from Foundation de France (00119142/WB-2021-36276), and HC by the Foundation for Medical Research (FRM, grant number 13683). Y.K. and C.R. were supported by the Institut National du Cancer INCa (CaLYS INCa_11692; CAFHeros INCa_16101). The experimental work was supported by grants from the Ligue Nationale Contre le Cancer (Labelisation), Inserm, Institut Curie (Incentive and Cooperative Program Tumor Micro-environment PIC TME/T-MEGA, PIC3i CAFi), INCa (STROMAE INCa-DGOS-9963, CaLYS INCa_11692, ChemoCAF INCa_16086, CAFHeros INCa_16101) and the Foundation "Chercher et Trouver". F.M.-G. acknowledges both the "French Pink Ribbon Association" and the "Simone and Cino del Duca Foundation" for attribution of their respective Grand Prix, and the FRM for the Rozen Price. F.M.-G. is very grateful to all her funders for providing support throughout the years. Under the remit of the U.K. QUOD Consortium supported by NHS Blood and Transplant, sections were obtained by David A. Ferenbach from renal biopsies from preimplantation renal transplants. This study was also supported by the George M. O'Brien Michigan Kidney Translational Core Center, funded by NIH/NIDDK grant 2P30-DK081943. The Nephrotic Syndrome Study Network Consortium (NEPTUNE), U54-DK-083912, is a part of the National Institutes of Health (NIH) Rare Disease Clinical Research Network (RDCRN), supported through a collaboration between the Office of Rare Diseases Research, NCATS, and the National Institute of Diabetes, Digestive, and Kidney Diseases. Additional funding and/or programmatic support for this project has also been provided by the University of Michigan, the NephCure Kidney International and the Halpin Foundation. The content is solely the responsibility of the authors and does not necessarily represent the official views of the National Institutes of Health. Art from Figs. 2 and 4 have been created with Biorender.com (F.M.-G. account).

## Author contributions

F.M.-G. conceived the project and proposed it to C.C., who both designed the concept of experiments. C.C. performed experiments and acquired data, together with R.M. and C.R. M.R. and A.S. built cohorts of patients, provided human samples and expertize in pathology analyses. R.L. and A.V.-S. performed the spatial transcriptomics. C.C., H.C., and Y.K. performed all bioinformatic and statistical analyses. D.A. and M.O.T. provided human kidney samples. S.E., W.J., and M.K. acquired and analyzed NEPTUNE data. F.M.-G. supervised the entire project and wrote the paper with C.C. with suggestions from the authors.

## Competing interests

F.M.-G. received research support from Innate-Pharma, Institut Roche, Roche, and Bristol-Myers-Squibb (BMS). Other authors declare no potential conflict of interest.
