## [Peer Review File · Nature Communications]

WNT-dependent interaction between inflammatory fibroblasts and FOLR2+ macrophages promotes fibrosis in chronic kidney diseaseREVIEWER COMMENTS

Reviewer #1 (Remarks to the Author):

This manuscript explores the role of inflammatory fibroblasts in regulating macrophage-dependent renal fibrogenesis. Using a scRNA-seq database of kidney biopsies from patients with and without CKD, the authors identify 4 mesenchymal populations that accumulate in CKD and subdivide these into a group of inflammatory fibroblasts (IFs) and matrix-producing fibroblasts (MFs). Based on SFRP1 (IF) and FAP (MF) stains of PKD kidneys, MF stains correlated with and IF staining inversely correlated with presence of fibrosis in the biopsies. In biopsies in earlier stages of ischemic renal disease, IFs were more, and MFs less, prominent. CD68+ macrophages and IFs stain to a similar extent and in similar regions based on the severity of fibrosis. scRNA-seq of myeloid cells in the biopsies reveals 2 populations of FOLR2+ macrophages, among which those lacking CD81 expression accumulate in CKD in parallel with IFs and those expressing CD81 become less prominent in CKD. Spatial transcriptomics support co-localization of IFs and FOLR2+ macrophages. Culturing fibroblasts on collagen yields IF-like fibroblasts with high SFRP1 and low α SMA levels. Cellchat analysis identifies receptor-ligand interactions between IFs and CD14+ monocytes. In culture, IFs drive CD14+ monocyte migration and differentiation to FOLR2+ macrophages. Inversely, CD14+ monocytes drive TCF4 and TCR12 upregulation in IFs and IF transformation towards MFs via a Wnt-dependent pathway. Wnt agonists similarly drive IF \leftrightarrow MF conversion. In biopsies from NEPTUNE patients, IF and FOLR2 gene signatures predict poorer renal outcomes through interdependent mechanisms. Thus, the authors have used complementary, sophisticated techniques to show in vivo that IF and FOLR2+ myeloid cells are prominent early in the fibrosing kidney. They show in vitro that IFs drive fibrogenic macrophage accumulations, and that monocytes drive IF to MF conversion via nuclear localization of beta-catenin. While the analysis is comprehensive, to some extent the findings might be predicted based on known biology. For example, matrix-producing fibroblasts generate collagen and are almost certainly preceded by an earlier fibroblast phenotype. Similarly, Wnts are fibrogenic in several models of kidney disease. The human data are a strength, but much of the in vivo data are correlative. In vivo functional pre-clinical data would markedly strengthen the manuscript. For example, demonstrating that SFRP1 deficiency in fibroblasts (or some other IF deficiency model) and/or that FOLR2 deficiency in myeloid cells protects from renal fibrogenesis would corroborate the in vivo functional relevance of the authors' findings. The authors should probably soften their conclusions regarding a bidirectional loop of IFs regulating macrophages that in turn regulate IF \leftrightarrow MF conversion, as these detailed interactions are not proven in vivo.

Reviewer #2 (Remarks to the Author):

In this manuscript, Cohen et al. highlighted an intermediate fibroblast population CXCL-iFibro with pro-inflammatory properties which give rise to ECM-secreting myofibroblasts and attracted CD14+ monocytes and polarize them into FOLR2+ macrophages. In turn, FOLR2+ macrophages induce the transition of CXCL-iFibro into ECM-secreting myofibroblasts through WNT/ β -catenin signaling. The CXCL-iFibro correlates with poor prognosis in CKD patients, which provides a prognostic marker as well as a potential therapeutic target to limit CKD progression. Although the observation is interesting, several fundamental issues need to be addressed.

Major comments:

1. CXCL-iFibro highly expressed CXCL12 has been reported in CDK (Kuppe et al., 2021). Therefore, it is not new in kidney fibroblast population identified in this manuscript.
2. The specificity of SFRP1 to mark CXCL-iFibro should be demonstrated. SFRP1 was used to mark CXCL-iFibro, while SFRP1 was also expressed in Detox-iFibro. The IHC of SFRP1 showed that many kidney cells were stained by SFRP1 including kidney tubule cells. This may raise a non-specificity concern (Fig 2d), which should be addressed and verified by more specific methods.
3. Co-staining of CD68 and SFRP1 is mainly in the same cell rather in two adjacent individual cells (Fig 3a, 3b). Does CD68 co-express with SFRP1 in myeloid cells (in RNA-seq data)?
4. Quantification results of average MFI (mean fluorescent intensity) were based on about 30 cells per experimental condition, which is too small to reach the conclusion and needs to increase the

sample size. In addition, larger-viewing area with more cells for IF staining should be provided (Fig 4a, 4b, 5a, 5f, 5g, 5i).

5. Cell-cell interaction analysis showed that the highest number of interactions and the highest interaction strength of CXCL-iFibro was with the CD14+ monocytes rather than FOLR2+ macrophage. However, the colocalization results in figure 3 showed that CXCL-iFibro colocalized with FOLR2+ macrophage. This seems contradictory and should be verified.

6. The collagen production is a marker of fibrogenic phenotype and should be analyzed to confirm the phenotype from CXL-iFibro to ECM-secreting myoFibro.

7. Macrophage lineage-tracing method is a more reliable method for tracing FOLR2+ macrophages in chronic kidney disease and should be considered.

Minor comments:

1. Line 116: Only 6 clusters of 13 mesenchymal cells clusters were used in this study. The features of the other 7 clusters of cells should be provided and explain why these cells were discarded.

2. Cell in cluster 5 seems mainly from P9 (Fig. 1a, b).

3. Gene symbols should be in italic, such as figure 1e, 1h, S1c, etc.

4. The scale bars of color are missing in Fig. 1e, 1g, 1h, etc.

5. Several terms of functional enrichment for each cluster were listed in Table S1. Whether these terms were selective or top-rank.

6. The gene lists for scoring CAF gene signatures should be provided (Fig. 1f, g).

7. Line 155: Authors defined cluster 0 as SFRP1+ SFRP4- FAP- RAMP1- (CXCL-iFibro) and cluster 4 as SFRP1- SFRP4+ FAP- RAMP1- (Detox-iFibro), while cluster 4 have a certain expression of SFRP1 and CXCL12.

8. The labels of the axis are missing in Fig. 2a, 2b, S3d-g, 6a, 6b, etc.

9. Line 171: The origin of myofibroblasts is still controversial. How to confirm that other cell types (such as epithelia, endothelia, etc.) are not the sources of myofibroblasts?

10. The IHC of fibrosis-enriched zones on kidney biopsy samples should be provided (Fig 2f, 2g).

11. Line 244: TREM2+ and FOLR2+ macrophages have been described in diabetic kidney disease (Fu et al., 2022).

12. Line 264: A subset of cells in Cluster 1 highly expressed dendritic cell markers (CD1C, CD1E) (Fig. S3f). It inaccurately annotated all cells in cluster 1 as TREM2+ macrophages.

13. The distributions of myeloid cell types in each patient should be provided (Fig. 3d-f).

14. Why the correlation between the number of FOLR2+ cells and fibrosis in PKD samples is opposite in fibrotic kidney biopsies? Since a strong correlation between the number of SFRP1+ CXCL-iFibro and FOLR2+ macrophages, the correlation between the number of FOLR2+ cells and fibrosis in fibrotic kidney biopsies should be similar to the trend in Fig. 2f (Fig3g, 3h).

15. IFN α -peri like, wound-myofibro and TGF β -myofibro colocalize together with endothelium and FOLR2+-CKD macrophages in Fig. 4c which is inconsistent with the result in Fig. 3l. Also, Detox-iFibro colocalize with FOLR2+-CKD macrophages (Fig. S4c).

16. The gating thresholds of flow cytometry analysis are not identical (Fig. 4i).

17. Whether TCF4 and target genes of TCF4 are upregulated following the trajectory of CXCL-iFibro to ECM-secreting myoFibro transition.

18. Line 378: The citation of the figure should be Fig. 5i.

Reviewer #3 (Remarks to the Author):

This manuscript by Cohen et al investigates the role of inflammatory fibroblasts in the progression of chronic kidney disease. Drawing on their background in cancer biology, the group mines publically available scRNAseq dataset to define an inflammatory subset of fibroblasts termed CXCL-iFibro. Pseudotime techniques are used to show that CXCL-iFibro accumulate early in CKD whereas ECM-myofibro accumulate later. These CXCL-iFibro, characterized by SFRP1, are shown to be in areas near macrophages and to be expressing pro-inflammatory cytokines. The investigators use primary human fibroblasts and CD14+ monocytes to show that SFRP1+ fibroblasts can increase monocyte migration and expression of FOLR2 and that these CD14+ monocytes can promote SFRP1 fibroblasts transition to myofibroblasts. The authors use propose that this myofibroblast transition is mediated through Wnt/beta-catenin signaling. Furthermore, they show that patients with biopsies with "high CXCL-iFibro" have a worse clinical prognosis from

bulk RNAseq data in the Neptune registry.

These studies shed light on a potentially important subset of stromal cells with inflammatory characteristics. However, some important concerns persist about these stromal/fibroblast populations and the role of Wnt/beta-catenin signaling.

Major:

1. There are many lessons that the field of fibrosis can learn from cancer biology. However, the nomenclature (e.g. CAF-S1, detox-iCAF, etc) used for cancer associated fibroblasts is a bit confusing as applied to kidney stromal cells. Perhaps this could be simplified in Introduction and some information about how the CXCL-iFibro and myoFibro populations compare to the two key classes of myofibroblasts identified in the Nature 2021 (Kuppe) paper from which the scRNAseq data is used: the fibroblast 1 (Scara5, Meg3+) non-activated population and the myofibroblasts (and associated subsets). They also describe a myofibroblast 2 subset (OGN+COL14A1+) that is postulated to be an intermediate state between pericytes and myofibroblasts. It is critical to know whether the populations described in this manuscript are distinct from those already published or similar but with different nomenclature. The presence/absence of markers that have more meaning to the renal fibrosis field (e.g. PDGRb, osteopontin, NOTCH, etc) should be mentioned at some point either in Results or supplemental data.

2. The staining in Figure 2 (e.g. SFRP1, FAP) does not look specific to stromal cells and higher magnification and colocalization with fibroblast-specific markers is needed. In addition, negative controls (can be in supplemental methods) are critical as fibrotic tissue can often have non-specific staining.

3. The data related to Wnt/beta-catenin is a bit weak. The nuclear staining of beta-catenin (Fig 5g) is not too convincing. The beta-catenin is so ubiquitous that without confocal, it is difficult to determine how much is nuclear. To strengthen the case for active beta-catenin in fibroblasts co-cultured with monocytes, would recommend westerns of nuclear isolates and/or qPCR of axin2, a well-established target of beta-catenin/TCF/LEF. The effect of beta-catenin inhibitor on SFRP4 would be more convincing with a Western. The inhibitor blocks beta-catenin interactions with TCF, and these interactions have been shown to be important for myofibroblast activation. However, they are not necessarily dependent upon Wnt from macrophages (another signal could lead to signaling that augments beta-catenin/TCF signaling such as TGF-beta). Wnt inhibitors (inhibitor of Wnt acyltransferase PORCN) would better test if Wnt signaling were needed for this process.

4. Is the CXCL-iFibro high/low predictive of patient outcome independent of macrophage number? Is this marker providing any information beyond what CD68+ cells would provide?

Minor:

1. The collagen plated fibroblasts are SFRP1+ and do not seem to be "activated". However, to enhance confidence that these accurately model the fibroblast population, some qPCR data showing a proinflammatory phenotype would be useful.

2. It wasn't clear what the difference is between FOLR2 resident macrophages and FOLR2 CKD macrophages. Functionally, are FOLR2 macrophages proinflammatory or profibrotic?

3. Which Wnt ligand is used for "wnt agonist 1/2"?

4. The Neptune database is great, but it is primarily diseases with nephrotic syndrome (not diabetes, hypertension). Some discussion about how inflammation (and potentially inflammatory fibroblasts) may differ in this population versus others.

5. The authors should be cautioned against interpreting CXCL-iFibro as being causative in CKD progression- the data is more consistent with its use as a marker of progression.

6. CXCL-iFibro as a prognostic marker is attractive. A weakness to be acknowledged is that the population that was tested to determine high/low values has not been validated prospectively. It seems that the threshold was chosen based upon the lowest p value.

POINT-BY-POINT RESPONSES TO REVIEWERS' COMMENTS

We first would like to thank the Editor and the three reviewers for their positive evaluation of our work, and for considering our paper suitable for publication in *Nature Communications*, pending modifications. We have now included in the new version of the text the modifications in apparent for addressing their concerns. All these modifications improved our manuscript, and we thank all of you for your suggestions.

For better visualization of the discussion below, initial comments of the reviewers are indicated in italic and our answers in normal style. We also highlighted below the text now added in the new version of the manuscript in blue.

Reviewer #1 (Remarks to the Author):

This manuscript explores the role of inflammatory fibroblasts in regulating macrophage-dependent renal fibrogenesis. Using a scRNA-seq database of kidney biopsies from patients with and without CKD, the authors identify 4 mesenchymal populations that accumulate in CKD and subdivide these into a group of inflammatory fibroblasts (IFs) and matrix-producing fibroblasts (MFs). Based on SFRP1 (IF) and FAP (MF) stains of PKD kidneys, MF stains correlated with and IF staining inversely correlated with presence of fibrosis in the biopsies. In biopsies in earlier stages of ischemic renal disease, IFs were more, and MFs less, prominent. CD68+ macrophages and IFs stain to a similar extent and in similar regions based on the severity of fibrosis. scRNA-seq of myeloid cells in the biopsies reveals 2 populations of FOLR2+ macrophages, among which those lacking CD81 expression accumulate in CKD in parallel with IFs and those expressing CD81 become less prominent in CKD. Spatial transcriptomics support co-localization of IFs and FOLR2+ macrophages. Culturing fibroblasts on collagen yields IF-like fibroblasts with high SFRP1 and low α SMA levels. Cellchat analysis identifies receptor-ligand interactions between IFs and CD14+ monocytes. In culture, IFs drive CD14+ monocyte migration and differentiation to FOLR2+ macrophages. Inversely, CD14+ monocytes drive TCF4 and TCR12 upregulation in IFs and IF transformation towards MFs via a Wnt-dependent pathway. Wnt agonists similarly drive IF \leftrightarrow MF conversion. In biopsies from NEPTUNE patients, IF and FOLR2 gene signatures predict poorer renal outcomes through interdependent mechanisms. Thus, the authors have used complementary, sophisticated techniques to show in vivo that IF and FOLR2+ myeloid cells are prominent early in the fibrosing kidney. They show in vitro that IFs drive fibrogenic macrophage accumulations, and that monocytes drive IF to MF conversion via nuclear localization of beta-catenin. While the analysis is comprehensive, to some extent the findings might be predicted based on known biology. For example, matrix-producing fibroblasts generate collagen and are almost certainly preceded by an earlier fibroblast phenotype. Similarly, Wnts are fibrogenic in several models of kidney disease. The human data are a strength, but much of the in vivo data are correlative. In vivo functional pre-clinical data would markedly strengthen the manuscript. For example, demonstrating that SFRP1 deficiency in fibroblasts (or some other IF deficiency model) and/or that FOLR2 deficiency in myeloid cells protects from renal fibrogenesis would corroborate the in vivo functional relevance of the authors' findings. The authors should probably soften their conclusions regarding a bidirectional loop of IFs regulating macrophages that in turn regulate IF \leftrightarrow MF conversion, as these detailed interactions are not proven in vivo.

We thank the Reviewer for the positive assessment on our manuscript, and for considering the interest of the use of complementary and sophisticated techniques, together with the study of human data from the NEPTUNE cohort. We are really grateful for this positive appreciation of our work.

As recommended by the Reviewer, we have now validated our observations from human beings in a pre-clinical *in vivo* mouse model. To do so, we have now studied a dataset from a time-course experiment in a well-established relevant mouse model of kidney fibrosis, as shown in Pavkovic et al.³⁸. Importantly, data acquired from this mouse model totally validated the evolution of the proportions of the different cellular populations we identified after injury and during fibrosis development in the human pathology. In brief, we used bulk RNAseq data from mouse kidneys collected at different time points after UUU (unilateral ureteral obstruction), i.e., control, day 3 (inflammation but no fibrosis), day 7 (inflammation and low fibrosis) and day 14 (high fibrosis), as described in Pavkovic et al.³⁸. First, we validated in this dataset that fibrosis increased progressively from day 3 to day 14, by computing an

ECM-score corresponding to an average expression of matrix associated genes. Second, we deconvoluted this bulk RNAseq dataset by using a comprehensive atlas based on single cell RNAseq data composed of 49 226 cells. Thanks to this deconvolution, we observed that inflammatory fibroblasts expand early after injury (day 3 to day 7), before decreasing when high degree of fibrosis concomitantly appears (day 14) in this mouse model. Reciprocally, ECM-secreting myofibroblasts are only detected after day 7 and their content peaks at day 14, as expected based on our observations in CKD patients. These new data validate the dynamic evolution of inflammatory fibroblasts during fibrosis development in CKD mouse model and strengthen the idea of inflammatory fibroblasts being an early and intermediate state in the differentiation process toward ECM-secreting myofibroblasts. These new data are now presented in **Fig. 2d, f-i** and described **p7-8** of the new version of the manuscript.

p7-8: Results: “To validate the temporal dynamics of inflammatory fibroblasts and ECM-secreting myofibroblasts, we took advantage of publicly available bulk RNAseq data from mice undergoing unilateral ureteral obstruction (UUO) at different time points³⁸. UUO is a classical and well characterized model of kidney fibrosis in mice, and we analyzed samples from control mice, and from mice 3 days after UUO (early time point with infiltration but no fibrosis), 7 days after UUO (intermediate level of fibrosis) and 14 days after UUO (with a high level of interstitial fibrosis). First, we calculated an ECM score characterized by the expression of ECM-related genes as described in Naba et al.³⁹, which allowed us to validate that this score increased progressively from day 0 to day 14 after UUO (**Fig. 2d**), and confirm the progressive increase in interstitial fibrosis. Then, to calculate the proportion of each cell type according to each time point, we performed a deconvolution of the bulk RNAseq data using BayesPrism⁴⁰. To do so, we first built a comprehensive cellular atlas (**Fig. 2e**) based on scRNAseq dataset from CKD and normal kidney tissues⁸. This cellular atlas was composed of 49 226 cells corresponding to 9 different cell types (**Fig. 2e**). Interestingly, deconvolution results showed that kidneys from control mice were composed of mainly proximal tubular epithelial cells (around 80%), whereas after injury the proportion of proximal tubular cells decreased (**Fig. 2f**). This result validated our approach, as it is largely known that the cellular composition of a normal mouse kidney is approximately 70-80% of proximal tubular epithelial cells, and that this proportion significantly decreases after injury⁴¹. In addition, we observed a progressive increase in the proportion of mesenchymal cells, as well as of the immune compartment and injured tubules (**Fig. 2g**). To have more insights into the mesenchymal compartment, we focused our next analysis specifically on these cells. While pericytes were not detected in high proportion, we observed a gradual switch from inflammatory fibroblasts to ECM-secreting myofibroblasts from day 3 to day 14 (**Fig. 2g**). Interestingly, inflammatory fibroblasts were virtually absent at day 14 (**Fig. 2g**). Finally, we confirmed our observations from pseudotime analysis in patients (**Fig. 2a-c**) as we saw inflammatory fibroblasts arising early after injury (day 3), being maintained at intermediate stages (day 7), before disappearing at advanced stages of interstitial fibrosis (day 14) (**Fig. 2h**), meanwhile the proportion of ECM-secreting myofibroblasts increased progressively up to day 14 (**Fig. 2i**). Altogether, these data identify a population of inflammatory fibroblasts, which arises early after kidney injury, before decreasing at late stages when ECM-secreting myofibroblasts expand.”

p33: Legend Fig. 2d, f-i: “(d) ECM-score calculated on the kidney bulk RNAseq data from mice undergoing UUO at different time points (n = 15 mice) (f) Barplot showing the results of the deconvolution of bulk RNAseq data from UUO mouse model analyzing 7 different cell types. (g) Same as (f) but considering exclusively mesenchymal cells. (h) Barplot showing the proportion of inflammatory fibroblasts estimated by deconvolution UUO at different time points relative to control mice. (i) same as (h) for ECM-secreting myofibroblasts.”

In addition to the validation of the human data in a pre-clinical relevant mouse model of kidney fibrosis, we agree that developing an *in vivo* model might be helpful to validate the mechanisms we highlighted in the setting of CKD. Still, although interesting, clearing CXCL-iFibro at early stages of CKD *in vivo* is highly challenging. As the content in this population increases after kidney injury, we need to use CAR T cells or Diphtheria Toxin to deplete it by using a specific surface marker (which might be different in human and mouse) leading to the total clearing of CXCL-iFibro at early stages of CKD in mice. We are currently working on the identification of common markers or mediators in human and mouse CXCL-iFibro and macrophages, that are required to setup the depletion experiment in mice. Thus, although interesting, developing such a model doesn't seem possible in the imparted time, and we consider developing it in a follow-up study. As recommended by the Reviewer, and in addition to the

experiments performed on mouse models described below, we have now softened our conclusions regarding the bidirectional loop, in the Abstract **p2**, as well as in the Discussion **p17**, of the new version of the manuscript:

p2: Abstract: “We demonstrate that CXCL-iFibro co-localize with macrophages in the kidney and participate in their attraction, accumulation, and switch into FOLR2+ macrophages from early CKD stages. In turn, *in vitro*, macrophages promote the switch of CXCL-iFibro into ECM-secreting myofibroblasts through a WNT/b-catenin-dependent pathway, thereby suggesting a reciprocal crosstalk between these populations of fibroblasts and macrophages”.

p4: Introduction: “By combining single cell analysis, spatial transcriptomics and *in vitro* functional assays using primary fibroblasts isolated from patients, we provide some clues suggesting that these pro-inflammatory CXCL-iFibro fibroblasts might attract and activate FOLR2+ macrophages.”

p17: Discussion: “Consistent with the accumulation of the CXCL-iFibro population at early stages of kidney disease, we show its probable function in the progression of the pathology by highlighting its reciprocal crosstalk with FOLR2+ macrophages.....Finally, we show that detecting CXCL-iFibro or FOLR2 at early stage of kidney disease in a large cohort of CKD patients is predictive of poor patient outcome. This confirms that the CXCL-iFibro population might be an early player in CKD progression and demonstrates the clinical relevance of our findings in CKD.”

Reviewer #2 (Remarks to the Author):

In this manuscript, Cohen et al. highlighted an intermediate fibroblast population CXCL-iFibro with pro-inflammatory properties which give rise to ECM-secreting myofibroblasts and attracted CD14+ monocytes and polarize them into FOLR2+ macrophages. In turn, FOLR2+ macrophages induce the transition of CXCL-iFibro into ECM-secreting myofibroblasts through WNT/ β -catenin signaling. The CXCL-iFibro correlates with poor prognosis in CKD patients, which provides a prognostic marker as well as a potential therapeutic target to limit CKD progression. Although the observation is interesting, several fundamental issues need to be addressed.

We thank the Reviewer for the positive evaluation of our work. We have now addressed all concerns raised, which improved the quality of our manuscript.

Major comments:

1. CXCL-iFibro highly expressed CXCL12 has been reported in CDK (Kuppe et al., 2021). Therefore, it is not new in kidney fibroblast population identified in this manuscript.

We agree with the Reviewer that CXCL12-expressing fibroblasts have been reported in Kuppe et al. and we have now softened our conclusions in the new version of our manuscript, in particular in the Abstract **p2**, as well as at the end of the introduction section **p4**.

p2: Abstract: “Here, we characterize in CKD a pro-inflammatory fibroblast population (named CXCL-iFibro), which corresponds to an early state of myofibroblast differentiation.”

p4: Introduction: “More precisely, we focused on a population of inflammatory fibroblasts (referred to as CXCL-iFibro), which represents an intermediate state in the differentiation process toward ECM protein-secreting myofibroblasts”. Nevertheless, by comparing the fibroblasts from CKD patients with a dataset of cancer-associated fibroblasts, we identified a functionally relevant fibroblast population, which has not yet been characterized, even in the recent publication from Kuppe et al. 2021. To illustrate our point, we sought to identify the similarities between our fibroblast annotations and those from Kuppe et al 2021. In that aim, we performed a label transfer between their dataset and our own. Interestingly, we found that the “pericyte-like” populations that we identified were very close to the “Pericytes” and “vascular smooth muscle cells” that Kuppe et al identified. The Detox-iFibro we found correspond to the population Kuppe defined as fibroblast 1 (SCARA5+ MEG3+) population, and the TGF β -myoFibro correspond to myofibroblast 1, the population characterized with the highest ECM score. In contrast, the CXCL-iFibro population we identified as a homogenous population based on the correspondence with CAF was dispersed in several clusters in the Kuppe et al. study (Fibroblast 2a (30.5% of cells), Myofibroblast 3a (22.1%), Myofibroblast 3b (17.2%), Fibroblast 2b (14.9%) and Myofibroblast 2b (12.3%)). Thus, annotating the cells regarding their role in cancer helped us to identify this functionally relevant population of CXCL-iFibro. We think that these new data help in understanding the

identification and the annotations of the different mesenchymal populations. We thank the Reviewer for this comment. The method is now described in the Methods section, **p23**, and the data are now presented in new **Fig. S1d-e** and **Table S3** and described **p6-7** in the new version of our manuscript.

p6-7: Results: “As Kuppe et al. described several clusters of mesenchymal cells using PDGFR β + sorted cells⁸, we sought to identify the similarities between our fibroblast annotations and those from this study (as reported in **Fig. S1d**). Hence, we performed a label transfer analysis on PDGFR β + sorted cells from⁸. The dataset of 2495 mesenchymal cells with our own annotations (described in **Fig. 1j**) was used as a reference, and the dataset of PDGFR β + sorted cells (described in **Fig. S1d**), with original annotations from⁸ was used as a query. Interestingly, the pericyte-like populations that we identified (IFNab-Peri-like and Contractile-Peri-like) showed a high degree of similarity with pericytes and vascular smooth muscle cells described by Kuppe et al (**Fig. S1e** and **Table S3**). Similarly, the Detox-iFibro we described corresponded to the Fibroblast 1 population that Kuppe described as being a SCARA5+ MEG3+ non-activated fibroblasts, and the TGF β -myoFibro was similar to the Myofibroblast 1, the population exhibiting the highest level of ECM protein secretion. In contrast, CXCL-iFibro, and to a lesser extent Wound-myofibro were identified as a mix of different fibroblast and/or myofibroblast populations, according to original annotations. In particular, the CXCL-iFibro population that we identified as being a homogenous population was dispersed throughout several clusters in the Kuppe et al. study and characterized as a mix of Fibroblast 2a (30.5% of cells), Myofibroblast 3a (22.1%), Myofibroblast 3b (17.2%), Fibroblast 2b (14.9%) and Myofibroblast 2b (12.3%) (**Fig. S1e** and **Table S3**). Thus, annotating mesenchymal cells based on similarities with CAF allowed us to identify a population of fibroblasts with inflammatory properties that has not been explored further previously.”

p23: Methods: “*Label transfer:* To identify similarities between the original annotation of cells from the PDGFR β + sorted cells single-cell RNAseq dataset from⁸ and our annotations, we used the Label Transfer algorithm using the *FindTransferAnchors* and the *TransferData* functions from Seurat. The dataset of 2495 mesenchymal cells with our own annotation described in **Fig. 1j** was used as a reference, and the dataset of PDGFR β + sorted cells, described in **Fig. S1d**, with original annotations from⁸ was used as a query. The alluvial plot was then generated using the *do_AlluvialPlot* function from SCpubr package, using the original annotations as the first group, and the predicted Id in the final group.”

2. *The specificity of SFRP1 to mark CXCL-iFibr should be demonstrated. SFRP1 was used to mark CXCL-iFibr, while SFRP1 was also expressed in Detox-iFibro. The IHC of SFRP1 showed that many kidney cells were stained by SFRP1 including kidney tubule cells. This may raise a non-specificity concern (Fig 2d), which should be addressed and verified by more specific methods.*

We agree with the reviewer that some cells from the Detox-iFibro cluster (initially referred to as cluster 0) expressed SFRP1, even if the proportion of these cells is lower in Detox-iCAF than in CXCL-iFibro (please, see below **Supplementary Fig.1 for the Reviewer**). Still, we want to emphasize that the majority of SFRP1+ cells are CXCL-iFibro. Thus, most of SFRP1+ cells correspond to CXCL-iFibro cells, which validates SFRP1 as the most appropriate marker for CXCL-iFibro identification. Nevertheless, to follow the Reviewer’s advice and be more precise in our description, we have now provided precision in the description of SFRP1 by indicating that it is significantly differentially expressed in CXCL-iFibro (cluster 0), but that few cells from Detox-iCAF (cluster 4) also expressed it. As requested, we modified our description of Detox-iFibro (cluster 4) as SFRP1+/- SFRP4+ FAP- RAMP1- in the new version of our manuscript (**p6**).

p6: Results: “To define specific markers of these clusters, we performed a pairwise analysis of the genes differentially expressed between the different clusters and we identified SFRP1 (Secreted frizzled Related Protein 1) as significantly up-regulated in cluster 0 (although few cells of clusters 3 and 4 also expressed SFRP1), FAP (Fibroblast Activation Protein) in both clusters 3 and 5, SFRP4 (Secreted frizzled Related Protein 4) in clusters 3 and 4 and RAMP1 (Receptor Activity Modifying Protein 1) in cluster 5 (**Fig. 1i** and **Fig. S1c**). By this way, cluster 0 could be identified as SFRP1+ SFRP4- FAP- RAMP1-; cluster 4 as SFRP1+/- SFRP4+ FAP- RAMP1-; cluster 3 as SFRP1+/- SFRP4+ FAP+ RAMP1- and cluster 5 as SFRP1- SFRP4- FAP+ RAMP1+.”

Supplementary Fig.1 for the Reviewer: Violin plot showing the expression of SFRP1 in the different cluster identified in 2495 mesenchymal cells from Kuppe et al.

To address the specificity concern of SFRP1 staining in stromal cells, we now show more images (**Fig. S2a**) to provide additive evidence of SFRP1 staining in the stroma of kidneys from patients with fibrosis. We also performed control staining without primary antibody, showing that the staining is not the result of a non-specific trapping of the secondary antibody in fibrotic areas. As indicated by the Reviewer, we observed that some tubular cells are positive for SFRP1 staining. To address the Reviewer's concern, we quantified the double positivity of SFRP1+ cells and α SMA+ cells. By this way, we validated that the majority of SFRP1+ cells are indeed fibroblasts. These data are now shown in **Fig. S2c (p8)**. We also would like to emphasize that some tubular cells can experience partial EMT after kidney injury and express mesenchymal markers (such as Vimentin or α SMA^{42, 43}). It is therefore possible that some tubular cells express some mesenchymal markers, including SFRP1 in fibrotic samples. This has been added in the manuscript, **p8**.

p8: Results: “Co-staining experiments between SFRP1 or FAP and α SMA showed that SFRP1+ and FAP+ cells were mainly α SMA+, consistent with them being activated fibroblasts (**Fig. S2c, d**). We also observed that some tubular cells could also be positive for SFRP1 (**Fig. 2j** and **Fig. 2l**). Consistent with this, it is known that, during CKD, some tubular cells can express some mesenchymal markers (such as VIM or α SMA) and undergo partial epithelial to mesenchymal transition^{42, 43}.”

p40: Legend of Fig. S2a-d: “(a) Additive representative images (in addition to those shown in **Fig. 2j, k**) to emphasize the stromal staining of SFRP1+ at low (upper panel) and high (lower panel) magnification. Left panel shows a representative image of the complete staining. The right panel shows the result of the staining using the same protocol without primary antibody, highlighting the specificity of the primary antibody. Scale bars = 1mm for the upper panel, 50mm for the middle panel, 20 mm for the lower panel. (b) Same as (a) but for FAP. Scale bars = 1mm for the upper panel, 50mm for the middle panel, 20 mm for the lower panel. (c) Representative images (left panel) and quantification (right panel) of IF showing co-staining between SFRP1 (red) and α SMA (green). Quantification shows the percentage of SFRP1+ α SMA+ double positive interstitial cells. N = 6 PKD patients. Low panels of images show the image resulting from the same protocol without primary antibody. (d) Same as in (c) but for FAP and α SMA co-staining.”

3. Co-staining of CD68 and SFRP1 is mainly in the same cell rather in two adjacent individual cells (Fig 3a, 3b). Does CD68 co-express with SFRP1 in myeloid cells (in RNA-seq data)?

Expression of SFRP1 in our single cell atlas is predominant in inflammatory fibroblasts (**Supplementary Fig. 2 for Reviewer**). A discrete expression is observed in the thick ascending limb and the thin descending limb of the Henle loop, possibly explaining the tubular staining we sometimes observe by IHC. However, we do not observe any SFRP1 expression in myeloid cells, including CD14+ or CD16+ monocytes, resident, FOLR2+ or TREM2+ macrophages.

Supplementary Fig.2 for the Reviewer: Violin plot showing the expression of SFRP1 in the CXCL-iFibro and the different myeloid cells cluster identified.

4. *Quantification results of average MFI (mean fluorescent intensity) were based on about 30 cells per experimental condition, which is too small to reach the conclusion and needs to increase the sample size. In addition, larger-viewing area with more cells for IF staining should be provided (Fig 4a, 4b, 5a, 5f, 5g, 5i).*

We thank the reviewer for highlighting this concern. As requested, we enhanced the robustness of our analysis by increasing the number of cells analysed to more than 100 in each condition. New quantifications are now provided in the corresponding legends of **Fig. 4 and 5**. We also modified the images consequently and provide now larger viewing areas to better illustrate our quantifications in the **new Fig. 4 and new Fig. 5**.

5. *Cell-cell interaction analysis showed that the highest number of interactions and the highest interaction strength of CXCL-iFibro was with the CD14+ monocytes rather than FOLR2+ macrophage. However, the colocalization results in figure 3 showed that CXCL-iFibro colocalized with FOLR2+ macrophage. This seems contradictory and should be verified.*

As mentioned by the Reviewer, we indeed observed that the highest number and strength of interactions were between CXCL-iFibro and CD14+ monocytes. This result drove the design of our co-culture experiments between primary fibroblasts and CD14+ PBMC. In contrast of being contradictory, this result fits with our hypothesis -supported by our data- that CXCL-iFibro expand early during CKD progression, attract CD14+ monocytes, which transitioned into FOLR2+ macrophages. We also would like to insist on the fact that CellChat analysis integrates secreted molecules, meaning that spatial proximity is not requested to infer cellular interactions. As recommended by the Reviewer, we have now clarified our hypothesis in the new version of the manuscript, **p12-13**.

p12-13: Results: “Because we observed a close vicinity between CXCL-iFibro and FOLR2+ macrophages (**Fig. 3h, i**), we hypothesized that CXCL-iFibro could attract CD14+ monocytes and modify their phenotype into a FOLR2+ macrophage phenotype, hypothesis consistent with the preferential interaction detected *in-silico* between CXCL-iFibro and CD14+ monocytes. To test this hypothesis, we designed co-culture experiments between collagen-cultured primary fibroblasts (CXCL-iFibro) and CD14+ monocytes isolated from peripheral blood mononuclear cells (PBMC) from healthy donors, to assess if CXCL-iFibro could indeed attract CD14+ monocytes and potentially promote their differentiation into a FOLR2+ macrophage phenotype.”

6. *The collagen production is a marker of fibrogenic phenotype and should be analyzed to confirm the phenotype from CXL-iFibro to ECM-secreting myoFibro.*

We thank the Reviewer for mentioning this notion. We have now considered this concern and performed RNA sequencing of patient-derived fibroblasts maintained in culture by using different conditions either on collagen- (n=3) or on plastic-dishes (n=2). We identified several ECM-related genes, as described in Nab et al. ³⁹, that are upregulated in the plastic-cultured cells, such as some collagens, integrins and laminins. These new data are now showed in **Fig. S5c, d**, and in the manuscript **p12**.

p12: Results: “We then performed RNA sequencing on collagen- or plastic-cultured fibroblasts to confirm the respective identity of these cells. Consistent with the inflammatory identity detected in CXCL-iFibro, we identified several inflammatory genes, including *CXCL12*, *IL1B* and *IL34*, upregulated in collagen-cultured compared to plastic-cultured cells (**Fig. S5a, b**). These proteins are of interest, because of their role in chemoattraction of immune cells (*CXCL12*)⁵⁸, pro-inflammatory response (*IL1B*)^{59, 60} or in promoting the differentiation and viability of monocytes and macrophages through the colony-stimulating factor-1 receptor (*IL34*)⁶¹. Similarly, consistent with their ECM-secreting myoFibro identity, plastic- cultured fibroblasts exhibited increased expression of ECM-related genes, such as *COL4A1* and *COL4A5* (**Fig. S5d**), which are known to be important in the fibrotic process and basement membrane integrity⁶². We also identified other components of the ECM, belonging to the integrin (*IGFBP1*) or the laminin (*LAMA3*) families (**Fig. S5d**). Altogether, these data suggest that collagen-cultured and plastic-cultured fibroblasts are reminiscent of CXCL-iFibro and ECM-secreting myoFibro, respectively.”

7. Macrophage lineage-tracing method is a more reliable method for tracing FOLR2+ macrophages in chronic kidney disease and should be considered.

We agree with the reviewer that lineage tracing would be a very interesting experience to perform, to identify the origins of FOLR2+ macrophages during CKD progression. But, this is a very challenging experiment and we hope the Reviewer will agree that we provide here a substantial number of *in vitro* functional assays and *in vivo* data from patients. Thus, lineage-tracing in CKD mouse models could be considered as beyond the scope of our study, integrated in a follow-up study.

Minor comments:

1. Line 116: Only 6 clusters of 13 mesenchymal cells clusters were used in this study. The features of the other 7 clusters of cells should be provided and explain why these cells were discarded.

We thank the Reviewer for this mention. We decided to focus on the first 6 clusters of cells (clusters 0 to 5) to simplify our message, as they represented more than 85% of the total number of cells. Clusters 6 to 12 contain less than 80 cells per cluster, and we did not want to overstate on such rare populations of cells. We have now explained our approach in more details in the new version of the manuscript, **p5**.
p5: Results: “Unsupervised graph-based clustering identified 13 clusters, visualized with the Uniform Manifold Approximation and Projection (UMAP) algorithm (**Fig. S1a**). To simplify our message and increase the relevance of our comparisons, we focused on the first 6 clusters, as they represented more than 85% of the total mesenchymal population and the other clusters contained less than 80 cells per cluster (**Fig. 1a** and **Fig. S1b**).”

2. Cell in cluster 5 seems mainly from P9 (Fig. 1a, b).

We do agree that cells in cluster 5 are mainly from patient P9 (accounting for 20% of the cells from this patient with 152 cells), but Patient P10 also presented 32 cells in cluster 5. Please find below a Table (**Table 1 for Reviewer**) indicating the numbers of cells, and the corresponding percentages of cells per cluster in each patient. To answer to Reviewer’s concern, we have now brought this information in **Fig. 1b right**, where we show the proportion of cells for the 6 first clusters in each patient, and in **Fig. S1b** for the 13 clusters. We modified the text consequently, in the Results section **p5**, as well as in the legends of Fig. 1 and Fig. S1, **p34** and **p40**.

p5: Results: “All clusters were found in at least 2 patients, albeit in different proportions from one patient to the other”.

p34: Legend of Fig. 1b: (b) Left, Same UMAP as in (a) showing cell repartition across patients (P1 to P12). Right, Barplot representing the proportion of cells in the first 6 clusters (clusters 0 to 5) in each patient (P1 to p12).

p40: Legend of Fig. S1b: (b) Bar plot showing the percentages of cells in the different clusters (0 to 12) per patient.

	P1	P2	P3	P4	P5	P6	P7	P8	P9	P10	P11	P12
0	3 (1.4%)	98 (39.7%)	15 (24.6%)	3 (37.5%)	8 (6.6%)	0 (0%)	38 (35.8%)	8 (30.8%)	263 (35.7%)	343 (39.7%)	0 (0%)	1 (14.3%)
1	101 (47%)	93 (37.7%)	39 (63.9%)	3 (37.5%)	70 (57.9%)	25 (49%)	27 (25.5%)	12 (46.2%)	58 (7.9%)	75 (8.7%)	13 (24.5%)	4 (57.1%)
2	106 (49.3%)	29 (11.7%)	4 (6.6%)	1 (12.5%)	39 (32.2%)	25 (49%)	13 (12.3%)	5 (19.2%)	115 (15.6%)	14 (1.6%)	29 (54.7%)	2 (28.6%)
3	2 (0.9%)	19 (7.7%)	0 (0%)	1 (12.5%)	4 (3.3%)	1 (2%)	3 (2.8%)	1 (3.8%)	62 (8.4%)	257 (29.7%)	5 (9.4%)	0 (0%)
4	2 (0.9%)	5 (2%)	2 (3.3%)	0 (0%)	0 (0%)	0 (0%)	17 (16%)	0 (0%)	86 (11.7%)	143 (16.6%)	6 (11.3%)	0 (0%)
5	1 (0.5%)	3 (1.2%)	1 (1.6%)	0 (0%)	0 (0%)	0 (0%)	8 (7.5%)	0 (0%)	152 (20.7%)	32 (3.7%)	0 (0%)	0 (0%)

Table 1 for Reviewer: Number of cells in each Patient, P1 to P12 (and corresponding % of cells per Patient) in the first 6 mesenchymal clusters (clusters 0-5)

3. Gene symbols should be in **italic**, such as figure 1e, 1h, S1c, etc.

We thank the Reviewer for the careful reading of our paper. As requested, we modified the **Figures**, accordingly.

4. The scale bars of color are missing in Fig. 1e, 1g, 1h, etc.

Here again, we thank the Reviewer for the careful reading of our paper. As requested, we modified the **Figures**, accordingly.

5. Several terms of functional enrichment for each cluster were listed in Table S1. Whether these terms were selective or top-rank.

These terms were top-ranked. As recommended, we added this information in the Results' section **p5**, as well as in the Methods section **p23**.

p5: Results: "Although all the fibroblastic clusters were associated with extracellular matrix (ECM) remodeling pathways, each cluster was associated with specific processes , among top-ranked pathways in each cluster (**Table S1**)."

p23: Methods: "The pathways listed in **Table S1** were top-ranked."

6. The gene lists for scoring CAF gene signatures should be provided (Fig. 1f, g).

These gene list for CAF-S1 have already been published by our lab in a previous publications (Kieffer et al, *Cancer Discovery*, 2020²⁹). We now provide CAF-S4 gene list in **Table S2**. We are sorry for having been unclear and have now provided more precise information in the Results, **p5-6**, in the Methods section, **p23**, and in the legend of the new Fig. 1g, **p34**.

p5-6: Results: "Indeed, the CAF-S1-specific gene signature ²⁹, which identifies populations of inflammatory and myofibroblastic FAP+ CAF associated with metastatic spread and immunosuppression ^{15, 16, 18, 19, 29} was highly detected in the fibroblastic clusters (clusters 0, 3, 4, 5) in the kidney disease dataset, while the CAF-S4 signature (perivascular-like CAF, gene signature in **Table S2**) highlighted the pericyte-like cells (clusters 1, 2) (**Fig. 1g**)."

p23: Methods: "Gene Signatures analysis of CAF-S1, CAF-S4 and CAF-S1 Clusters: Specific gene signatures were previously published in ²⁹ for CAF-S1 and CAF-S1 clusters. CAF-S4 gene signature is listed in **Table S2**. Z-score average was calculated for each signature, and then plotted on the generated UMAP using *FeaturePlot* function from Seurat."

p34: Legend of Fig. 1g: “(g) UMAP (top) and Violin plot (bottom) showing the average z-score of genes that compose specific signatures of CAF-S1²⁹ and CAF-S4 (Table S2).”

7. Line 155: Authors defined cluster 0 as SFRP1+ SFRP4- FAP- RAMP1- (CXCL-iFibro) and cluster 4 as SFRP1- SFRP4+ FAP- RAMP1- (Detox-iFibro), while cluster 4 have a certain expression of SFRP1 and CXCL12.

As discussed above, we agree with the Reviewer that some cells from the Detox-iFibro cluster also expressed SFRP1, even if the proportion of these cells is lower in Detox-iCAF than in CXCL-iFibro (Supplementary Fig. 1. for the Reviewer). Still, we want to emphasize that most of SFRP1+ cells correspond to CXCL-iFibro cells, which validates SFRP1 as the most appropriate marker for CXCL-iFibro identification. Nevertheless, to follow the Reviewer’s advice and be more precise in our description, we modified our description of Detox-iFibro (cluster 4) as SFRP1+/- SFRP4+ FAP- RAMP1- in the new version of our manuscript (p6).

p6: Results: “To define specific markers of these clusters, we performed a pairwise analysis of the genes differentially expressed between the different clusters and we identified SFRP1 (Secreted frizzled Related Protein 1) as significantly up-regulated in cluster 0 (although few cells of clusters 3 and 4 also expressed SFRP1), FAP (Fibroblast Activation Protein) in both clusters 3 and 5, SFRP4 (Secreted frizzled Related Protein 4) in clusters 3 and 4 and RAMP1 (Receptor Activity Modifying Protein 1) in cluster 5 (Fig. 1i and Fig. S1c). By this way, cluster 0 could be identified as SFRP1+ SFRP4- FAP- RAMP1-; cluster 4 as SFRP1+/- SFRP4+ FAP- RAMP1-; cluster 3 as SFRP1+/- SFRP4+ FAP+ RAMP1- and cluster 5 as SFRP1- SFRP4- FAP+ RAMP1+.”

8. The labels of the axis are missing in Fig. 2a, 2b, S3d-g, 6a, 6b, etc.

We thank the Reviewer for the careful reading of our paper. As recommended, we modified the Figures, accordingly.

9. Line 171: The origin of myofibroblasts is still controversial. How to confirm that other cell types (such as epithelia, endothelia, etc.) are not the sources of myofibroblasts?

We agree with the Reviewer that several cell types could be the source of myofibroblasts during kidney disease. Nevertheless, there are already published evidence showing that major sources of ECM secreting myofibroblasts are pericytes and resident fibroblasts in kidney^{5,8}. Lineage tracing experiments in mice confirm these findings^{3,4,5,6,7}. We thus defined pericytes-like mesenchymal cells as the root of the trajectory, because of evidence in the literature for that statement, which were not demonstrated by lineage tracing for other suspected origins in kidney fibrosis. Still, we considered the Reviewer’s comment, and we softened our message in the manuscript, p7.

p7: Results: “Despite a longstanding debate regarding the origins of myofibroblasts in the kidney, several reports identified pericytes, as a major source of myofibroblasts in kidney fibrosis development^{3,4,5,6,7,8}.”

10. The IHC of fibrosis-enriched zones on kidney biopsy samples should be provided (Fig 2f, 2g).

We added some images showing field with fibrosis by Masson’s trichrome staining in the new (Fig. 2l, m), middle panels.

11. Line 244: TREM2+ and FOLR2+ macrophages have been described in diabetic kidney disease (Fu et al., 2022).

We apologize for this and have now added this reference in the new version of our manuscript, p11 and p17.

p11: Results: “As these different subtypes of FOLR2+ and TREM2+ macrophages have been poorly described but very recently detected in diabetic kidney disease⁵³, we tested whether”

p17: Discussion: “...as CXCL-iFibro are able to attract CD14+ monocytes and to polarize them into FOLR2+ macrophages, which have been recently described in cancer and diabetic kidney^{49,50,51,53}.”

12. Line 264: A subset of cells in Cluster 1 highly expressed dendritic cell markers (CD1C, CD1E) (Fig. S3f). It inaccurately annotated all cells in cluster 1 as TREM2+ macrophages.

Our unsupervised clustering indeed identified this cluster composed of TREM2+ macrophages, as well as dendritic cells expressing CD1C and CD1E. We do agree with the Reviewer and modified the annotation of this cluster accordingly. As recommended by the Reviewer, this cluster is now annotated TREM2+ macrophages / DC in **Fig. 3** and in **Fig. S3** and described in the new text **p10**.

p10: Results: “Finally, cluster 1 was composed of TREM2+ macrophages, and from dendritic cells (DC) positive for MRC1, CD1E and CD1C expression (**Fig. S3f**). We thus annotated this cluster 1 as TREM2+ macrophages / DC (**Fig. 3d, f** and **Fig. S3g**).”

13. The distributions of myeloid cell types in each patient should be provided (Fig. 3d-f).

As requested, we have now provided the distributions of each myeloid cluster in the new version of the manuscript in the new **Fig. 3g** described **p10**, and its corresponding legend **p35**.

p10: Results: “These clusters were represented, albeit in different proportions, in several patients (**Fig. 3g**).”

p35: Legend of Fig. 3g: “Barplot showing the proportion of each myeloid cell cluster in each patient. Myeloid cell data were available for 10 patients.”

14. Why the correlation between the number of FOLR2+ cells and fibrosis in PKD samples is opposite in fibrotic kidney biopsies? Since a strong correlation between the number of SFRP1+ CXCL-iFibro and FOLR2+ macrophages, the correlation between the number of FOLR2+ cells and fibrosis in fibrotic kidney biopsies should be similar to the trend in Fig. 2f (Fig3g, 3h).

We thank the Reviewer for highlighting this point. Indeed, FOLR2+ and CXCL-iFibro are highly correlated (both at the protein and RNA levels). To address this concern, we have now increased the number of patients with high fibrosis analyzed for FOLR2+ macrophage content. In particular, we studied the same patients for both SFRP1 and FOLR2 staining to provide a better comparison between the two quantifications, and we could reproduce a similar trend to that observed with SFRP1, as expected. We thank the Reviewer for mentioning this point. This data is now shown in **Fig. 3i**.

15. IFN $\alpha\beta$ -peri like, wound-myofibro and TGF β -myofibro colocalize together with endothelium and FOLR2+-CKD macrophages in Fig. 4c which is inconsistent with the result in Fig. 3l. Also, Detox-iFibro colocalize with FOLR2+-CKD macrophages (Fig. S4c).

It is true that after deconvolution and non-negative matrix factorization, some FOLR2+ CKD macrophages were identified in the same spots as endothelium, IFN $\alpha\beta$ -Pericytes-like cells, as well as wound- and TGF β -myofibro (factor 10 for Patient 2 in **Fig. S4c** and to a lesser extent factor 5 for Patient 1 in **Fig. 3l**). Nevertheless, most co-localization for FOLR2+ CKD macrophages occurred with CXCL-iFibro in both patients (factor 11 and 6, for Patients 1 and 2, respectively). Thus, these results from deconvolution output of spatial transcriptomic data are consistent with our other data from patients and functional assays, somehow reinforcing the validity of our observations. Still, to be totally transparent with the reader, we modified the manuscript **p11**:

p11, Results: “Strikingly, we observed that FOLR2+-CKD macrophages mainly colocalized with CXCL-iFibro and plasma cells (**Fig. 3l, m** and **Fig. S4b, c**). “

16. The gating thresholds of flow cytometry analysis are not identical (Fig. 4i).

We thank the reviewer for highlighting this error. We have now corrected it, as shown in new **Fig. 4i**.

17. Whether TCF4 and target genes of TCF4 are upregulated following the trajectory of CXCL-iFibro to ECM-secreting myoFibro transition.

TCF4 was expressed mainly in cluster 3 and 5 (ECM-secreting myofibroblasts), as now shown in the new **Fig. S6c**. On the same hand, we evaluated the expression of *TCF4* target genes, by looking at the expression of the signature of *TCF4* target genes defined by Dorothea regulons. We observed that expression of *TCF4*-target genes was increased in the cluster 3 (Wound myoFibro), as now shown in **Fig. S6e**. Furthermore, we tested the expression of *TCF4* according to the pseudotime defined in **Fig. 2** and identified that *TCF4* expression increased progressively along the differentiation from CXCL-iFibro to ECM-secreting myofibroblast process (new **Fig. S6d**). As requested, these new data are now shown in (**Fig. S6c, d**) and described **p14**.

p14: Results: “Moreover, *TCF4* expression was more prominent in ECM-secreting myoFibro (Fig. S6c). *TCF4* expression along the pseudotime defined in Fig. 2b showed that its expression increased late in the differentiation, in ECM-secreting myoFibroblasts (Fig. S6d). Finally, *TCF4* target gene expression was also increased in Wound-myofibro, strengthening the idea that *TCF4* could be instrumental in the transition from CXCL-iFibro to ECM-secreting myoFibro (Fig. S6e).”

18. Line 378: The citation of the figure should be Fig. 5i.

We thank the Reviewer for the careful reading of our paper, and we corrected the text accordingly.

Reviewer #3 (Remarks to the Author):

This manuscript by Cohen et al investigates the role of inflammatory fibroblasts in the progression of chronic kidney disease. Drawing on their background in cancer biology, the group mines publically available scRNAseq dataset to define an inflammatory subset of fibroblasts termed CXCL-iFibro. Pseudotime techniques are used to show that CXCL-iFibro accumulate early in CKD whereas ECM-myofibro accumulate later. These CXCL-iFibro, characterized by SFRP1, are shown to be in areas near macrophages and to be expressing pro-inflammatory cytokines. The investigators use primary human fibroblasts and CD14+ monocytes to show that SFRP1+ fibroblasts can increase monocyte migration and expression of FOLR2 and that these CD14+ monocytes can promote SFRP1 fibroblasts transition to myofibroblasts. The authors use propose that this myofibroblast transition is mediated through Wnt/beta-catenin signaling. Furthermore, they show that patients with biopsies with “high CXCL-iFibro” have a worse clinical prognosis from bulk RNAseq data in the Neptune registry.

These studies shed light on a potentially important subset of stromal cells with inflammatory characteristics. However, some important concerns persist about these stromal/fibroblast populations and the role of Wnt/beta-catenin signaling.

We thank the Reviewer for the positive evaluation of our work, and for highlighting that we identify an important subset of stromal cells as a key player in CKD. We have now addressed the different points raised.

Major:

1. There are many lessons that the field of fibrosis can learn from cancer biology. However, the nomenclature (e.g. CAF-S1, detox-iCAF, etc) used for cancer associated fibroblasts is a bit confusing as applied to kidney stromal cells. Perhaps this could be simplified in Introduction and some information about how the CXCL-iFibro and myoFibro populations compare to the two key classes of myofibroblasts identified in the Nature 2021 (Kuppe) paper from which the scRNAseq data is used: the fibroblast 1 (Scara5, Meg3+) non-activated population and the myofibroblasts (and associated subsets). They also describe a myofibroblast 2 subset (OGN+COL14A1+) that is postulated to be an intermediate state between pericytes and myofibroblasts. It is critical to know whether the populations described in this manuscript are distinct from those already published or similar but with different nomenclature. The presence/absence of markers that have more meaning to the renal fibrosis field (e.g. PDGRb, osteopontin, NOTCH, etc) should be mentioned at some point either in Results or supplemental data.

We agree with the reviewer that applying knowledge from the cancer biology field to the fibrosis field is promising but challenging. As requested by the Reviewer, we tried to make our message simpler, and even if CAF heterogeneity is becoming more and more complex, we have simplified the introduction section concerning CAF populations in the new version of the manuscript, p3.

p3: Introduction: “In cancer, the molecular heterogeneity and functional diversity of fibroblasts (referred to as Cancer-Associated Fibroblasts, or CAF) have been highlighted in recent years. Indeed, several CAF populations have been identified in different cancer types by combining the study of several CAF markers, such as Fibroblast Activation Protein (FAP), Smooth Muscle- α Actin (SMA) and Regulator of G protein signaling 5 (RGS5)^{10, 11, 12, 13, 14, 15, 16, 17, 18, 19, 20, 21, 22, 23, 24}, later confirmed with the development of single cell RNA sequencing (scRNAseq)^{14, 17, 20, 21, 25, 26, 27, 28, 29, 30, 31, 32, 33}. These CAF populations are phenotypically and functionally heterogeneous. FAP+ SMA+ RGS5- (also referred to as CAF-S1) fibroblasts are characterized by an accumulation of ECM proteins and inflammatory

signatures, while FAP- SMA+ RGS5+ (CAF-S4) fibroblasts are defined by a perivascular signature^{15, 16, 18, 19, 25, 34}. Among the CAF-S1 population, inflammatory CAF (iCAF) and myofibroblastic CAF (myCAF) were first discovered in pancreatic cancer^{13, 26, 35} and have now been confirmed in a high diversity of other cancer types^{17, 20, 24, 29, 34, 36, 37}. iCAF are characterized by the secretion of inflammatory mediators, while myCAF exhibit a high expression of ECM proteins but a lack of inflammatory cytokines.”

In addition, as requested by the Reviewer, to clarify our message, we also added some of the classical markers used in the renal fibrosis field, such as *PDGFRA*, *PDGFRB*, *POSTN*, *NOTCH3* in the **new Fig. 1** and **p5**.

p5: Results: “We first confirmed that almost all cells expressed *VIM* (*VIMENTIN*) and *PDGFRB*, confirming their mesenchymal origin (Fig. 1e). We also observed that high expression of *RGS5* (Regulator of G protein signaling 5) and *NOTCH3*, specific pericyte markers, were mainly detected in clusters 1 and 2, while the pan-fibroblast marker genes *PDGFRA* and *DCN* (*DECORIN*) were expressed in clusters 0, 3, 4, and 5 (**Fig. 1e**)... In agreement with these observations, *CXCL12* was highly expressed in cluster 0, and the expression of *COL1A1*, *COL3A1* and *POSTN* (main ECM components in fibrosis and marker of myofibroblasts) was mainly detected in clusters 3 and 5 (**Fig. 1f**).”

Furthermore, as requested by the Reviewer, we have now compared the original annotations from Kuppe et al by performing a label transfer between their dataset annotations and our own. This enabled us to compare the annotations in an unbiased way and to highlight the identification of a new population of stromal cells with inflammatory properties. Indeed, while mesenchymal populations were similarly annotated, the CXCL-iFibro was more precisely defined in our study thanks to the comparison with CAF populations. In more details, we found that the “pericyte-like” populations that we identified were very close to the “Pericytes” and “vascular smooth muscle cells” that Kuppe et al identified.

The Detox-iFibro we found correspond to the population Kuppe defined as fibroblast 1 (SCARA5+ MEG3+) population, and the TGFβ-myoFibro correspond to myofibroblast 1, the population characterized with the highest ECM score. In contrast, the CXCL-iFibro population we identified as an homogenous population based on the correspondence with CAF was dispersed in several clusters in the Kuppe et al. study (Fibroblast 2a (30.5% of cells), Myofibroblast 3a (22.1%), Myofibroblast 3b (17.2%), Fibroblast 2b (14.9%) and Myofibroblast 2b (12.3%)). Thus, annotating the cells regarding their role in cancer helped us to identify this functionally relevant population of CXCL-iFibro. We think that these new data help in understanding the identification and the annotations of the different mesenchymal populations. We thank the Reviewer for this comment. The method is now described in the Methods section, **p23**, and the data are now presented in new **Fig. S1d-e** and **Table S3** and described **p6-7** in the new version of our manuscript.

Concerning the OGN+ COL14A1+ population identified in Kuppe et al, it seems to correspond to our Wound-myoFibro and Detox-iFibro clusters. *OGN* is a more specific marker of these two clusters than *COL14A1*, which is expressed in almost all mesenchymal clusters independently of the annotations, as shown on the Violin plots below.

Taken as a whole, these different observations indicate that annotating mesenchymal cells regarding their role in cancer helped us to identify a functionally relevant population of CXCL-iFibro. We think that these new data help in understanding the identification and the annotations of the different mesenchymal populations. We thank the Reviewer for this comment. The label transfer is now described in the Methods section, **p23**, and the data are now presented in new **Fig. S1d-e** and **Table S3** and described **p6-7** in the new version of our manuscript.

p6-7: Results: “As Kuppe et al. described several clusters of mesenchymal cells using PDGFR β + sorted cells⁸, we sought to identify the similarities between our fibroblast annotations and those from this study (as reported in **Fig. S1d**). Hence, we performed a label transfer analysis on PDGFR β + sorted cells from⁸. The dataset of 2495 mesenchymal cells with our own annotations (described in **Fig. 1j**) was used as a reference, and the dataset of PDGFR β + sorted cells (described in **Fig. S1d**), with original annotations from⁸ was used as a query. Interestingly, the pericyte-like populations that we identified (IFNab-Peri-like and Contractile-Peri-like) showed a high degree of similarity with pericytes and vascular smooth muscle cells described by Kuppe et al (**Fig. S1e** and **Table S3**). Similarly, the Detox-iFibro we described corresponded to the Fibroblast 1 population that Kuppe described as being a SCARA5+ MEG3+ non-activated fibroblasts, and the TGF β -myoFibro was similar to the Myofibroblast 1, the population exhibiting the highest level of ECM protein secretion. In contrast, CXCL-iFibro, and to a lesser extent Wound-myofibro were identified as a mix of different fibroblast and/or myofibroblast populations, according to original annotations. In particular, the CXCL-iFibro population that we identified as being a homogenous population was dispersed throughout several clusters in the Kuppe et al. study and characterized as a mix of Fibroblast 2a (30.5% of cells), Myofibroblast 3a (22.1%), Myofibroblast 3b (17.2%), Fibroblast 2b (14.9%) and Myofibroblast 2b (12.3%) (**Fig. S1e** and **Table S3**). Thus, annotating mesenchymal cells based on similarities with CAF allowed us to identify a population of fibroblasts with inflammatory properties that has not been explored further previously.”

p23: Methods: “*Label transfer*: To identify similarities between the original annotation of cells from the PDGFR β + sorted cells single-cell RNAseq dataset from⁸ and our annotations, we used the Label Transfer algorithm using the *FindTransferAnchors* and the *TransferData* functions from Seurat. The dataset of 2495 mesenchymal cells with our own annotation described in **Fig. 1j** was used as a reference, and the dataset of PDGFR β + sorted cells, described in **Fig. S1d**, with original annotations from⁸ was used as a query. The alluvial plot was then generated using the *do_AlluvialPlot* function from SCpubr package, using the original annotations as the first group, and the predicted Id in the final group.”

2. The staining in Figure 2 (e.g. SFRP1, FAP) does not look specific to stromal cells and higher magnification and colocalization with fibroblast-specific markers is needed. In addition, negative controls (can be in supplemental methods) are critical as fibrotic tissue can often have non-specific staining.

To address the specificity concern, we performed new staining experiments and now show more images to provide additive evidence of SFRP1 and FAP staining in the stroma of kidneys from patients with fibrosis (**Fig. S2a, b**). Moreover, as requested, we also performed control staining without primary antibody, showing that the staining is not the result of a non-specific trapping of the secondary antibody in fibrotic areas (**Fig. S2a, b**) and their corresponding legends **p40**.

p8: Results: “We next sought to validate the presence of these different clusters of fibroblasts in human tissue sections at different stages of the disease (Fig. 2j-m and Fig. S2a, b for specificity of staining).”

p40: Legends of Fig. S2a, b: “(a) Additive representative images (in addition to those shown in Fig. 2j, k) to emphasize the stromal staining of SFRP1+ at low (upper panel) and high (lower panel) magnification. Left panel shows a representative image of the complete staining. The right panel shows the result of the staining using the same protocol without primary antibody, highlighting the specificity of the primary antibody. Scale bars = 1mm for the upper panel, 50mm for the middle panel, 20 mm for the lower panel. (b) Same as (a) but for FAP. Scale bars = 1mm for the upper panel, 50mm for the middle panel, 20 mm for the lower panel.”

As indicated by the Reviewer, we observed that some tubular cells are positive for SFRP1 staining. To address the Reviewer’s concern, we performed new co-staining experiments, and we quantified the double positivity of SFRP1+ cells and α SMA+ cells, on the one hand, and of FAP+ cells and α SMA+ cells, on the other hand. By this way, we validated that the majority of SFRP1+ or FAP+ cells are indeed fibroblasts. These data are now shown in Fig. S2c and Fig. S2d and described p8.

p8: Results: “Co-staining experiment between SFRP1 or FAP and α SMA showed that SFRP1+ and FAP+ cells were mainly α SMA+, defining that they were activated fibroblasts (Fig. S2c, d).”

p40: Legends Fig. S2c, d: “(c) Representative images (left panel) and quantification (right panel) of IF showing co-staining between SFRP1 (red) and α SMA (green). Quantification shows the percentage of SFRP1+ α SMA+ double positive interstitial cells. N = 6 PKD patients. Low panels of images show the image resulting from the same protocol without primary antibody. (d) Same as in (c) but for FAP and α SMA co-staining.”

We also would like to emphasize that some tubular cells can experience partial EMT after kidney injury and express mesenchymal markers (such as Vimentin or α SMA for example^{42,43}). It is therefore possible that some tubular cells express some mesenchymal markers, including SFRP1 in fibrotic samples. To be complete in our description, we also added this notion in the new version of the Text, p8.

p8: Results: “We also observed that some tubular cells could also be positive for SFRP1 (Fig. 2j and Fig. 2l). Consistent with this, it is known that, during CKD, some tubular cells could express some mesenchymal markers (such as VIM or α SMA) and undergo partial epithelial to mesenchymal transition^{42,43}.”

3. The data related to Wnt/beta-catenin is a bit weak. The nuclear staining of beta-catenin (Fig 5g) is not too convincing. The beta-catenin is so ubiquitous that without confocal, it is difficult to determine how much is nuclear. To strengthen the case for active beta-catenin in fibroblasts co-cultured with monocytes, would recommend westerns of nuclear isolates and/or qPCR of axin2, a well-established target of beta-catenin/TCF/LEF. The effect of beta-catenin inhibitor on SFRP4 would be more convincing with a Western. The inhibitor blocks beta-catenin interactions with TCF, and these interactions have been shown to be important for myofibroblast activation. However, they are not necessarily dependent upon Wnt from macrophages (another signal could lead to signaling that augments beta-catenin/TCF signaling such as TGF-beta). Wnt inhibitors (inhibitor of Wnt acyltransferase PORCN) would better test if Wnt signaling were needed for this process.

We thank the Reviewer for highlighting this concern. As requested, we have now provided a substantial number of additive experiments to demonstrate the role of the WNT/ β -catenin pathway in the crosstalk between fibroblasts and myeloid cells.

First, as requested by the Reviewer, we have now performed cytoplasmic and nuclear extracts followed by western blot experiments to demonstrate the translocation of the β -catenin in the nucleus of fibroblasts after co-culture with monocytes. To do so, we have co-cultured fibroblasts and CD14+ monocytes, isolated fibroblasts, performed cytoplasmic and nuclear extracts from fibroblasts and analyzed β -catenin nuclear fraction, using Histone H3 and EIF4A1 proteins, as nuclear and cytoplasmic controls, respectively (protocol described in the **Methods** section p29). By this way, we observed that co-culture of fibroblasts with CD14+ monocytes stimulated the translocation of β -catenin in the nucleus of fibroblasts. Using the exact same protocols, we also now demonstrate that the treatments with β -

catenin inhibitor and PORCN inhibitor significantly reduce β -catenin nuclear translocation in fibroblasts upon co-culture with monocytes. Moreover, following inhibition, we now also confirm by western blots that CXCL-iFibro do not experience any increase in either α SMA or SFRP4 staining when cocultured with CD14⁺ monocytes. All these data are now shown in the **new Fig. 5h, i** (β -catenin nuclear translocation) and **new Fig. 5j, k** (α SMA and SFRP4 regulation) and described in the Results section **p14**, and corresponding legends **p36-37**.

Finally, as recommended by the Reviewer, we have now provided higher magnifications of images in **Fig. 5a**, **Fig. 5f** and **Fig. 5i** to better visualize α SMA and SFRP4 staining in fibroblast in the different culture conditions (plastic/collagen; WNT agonists; presence of monocytes +/- β -catenin inhibitor).

p14: Results: “Without treatment, we confirmed by both IF and western blots that co-culture of CXCL-iFibro with CD14⁺ monocytes promoted the nuclear translocation of β -catenin in fibroblasts (**Fig. 5h, i**). Moreover, this was concomitant to the up-regulation of ECM-secreting myoFibro markers (**Fig. 5j, k**). Interestingly, we observed that inhibition of β -catenin/TCF interaction prevented the macrophage-induced switch from CXCL-iFibro into ECM-secreting myoFibro (**Fig. 5j, k**). Indeed, following inhibition, CXCL-iFibro showed a significant reduction of β -catenin nuclear translocation in fibroblasts and did not experience any increase in either α SMA or SFRP4 staining when cocultured with CD14⁺ monocytes (**Fig. 5i, k**), indicating that the WNT/ β -catenin pathway was required for the differentiation of CXCL-iFibro into ECM-secreting myoFibro. Finally, because macrophages have been shown to secrete WNT ligands to promote tissue repair in the intestine or kidney, we confirmed that blocking WNT ligand secretion by C-59 (a PORCN inhibitor) reduced β -catenin nuclear translocation and prevented the fibroblast phenotypic switch from CXCL-iFibro into ECM-secreting myoFibro (**Fig. 5i, k**). Altogether, these data show that macrophages stimulate the differentiation of the CXCL-iFibro into ECM-secreting myoFibro through activation of the WNT/ β -catenin pathway. .”

p29: Methods: “**Nuclear translocation of β -catenin in fibroblasts upon co-culture with CD14⁺ PBMC evaluated by western blots.** 5×10^4 fibroblasts were seeded in 24-well plates coated with type I collagen in DMEM supplemented with 10% heat-inactivated FBS and 1% Penicillin streptomycin at 1.5% O₂ overnight for complete adherence. Then, the medium was removed and 2.5×10^5 CD14⁺ monocytes resuspended in 500 μ l of DMEM supplemented with 1% heat inactivated FBS and 1% Penicillin streptomycin were added. At that time, cells were stimulated with either DMSO or β -catenin/TCF Inhibitor III, iCRT3 (Sigma Aldrich, 219332) or PORCN inhibitor, C59 (Sigma Aldrich, 5004960001) at the dose of 20 μ M. 24h after co-culture, adherent and non-adherent cells were harvested, and the fibroblasts were separated from the monocytes by negative selection using a specific isolation kit (Miltenyi #130-050-201). Nuclear and cytoplasmic fractionation was performed by following the manufacturer’s protocol for NE-PER™ Nuclear and Cytoplasmic Extraction reagents (ThermoFisher #78833). Briefly, fibroblasts were collected in 1,5 ml microcentrifuge tubes and washed with PBS, and ice-cold cytoplasmic reagents (CER I and II) were added. Cells were vigorously vortexed at the highest speed and centrifuged at 16,000g for 10 min; then the supernatant containing the cytoplasmic extracts was transferred to a new ice-cold tube. The cell pellet was resuspended with a Nuclear extraction reagent (NER), vortexed and the extract was obtained after centrifugation. All the steps were performed on ice and the samples were next sonicated for 15 min (cycles of 30 sec ON / 30 sec OFF) and centrifuged during 10 min at 13.000 x g at 4°C. The western blot was then performed as detailed in **#Protein extraction and western blot**. EIF4A1 and Histone-H3 were used as the respective cytoplasmic and nuclear makers for monitoring fraction quality by Western blot. Primary antibodies used were EIF4A1 (1:1000, Cell signaling #2490), Histone H3 (1:10000, abcam #ab1791) and β -catenin (1:1000, cell signaling #9562).”

4. Is the CXCL-iFibro high/low predictive of patient outcome independent of macrophage number? Is this marker providing any information beyond what CD68⁺ cells would provide?

We thank the Reviewer for raising this interesting point. We checked CD68 expression in the NEPTUNE dataset. Unfortunately, CD68 mRNA data did not pass the quality control and could not be used for further analysis. Still, to answer the Reviewer’s concern, we had access to the expression rate of *CD14* (a marker of monocytes, macrophages and dendritic cells) and of *MRC1* (marker of “M2” macrophages, which has been suggested to participate in CKD progression^{71, 72}). By univariate analysis, we identified

that neither MRC1 nor CD14 were associated with poor patient outcome. Using a multivariate Cox model, we now show that CXCL-iFibro expression score and FOLR2 expression were associated with kidney outcomes, independently of MRC1 expression. We thank the Reviewer for these interesting comments, which improve our manuscript. These data are now shown (**Fig. S8f, g**) and described **p16**.

p16: Results: “Finally, we wanted to assess if the ability of CXCL-iFibro expression score or FOLR2+ macrophages expression to predict kidney outcome was independent of the expression of other monocytes/macrophages markers. To do so, we evaluated the role on CKD progression of CD14, a wide marker of myeloid cells (monocytes, macrophages, and dendritic cells), and MRC1, a marker of M2-like macrophages, suspected to be involved in CKD progression^{71,72}. Interestingly, neither CD14 nor MRC1 were associated with poor patient outcome in univariate analysis (p=0.14 for CD14, p=0.095 for MRC1). As for MRC1, the p-value was less than 0.1, we added it in the multivariate Cox model. Interestingly, the CXCL-iFibro expression score and FOLR2 expression remained significantly associated with kidney outcomes (**Fig. S8f, g**). These data suggest that the early identification of CXCL-iFibro or FOLR2+ macrophages could refine the risk prediction of CKD progression, independently of the number of CD14+ cells or MRC1+ cells in the kidney.

We therefore determined that identifying the presence of CXCL-iFibro or FOLR2+ macrophages at early stages of CKD allows a more precise characterization of the risk of progression than identifying “M2” macrophages. Thus, we think that we provide here data that support our pathophysiological hypothesis, i.e. CXCL-iFibro attract and modify CD14+ PBMC to switch them into FOLR2+ macrophages. Altogether, we pinpoint a pathophysiological process that could represent promising therapeutic targets. Validating prospectively the ability of CXCL-iFibro or FOLR2+ macrophages capacity as biomarkers will make the object of follow-up studies. We now discuss this point in the new version of the manuscript, in the Discussion section, **p19**.

p19: Discussion: “Similarly, identifying the right threshold of CXCL-iFibro expression that most efficiently discriminates between patients with low or high risk of CKD progression should be assessed in a specific follow-up prospective study. Despite these relative limitations, our study gives a strong argument in favor of the early role of CXCL-iFibro in fibrosis development.”

Minor:

1. *The collagen plated fibroblasts are SFRP1+ and do not seem to be “activated”. However, to enhance confidence that these accurately model the fibroblast population, some qPCR data showing a proinflammatory phenotype would be useful.*

We agree with the reviewer that identifying the pro-inflammatory phenotype of collagen-cultured fibroblasts is an interesting point. To do so, we performed RNA sequencing of patient-derived fibroblasts maintained in culture by using different conditions either on collagen- (n=3) or on plastic-dishes (n=2). We identified several pro-inflammatory genes, such as *CXCL12*, *IL1B* and *IL34*, which are up-regulated in collagen-cultured fibroblasts. As requested, these new data are now showed in **Fig. S5a, b**, and described in the manuscript **p12**.

p12: Results: “We then performed RNA sequencing on collagen- or plastic-cultured fibroblasts to confirm the respective identity of these cells. Consistent with the inflammatory identity detected in CXCL-iFibro, we identified several inflammatory genes, including *CXCL12*, *IL1B* and *IL34*, upregulated in collagen-cultured compared to plastic-cultured cells (**Fig. S5a, b**). These proteins are of interest, because of their role in chemoattraction of immune cells (*CXCL12*)⁵⁸, pro-inflammatory response (*IL1B*)^{59,60} or in promoting the differentiation and viability of monocytes and macrophages through the colony-stimulating factor-1 receptor (*IL34*)⁶¹. Similarly, consistent with their ECM-secreting myoFibro identity, plastic- cultured fibroblasts exhibited increased expression of ECM-related genes, such as *COL4A1* and *COL4A5* (**Fig. S5d**), which are known to be important in the fibrotic process and basement membrane integrity⁶². We also identified other components of the ECM, belonging to the integrin (*IGFBP1*) or the laminin (*LAMA3*) families (**Fig. S5d**). Altogether, these data suggest that collagen-cultured and plastic-cultured fibroblasts are reminiscent of CXCL-iFibro and ECM-secreting myoFibro, respectively.”

2. *It wasn't clear what the difference is between FOLR2 resident macrophages and FOLR2 CKD macrophages. Functionally, are FOLR2 macrophages proinflammatory or profibrotic?*

As underlined by the Reviewer, we found interesting to identify 2 clusters of FOLR2+ macrophages that are differently distributed between control and CKD patients, as shown by single cell analysis. Another argument to confirm that these clusters identified 2 different cellular populations comes from the spatial transcriptomics experiment. Indeed, we identified that these 2 clusters were spatially differentially localized, with FOLR2+ CKD macrophages in the same spots as CXCL-iFibro, but not FOLR2 resident macrophages. As requested, we added this mention in the new version of the manuscript, **p11**.

p11: Results: “Strikingly, we observed that FOLR2+CKD macrophages mainly colocalized with CXCL-iFibro and plasma cells (Fig. 3l, m and Fig. S4b, c). On the other hand, other fibroblast subsets (IFN α β -peri like, Detox-iFibro, Wound-myofibro and TGF β -myofibro colocalize together with monocytes, FOLR2 resident macrophages, B cells, T cells, dendritic cells and TREM2+ macrophages (Fig. 3l, m).”

Differential gene expression and functional enrichment analysis identified that FOLR2 CKD macrophages show an inflammatory phenotype, with enrichment in the cellular response to stress, inflammatory response and chemotaxis, while FOLR2 resident macrophages display a more “scavenging” phenotype, with activation of RHO-GTPase pathway, regulation of protein catabolic process, cytokine production and regulation of proteolysis. As recommended by the Reviewer, we added this interesting data in the new version of our manuscript in **Table S4** and described **p10**.

p10: Results: “Differential gene expression and functional enrichment analysis confirmed that these 2 FOLR2+ populations were also characterized by a specific transcriptional profile. FOLR2+ resident macrophages showed activation of RHO-GTPase pathway, regulation of protein catabolic process, cytokine production and regulation of proteolysis, potentially corresponding to a scavenging-macrophage phenotype, while FOLR2+ CKD macrophages showed an enrichment in the cellular response to stress, inflammatory response and chemotaxis, corresponding to a more inflammatory phenotype (**Table S4**).”

3. *Which Wnt ligand is used for “wnt agonist 1/2”?*

We used 2 Wnt agonists, for which we can provide the following information found on the datasheets:

Wnt agonist 1 (Sigma-Aldrich 681665, CAS 853220-52-7): A cell-permeable pyrimidine compound that acts as a potent and selective activator of Wnt signaling without inhibiting the activity of GSK-3 β (IC₅₀ >60 μ M). It is shown to mimic the effect of Wnt and induce β -catenin and TCF (T-cell fate)-dependent transcriptional activity (EC₅₀ = 700 nM in HEK-293T cells)

Wnt agonist 2 (Sigma-Aldrich 681667, SKL2001): A cell-permeable imidazolyl-isoxazolamide compound that upregulates β -catenin-regulated transcription (CRT; Effective conc. 10 - 40 μ M) by disrupting β -catenin and Axin interaction, thereby preventing β -catenin phosphorylation (Ser33/Ser37/Thr41/Ser45) and proteasomal degradation, without affecting the activities of GSK-3 α/β or 18 other kinases (\leq 8.5% inhibition by 10 μ M SKL2001). Wnt pathway activation by SKL2001 is shown to effectively promote osteoblastogenesis in both human and murine mesenchymal cultures (10 to 40 μ M) as well as suppress MDI- (dexamethasone and insulin) stimulated adipogenesis of murine 3T3-L1 preadipocytes (5 to 30 μ M).

As requested, we have now added these information in the new version of the manuscript, in the method section, **p28**.

p28: Methods' section: “Wnt agonist 1 is a cell-permeable pyrimidine compound that acts as a potent and selective activator of Wnt signaling without inhibiting the activity of GSK-3 β . It is shown to mimic the effect of Wnt and induce β -catenin and TCF (T-cell fate)-dependent transcriptional activity. Wnt agonist 2 is a cell-permeable imidazolyl-isoxazolamide compound that upregulates β -catenin-regulated transcription by disrupting β -catenin and Axin interaction, thereby preventing β -catenin phosphorylation (Ser33/Ser37/Thr41/Ser45) and proteasomal degradation, without affecting the activities of GSK-3 α/β .”

4. *The Neptune database is great, but it is primarily diseases with nephrotic syndrome (not diabetes, hypertension). Some discussion about how inflammation (and potentially inflammatory fibroblasts) may differ in this population versus others.*

We agree with the reviewer and add some discussion about this point in the new version of the manuscript, **p19**. **p19: Discussion:** “Datasets combining kidney transcriptomic data with clinical and biological follow-up are quite scarce, but we had the opportunity to study the NEPTUNE cohort, which focuses on patients with glomerular diseases and nephrotic syndromes, including patients with CKD. Here, we demonstrate that the detection of CXCL-iFibro at early stages of CKD is clinically relevant. Indeed, very interestingly, we observed that the presence of CXCL-iFibro in patients with mild- to moderate-CKD is predictive of poor patient outcome. Whether this observation is similar in other diseases, such as diabetic or vascular nephropathy, needs to be validated in prospective studies. Similarly, identifying the right threshold of CXCL-iFibro expression that most efficiently discriminates between patients with low or high risk of CKD progression should be assessed in a specific follow-up prospective study. Despite these relative limitations, our study gives a strong argument in favor of the early role of CXCL-iFibro in fibrosis development. .”

5. The authors should be cautioned against interpreting CXCL-iFibro as being causative in CKD progression- the data is more consistent with its use as a marker of progression.

We modified our conclusions accordingly to reviewer’s remark. As recommended, we have now softened our conclusions, **p16**.

p16: Results: “Altogether, these data show that CXCL-iFibro and FOLR2+ macrophages are interconnected biomarkers in CKD progression and demonstrate that our findings could be relevant in clinical practice.

6. CXCL-iFibro as a prognostic marker is attractive. A weakness to be acknowledged is that the population that was tested to determine high/low values has not been validated prospectively. It seems that the threshold was chosen based upon the lowest p value.

We agree that we did not validate this marker prospectively, and that this needs to be the purpose of a follow up study. The reviewer is right saying that the threshold presented in the Figure 6 is based upon the lowest p-value. However, we also show the entire set of data with all thresholds and their corresponding p-values in **Fig. S8b**. This iteration study indicates that several thresholds are statistically significant and associated with $p < 0.05$, thereby re-enforcing our conclusion. We completed these data by now providing additive means of presenting the survival curves in **Fig. S8a** and **Fig. S8c**, where we show the survival relative to the quartiles of expression of CXCL-iFibro expression score or FOLR2 expression. These data show that the level of expression of CXCL-iFibro is clearly associated with poor patient outcome, and that we did not specifically select the only threshold allowing to detect a difference. These data are presented in the new **Fig. S8a-c** and described **p15**. We also discussed this point in the Discussion section, **p19**.

p15: Results: “Optimal stratification of patients for survival analyses was performed using an iterative method. Patients were ranked by their expression of CXCL-iFibro transcriptomic signature and thresholds were defined for each level of expression (**Fig. S8b, Table S11**). For each threshold, patients were separated into low and high CXCL-iFibro expression scores and a Log-rank test was applied. We selected the threshold, which displayed the most significant p-value and separated 50 patients with low-score from 84 patients with high-score (**Fig. S8b, Table S11**).

p19: Discussion: “. Datasets combining kidney transcriptomic data with clinical and biological follow-up are quite scarce, but we had the opportunity to study the NEPTUNE cohort, which focuses on patients with glomerular diseases and nephrotic syndromes, including patients with CKD. Here, we demonstrate that the detection of CXCL-iFibro at early stages of CKD is clinically relevant. Indeed, very interestingly, we observed that the presence of CXCL-iFibro in patients with mild- to moderate-CKD is predictive of poor patient outcome. Whether this observation is similar in other diseases, such as diabetic or vascular nephropathy, needs to be validated in prospective studies. Similarly, identifying the right threshold of CXCL-iFibro expression that most efficiently discriminates between patients with low or high risk of CKD progression should be assessed in a specific follow-up prospective study. Despite these relative limitations, our study gives a strong argument in favor of the early role of CXCL-iFibro in fibrosis development.”

REVIEWERS' COMMENTS

Reviewer #1 (Remarks to the Author):

The authors have provided additional in vivo correlative data and some in vitro mechanistic data.

Reviewer #2 (Remarks to the Author):

Thank the authors for the detailed descriptions and for resolving most of my concerns. This revised manuscript has been improved to some extent, but I still have some minor suggestions and concerns:

1. Recently, several scRNA-seq studies of UUO have been available from published studies. Authors can use scRNA-seq data of UUO to validate the observations from human beings in a pre-clinical in vivo mouse model rather than use bulk RNA-seq data by deconvolution.
2. The authors explained that some tubular cells are positive for SFRP1 staining may explained by experiencing partial EMT after kidney injury. Whether tubular cells are negative for SERP1 in normal kidney.

Reviewer #3 (Remarks to the Author):

The revised version of this manuscript has adequately addressed my concerns.

Point by point response to reviewers' comments

We thank the Editor and the Reviewers for their positive evaluation of our work. We have now included in the new version of the text the modifications in apparent for addressing the last concerns of Reviewer #2.

Reviewer #1 (Remarks to the Author):

The authors have provided additional in vivo correlative data and some in vitro mechanistic data.

We thank the reviewer for the positive evaluation of our work.

Reviewer #2 (Remarks to the Author):

Thank the authors for the detailed descriptions and for resolving most of my concerns. This revised manuscript has been improved to some extent, but I still have some minor suggestions and concerns: 1. Recently, several scRNA-seq studies of UUO have been available from published studies. Authors can use scRNA-seq data of UUO to validate the observations from human beings in a pre-clinical in vivo mouse model rather than use bulk RNA-seq data by deconvolution.

We agree with the reviewer that several scRNAseq data of UUO have been recently published. To validate our findings in mice, we have now added the result of a label transfer analysis using our dataset as reference, and a dataset from mice undergoing UUO at several time points (day2, day7 and reversal UUO). We observed an early increase of inflammatory fibroblasts, with a peak at day 7, and a decrease in the reversal UUO. This data is in line with our analysis of bulk RNAseq. We included this new result in **Fig. 2j**, and in the manuscript, **section Results, p8**:

*“Finally, to increase the resolution of our analysis, we analyzed a single cell RNAseq dataset from mice who underwent UUO, with available data for sham-operated mice, UUO day2, UUO day7 and reversal UUO. Reversal UUO corresponds to the reimplantation of the ureter in the bladder after 7 days of UUO to observe a healing phase, where a decrease in Collagen deposition, myofibroblast activation and macrophage infiltration was observed⁴². To assess the temporal dynamics of inflammatory fibroblasts in this model, we performed a label transfer analysis using our cellular atlas as reference. Interestingly, we observed a progressive increase in the proportion of inflammatory fibroblasts from day 2 after UUO to day 7 after UUO, followed by a decrease in the reversal UUO (**Fig. 2j**), confirming that inflammatory fibroblasts are associated with fibrosis expansion.”*

2. The authors explained that some tubular cells are positive for SFRP1 staining may explained by experiencing partial EMT after kidney injury. Whether tubular cells are negative for SERP1 in normal kidney.

We thank the reviewer for this comment. Indeed, in our first version, all patients included underwent a biopsy because of kidney dysfunction, meaning that they cannot be considered as healthy, even with very low degree of fibrosis. We thus now performed SFRP1 staining in 2 kidney biopsies from kidney young donors and show no signal in these kidneys. We included this new data in **Fig. S2e**, and in the **Results section, p9**:

*“We also observed that in CKD but not in normal kidneys some tubular cells could also be positive for SFRP1 (**Fig. 2k, l and Fig. S2e**).”*

Reviewer #3 (Remarks to the Author):

The revised version of this manuscript has adequately addressed my concerns.

We thank the reviewer for the positive evaluation of our work.